# Structures illustrate step-by-step mitochondrial transcription initiation

Quinten Goovaerts[1,2,4], Jiayu Shen[3,4], Brent De Wijngaert[1,2], Urmimala Basu[3], Smita S. Patel[3✉] & Kalyan Das[1,2✉]

Transcription initiation is a key regulatory step in gene expression during which RNA polymerase (RNAP) initiates RNA synthesis de novo, and the synthesized RNA at a specific length triggers the transition to the elongation phase. Mitochondria recruit a single-subunit RNAP and one or two auxiliary factors to initiate transcription. Previous studies have revealed the molecular architectures of yeast[1] and human[2] mitochondrial RNAP initiation complexes (ICs). Here we provide a comprehensive, stepwise mechanism of transcription initiation by solving high-resolution cryogenic electron microscopy (cryo-EM) structures of yeast mitochondrial RNAP and the transcription factor Mtf1 catalysing two- to eight-nucleotide RNA synthesis at single-nucleotide addition steps. The growing RNA–DNA is accommodated in the polymerase cleft by template scrunching and non-template reorganization, creating stressed intermediates. During early initiation, non-template strand scrunching and unscrunching destabilize the short two- and three-nucleotide RNAs, triggering abortive synthesis. Subsequently, the non-template reorganizes into a base-stacked staircase-like structure supporting processive five- to eight-nucleotide RNA synthesis. The expanded non-template staircase and highly scrunched template in IC8 destabilize the promoter interactions with Mtf1 to facilitate initiation bubble collapse and promoter escape for the transition from initiation to the elongation complex (EC). The series of transcription initiation steps, each guided by the interplay of multiple structural components, reveal a finely tuned mechanism for potential regulatory control.

Transcription initiation is a multistep process catalysed by DNA-dependent RNAPs. The basic steps of transcription initiation are conserved in all domains of life; however, owing to the increasing complexity of the transcription machinery from bacteriophages to humans[3–6], the exact mechanism of initiation may vary among systems. Initial RNA synthesis occurs by a conserved DNA scrunching mechanism[7–10], which enables the RNA–DNA and transcription bubble to expand in the limited space of the polymerase cleft. The stressed IC intermediates trigger off-pathway backtracking and abortive RNA synthesis during early transcription initiation steps[11–14]. However, after 8- to 12-nucleotide (nt) RNA synthesis, the stressed IC undergoes conformational changes to switch into a stable EC. The dynamic IC intermediates are challenging to characterize structurally and biochemically; hence, such states from bubble opening to the EC transition have not been systematically captured in high-resolution structures.

Mitochondrial RNAP (mtRNAP) is essential for mitochondrial DNA transcription and replication in eukaryotes[15,16]. The mtRNAPs are closely related to the single-subunit bacteriophage T7 RNAP[17]. The yeast (Saccharomyces cerevisiae) mtRNAP (y-mtRNAP; Rpo41) shares structural and functional similarities with the human mtRNAP (h-mtRNAP;

POLRMT). y-mtRNAP requires one factor, Mtf1, and h-mtRNAP requires two factors, TFAM and TFB2M[4,5,18–20]; T7 RNAP does not require any factor. We published high-resolution structures of y-mtRNAP–Mtf1 with a DNA promoter as a partially melted IC (PmIC) and a transcribing IC3 state[1]. These structures resolved the complete transcription bubble, showing base-specific interactions of Mtf1 with the conserved non-template (NT) sequence (−4) AAG (−2) in the PmIC; a complete bubble has not been traced in the IC structures of h-mtRNAP[2] or T7 RNAP[21]. The y-mtRNAP IC3 structure showed a 2-nt RNA and an incoming NTP at the polymerase active site base-paired with template +1 to +3 nucleotides.

Here we report single-particle cryo-EM structures of transcribing complexes of y-mtRNAP–Mtf1, each with a fully resolved transcription bubble and an RNA–DNA hybrid. The PmIC to IC8 structures visualize the entire transcription initiation process from promoter opening to the EC transition at single-nucleotide addition steps (Supplementary Videos 1 and 2); the IC8 intermediate is at the juncture of the IC-to-EC transition[8]. These structures enabled us to determine: the structural arrangements that help accommodate the growing RNA–DNA duplex and the transcription bubble in the

[1]Laboratory of Virology and Chemotherapy, Rega Institute for Medical Research, KU Leuven, Leuven, Belgium. [2]Department of Microbiology, Immunology and Transplantation, KU Leuven, Leuven, Belgium. [3]Department of Biochemistry and Molecular Biology, Robert Wood Johnson Medical School, Rutgers University, Piscataway, NJ, USA. [4]These authors contributed equally: Quinten Goovaerts, Jiayu Shen. ✉e-mail: patelss@rwjms.rutgers.edu; kalyan.das@kuleuven.be

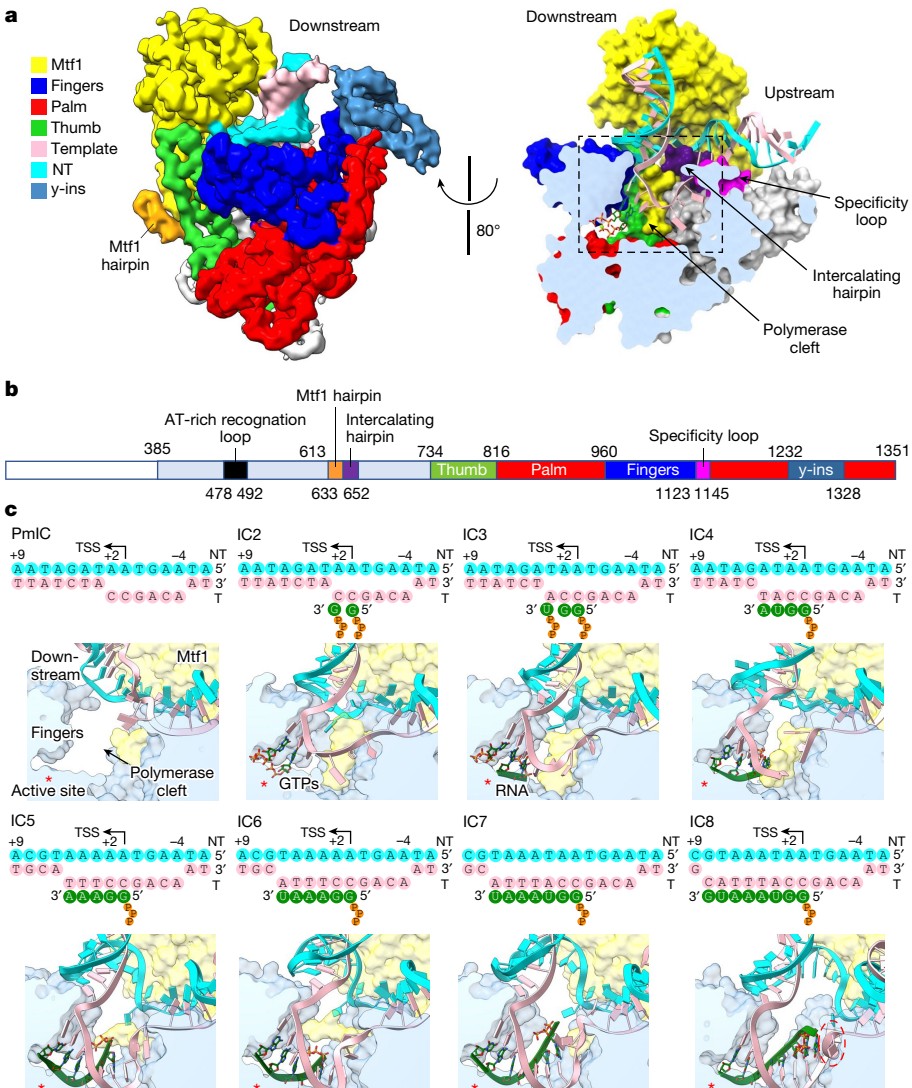

**Fig. 1 | Overview of the transcription initiation states of y-mtRNAP from PmIC to IC8 at single-nucleotide addition steps. a**, Left: the unsharpened experimental cryo-EM density map of IC2 at 1.8$\sigma$ and coloured by segment– Mtf1 (yellow) and y-mtRNAP subdomains: fingers (blue), palm (red), thumb (green), template (pink), NT (cyan), y-ins (light blue). Right: a cross-section of IC2 in surface representation viewed at about 80° showing the polymerase cleft, intercalating hairpin and specificity loop. Dashed box indicates the area displayed in all structures of part **c**. **b**, A colour-coded one-dimensional

representation of the structural elements of y-mtRNAP. **c**, A schematic representation of the transcription bubbles in PmIC to IC8. The positioning of the transcription bubbles in the polymerase clefts of the respective structures: cyan, NT; pink, template DNA; green, RNA; yellow, Mtf1; light blue, y-mtRNAP. The polymerase active site is denoted by a red asterisk, and the scrunched template in IC8 is outlined by a dashed oval. TSS, transcription start site; T, template.

polymerase cleft; the factors that govern abortive synthesis over RNA elongation, and why only short RNAs of specific length are aborted; and how the transition from initiation to elongation is triggered and orchestrated.

## Cryo-EM structures of IC2 to IC8

We used three pre-melted (−4 to +2) DNA scaffolds of a consensus yeast mitochondrial DNA promoter with slightly different coding sequences to generate the IC2–IC8 states (Extended Data Fig. 1a). The pre-melted bubble complex activity is comparable to that of the duplex promoter (Extended Data Fig. 1b,c). The ΔN100 (100 amino-terminal residues deleted) y-mtRNAP, Mtf1, and the specific DNA promoter were mixed to generate PmIC that was purified over a size-exclusion column[22]. The ΔN100 is fully active and more stable for structural studies than full-length y-mtRNAP[23]. Defined lengths of RNA–DNA hybrids were

synthesized in situ by catalytic addition of appropriate NTPs to the specific PmIC. IC2 was generated with GTP, the IC4 to IC7 states were generated by incubating a 2-nt RNA, pppGpG, and appropriate NTPs, and IC8 was generated using GTP, UTP and ATP. Single-particle cryo-EM structures of IC2, IC4, IC5, IC6, IC7 and IC8 were determined at 3.47, 3.44, 3.39, 3.62, 3.75 and 3.62 Å resolution, respectively (Extended Data Tables 1 and 2). The experimental density maps clearly resolved the transcription bubble and the surroundings in each IC state (Extended Data Fig. 1d). The N-terminal extension, which plays a regulatory role in y-mtRNAP[23], was disordered. This region in h-mtRNAP contains the tether helix and PPR domain, which interacts with upstream DNA[2]. The IC8-to-EC transition was modelled using the h-mtRNAP EC crystal structure (Protein Data Bank: 4BOC)[24] containing a 9-base-pair (bp) RNA–DNA, and these nine structural states are sequentially morphed into videos showing the dynamic process of transcription initiation and transition to elongation (Supplementary Videos 1 and 2).

The y-mtRNAP has a right-hand-shaped carboxy-terminal polymerase domain with palm, fingers and thumb, the subdomains found in all single-subunit RNAPs[25] (Fig. 1a,b). The subdomains enclose the polymerase cleft, which contains the active site for RNA synthesis and the space to accommodate the transcription bubble and RNA–DNA duplex (Fig. 1c). The Mtf1 is positioned on the top of the polymerase cleft like a lid (Fig. 1a). The NT strand in the bubble region interacts primarily with Mtf1 and the template strand interacts primarily with y-mtRNAP. y-mtRNAP and Mtf1 are held together through two sets of interactions: that of the Mtf1 N-terminal domain (residues 1–252) with the y-mtRNAP thumb tip (residues 772–779) and that of the Mtf1 C-terminal domain (residues 255–341) with a y-mtRNAP hairpin (residues 613–633), termed the Mtf1 hairpin. The Mtf1 hairpin homologue lever loop/B2 hairpin in h-mtRNAP supports TFB2M binding[1,2].

The y-ins domain of about 100 amino acids (1232–1328) is a unique insertion in y-mtRNAP that is absent in T7 RNAP and h-mtRNAP. Owing to the dynamic behaviour of the downstream DNA and its surroundings, y-ins has a lower resolution than the core. The AlphaFold[26] model for y-ins fitted reliably to the density (Extended Data Fig. 2a), showing a four-helix bundle and a small two-stranded antiparallel β-sheet. A flexible cleft formed by the N-terminal domain of Mtf1 and the y-ins that we refer to as the downstream cleft interacts with and supports the downstream promoter (Extended Data Fig. 2b). The y-ins domain interacts nonspecifically with the downstream DNA backbone; the positional adaptability of y-ins allows the downstream cleft to accommodate the DNA duplex as the transcription bubble expands within the adjacent polymerase cleft. Biochemically, the Δy-ins mutant is inactive for transcription initiation (Extended Data Fig. 1c), suggesting its greater role in supporting the duplex promoter for transcription bubble opening.

## IC2 and IC3 capture pre-catalytic states

IC2 was captured with two GTP molecules bound at the transcription start site poised for catalysis (Fig. 2a). The +1 GTP is bound at the priming site (P site, post-insertion site), and the +2 GTP is bound at the NTP-binding site (N site, insertion site). One Mg[2+] ion chelates the phosphates of the +2 GTP, and the second ion is not resolved. Considering the abortive 2-nt synthesis cycle (Fig. 2b), we expected to trap one or more of the following states in the absence of +3 NTP: a synthesized 2-nt pppGpG RNA base-paired with the template +1 and +2 nucleotides; two GTPs bound at the +1 and +2 positions before catalysis; and/or the PmIC, after release of the 2-nt RNA as an abortive product. The in vitro transcription assays on the bubble promoter (Extended Data Fig. 1b) showed a substantial amount of 2-nt RNA product from the supplied GTP, indicating that the GTP-bound IC2 is captured in the cryo-EM sample under steady-state RNA synthesis conditions. The absence of the 2-nt RNA at the active site indicates that the GTP-bound IC2 is more stable than the 2-nt RNA-bound state. Following 2-nt RNA dissociation, the initiating nucleotides bind rapidly and deplete the PmIC state (Fig. 2b). By contrast, IC3 was captured with PmIC coexisting in solution[1], most likely because both 2-nt RNA + UTPαS binding and 3-nt synthesis are slow steps.

The transition from PmIC to IC2 involves bubble opening, leading to the exposure of the +1/+2 template bases that align with the two initiating GTPs positioned at the active site (Fig. 2a). This transition occurs with a large structural rearrangement whereby the −1, +1 and +2 template nucleotides at their N1 positions shift by about 20, 27 and 22 Å, respectively, and the downstream DNA bends by an additional 60° with respect to the upstream DNA. To support the fully open bubble and DNA bending, the Mtf1 N-terminal domain moves towards the y-ins domain by about 7 Å (Extended Data Fig. 3a) and closes the cleft around the downstream DNA. Harmoniously, the tip of the y-mtRNAP thumb, interacting with Mtf1 (residues 105, 153–158), moves by about 5 Å to maintain the protein–protein interactions. The downstream cleft

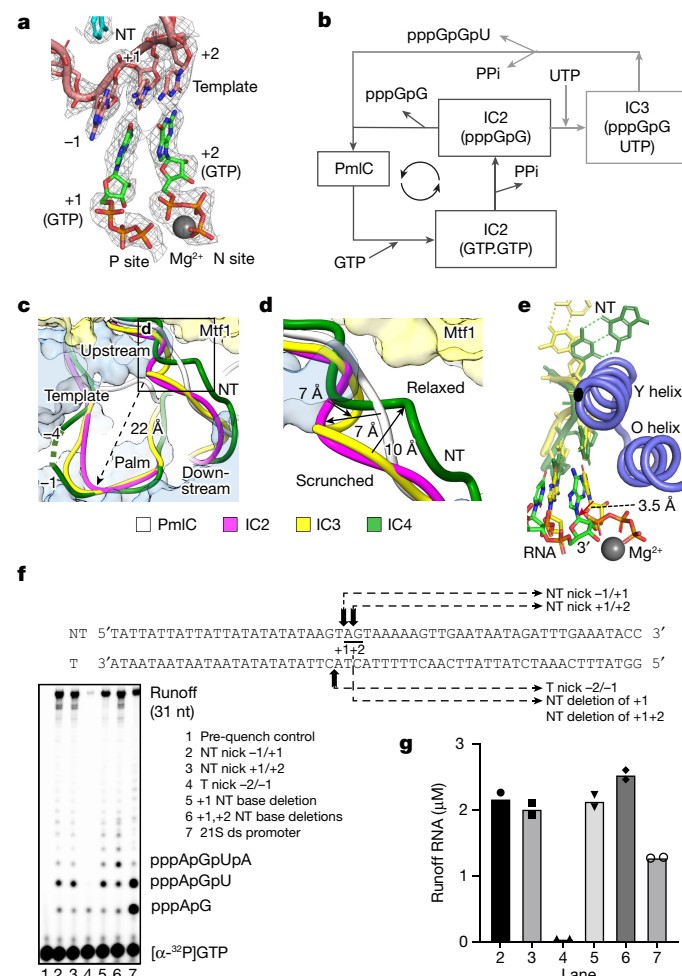

**Fig. 2 | Structures of IC2 to IC4 states reveal the basis for abortive synthesis. a**, A cryo-EM density map of IC2 at 2.5σ defining the GTPs (green) at the polymerase active site base-paired with the template (pink) and poised for catalysis; Mg[2+] ion in grey. **b**, A schematic representation of IC2 and IC3 abortive cycles. The PmIC binds the initiating GTP molecules to form IC2 (2GTPs). The phosphodiester bond formation between 2 GTPs converts IC2 (GTP.GTP) to IC2 (pppGpG), releasing a pyrophosphate (PPi). In the absence of a next NTP, the 2-nt RNA dissociates, and IC2 returns to PmIC, to continue a cyclic process that leads to the accumulation of 2-nt pppGpG RNA (Extended Data Fig. 1b). A similar abortive cycle occurs at IC3 that synthesizes 3-nt RNA, pppGpGpU, from pppGpG and UTP. **c**, Stepwise scrunching of template (bottom) and NT (top) in the polymerase cleft as the transcription bubble grows. **d**, A zoomed view of the area outlined in **c**, showing the NT strand scrunched into loops in IC2 and IC3 and then relaxed in IC4. **e**, RNAP Cα superposition of IC3 (yellow) and IC4 (green) showing that the 3′ end of the RNA in IC4 has moved halfway (about 3.5 Å) towards the P site from the N site; the RNAP fingers helices of IC4 are in blue. **f**, Transcription runoff synthesis on nicked and nucleotide-deleted 21S promoters with y-mtRNAP and Mtf1 in the presence of ATP, GTP, UTP and 3′-dCTP at 25 °C for 15 min. ds, double-stranded. **g**, Amount of runoff products from nicked, nucleotide-deleted and unaltered promoters. The data points from two biological repeats are shown.

starts opening in IC3 and continues to open further in IC4; the positions of y-mtRNAP and Mtf1 in IC4 align with those in PmIC (Extended Data Fig. 3b,c).

## NT scrunching drives abortive synthesis

Abortive synthesis is an off-pathway event of transcription initiation occurring in all cellular RNAPs, including bacterial and bacteriophage T7 RNAPs[14,27–29]. Unlike in T7 and bacterial systems in which transcription

is aborted at all IC states, y-mtRNAP–Mtf1 produces mostly 2- and 3-nt RNA abortives; the RNA synthesis after 3 nt becomes relatively processive (Extended Data Fig. 1c). In bacterial RNAPs, template scrunching and steric clashes of structural elements inside the polymerase cleft create stressed intermediates resulting in abortive RNA synthesis[8,14,30,31].

The template DNA adopts a U-shaped structure in the y-mtRNAP cleft starting from IC2 (Fig. 2c). This U-shaped structure is conserved in the T7 RNAP IC[21] and h-mtRNAP EC[24], suggesting an overlapping template track in single-subunit RNAP ICs. As the RNA–DNA hybrid grows in size from IC2 to IC3 to IC4, the template nucleotides from −4 to −1 are increasingly scrunched. This template scrunching is highest in IC4 among the three states (Fig. 2c), yet IC4 is substantially less abortive, suggesting that template scrunching is not the primary contributor to the dissociation of 2- and 3-nt RNAs. Notably, the NT strand scrunches in IC2 and IC3 into tight loops and relaxes in IC4 (Fig. 2d and Extended Data Fig. 4). The scrunched NT loop in IC2 and IC3 is stabilized by interactions with the y-mtRNAP intercalating hairpin (641–642), thumb (780–787) and Mtf1 C-tail (334–336) residues. Studies show that partial deletion of the C tail reduces abortive synthesis[29], presumably by relaxing the scrunched NT in IC2. The IC2 and IC3 with scrunched NT are in high-energy states, and unscrunching, like unwinding of a watch spring, would release stress energy. The fewer interactions of short RNAs with y-mtRNAP and template make the IC2 and IC3 states less stable, increasing the probability of dissociating these short RNAs and returning to PmIC (Fig. 2b).

In IC4, the NT strand (+1 to +4) has a poor density and no noteworthy interactions with protein atoms. The B-factor of the NT region (126 Å$^2$) is about 1.6× the average (80 Å$^2$) for the structure; IC3 has a B-factor of about 0.9× the average for the region. The NT strand in IC4 carries less stress energy and favours RNA extension over abortive synthesis. Notably, the 4-bp RNA–DNA duplex in IC4 is in a half-translocated state; that is, the 3′ end of the nascent RNA has moved by about 3.5 Å at the C1′ position from the N site but has not reached the P site (Fig. 2e). No stable PmIC or other intermediate state was observed coexisting in the IC4 cryo-EM sample.

To test the hypothesis that scrunching and unscrunching of the NT loop cause abortive synthesis of 2- and 3-nt RNAs, we introduced nicks and nucleotide deletions to disrupt NT loop formation as investigated experimentally in T7 RNAP studies[32]. Transcription runoff assays with the modified promoters show markedly reduced abortive products from nicked and nucleotide-deleted NTs (Fig. 2f); the promoter with a nick in the NT strand between −1/+1 or +1/+2 makes 30-fold fewer 2-nt, sixfold fewer 3-nt abortives and about twofold more runoff products compared to the unmodified promoter (Fig. 2g). These data provide evidence that NT scrunching causes abortive synthesis. Notably, a nick in the template strand between −2/−1 impairs all RNA synthesis. By contrast, a nicked template in the T7 promoter showed little impact on transcription initiation[32]. The different outcomes in y-mtRNAP versus T7 RNAP, which does not involve an initiation factor, may be due to differences in their promoter opening mechanisms.

## NT stacks as an ordered structure in IC5

The 5-bp RNA–DNA hybrid in IC5 is in a pre-translocated state with the RNA 3′ end at the N site (Fig. 3a,b). Unexpectedly, the downstream +6 base pair is already melted in IC5. Thereby, IC5-to-IC6 transition would require less energy to flip the unpaired +6 template nucleotide into the polymerase cleft for facilitating the next nucleotide addition to form the 6-bp RNA–DNA. Major changes are observed for the C tail of Mtf1 and the NT strand in the IC4-to-IC5 transition. In the IC2–IC4 states, the C tail is positioned between the RNA–DNA duplex and the thumb subdomain, and the C tail is involved in template alignment and DNA scrunching[29] (Extended Data Fig. 5). In IC5, the 5-bp RNA–DNA pushes the C tail of Mtf1 to a new location at the centre of the transcription bubble (Fig. 3c). Simultaneously, the NT strand moves by about 15 Å

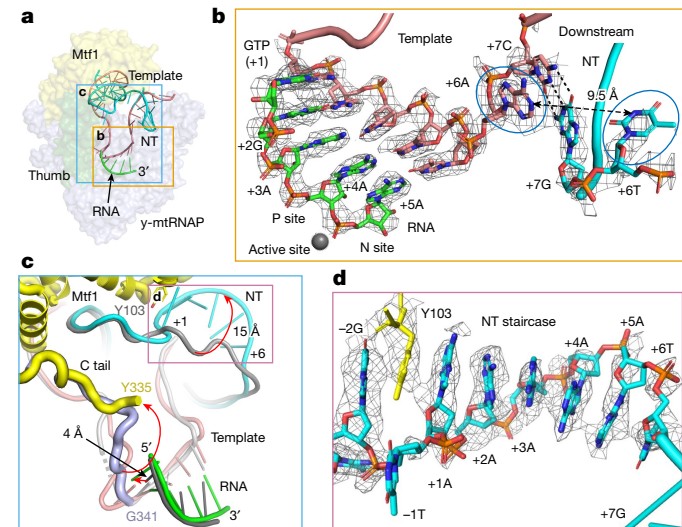

**Fig. 3 | Structural rearrangements in IC4-to-IC5 transition. a**, The transcription bubble in IC5 (green, RNA; blue NT; pink, template) enclosed by proteins shown as transparent surfaces (Mtf1, yellow; thumb, green; y-mtRNAP, light blue). This colour scheme is maintained in all individual structures from IC2 to IC8. In the figures comparing two structures, the leading structures follow the colour scheme, and the trailing structures are in grey or light blue. **b**, Experimental density for the 5-bp RNA–DNA and the unpaired sixth template and NT bases (in blue circles) at 3σ and 2.3σ, respectively, in IC5. Minimum structural rearrangements would be required for the sixth template nucleotide to enter the polymerase cleft for transition to IC6. **c**, Repositioning of the NT strand and C tail of Mtf1 during the IC4-to-IC5 transition. The transcription bubble in IC5 (green, RNA; cyan, NT; pink, template; yellow, C tail) and that in IC4 (in greyscale; dark grey, RNA; grey, NT or template; and blue, C tail) are compared. The growing RNA in IC5 pushes the C tail towards the centre of the polymerase cleft and repositions NT. The rearrangement of these structural elements from PmIC to IC8 is depicted in Extended Data Fig. 5. **d**, The NT strand bases +6 to +1 stack into a spiral staircase-like structure in IC5. The stacking extends upstream towards the −2G NT base and involve the aromatic side chain of Mtf1 Y103; the experimental density covering the NT strand and Mtf1 Y103 is shown at 2.3σ.

from the centre of the polymerase cleft in IC4 towards the N-terminal domain of Mtf1 in IC5. The repositioned NT strand forms a distinctive stacked structure resembling a spiral staircase, in which the nucleotides +1 to +6 are clearly discernible by the cryo-EM density map (Fig. 3d). Notably, the melted +6 NT base in IC5 is part of the staircase; the stabilization gained from stacking might be the driving force for pre-melting of the +6 base pair.

The NT staircase extends upstream to stack with the aromatic rings of Mtf1 Y103 and the NT (−2)G base (Fig. 3d). The Y103A substitution reduces runoff synthesis by about 100-fold (Extended Data Fig. 6a). This marked effect occurs because Y103 is involved early in the promoter melting at the PmIC state[1] and continues to stabilize the melted NT by stacking with the conserved NT (−2)G base in all IC states. The NT staircase is supported by backbone interactions with the F16/Y18 groove in the Mtf1 N-terminal domain (Extended Data Fig. 6b,c). The F16/Y18 groove also interacts with the downstream DNA in earlier IC states (Extended Data Fig. 6d,e). Thus, F16A/Y18A substitution reduces runoff RNA synthesis (Extended Data Fig. 6a). The +1-adenine base in the staircase has salt-bridge interactions with Mtf1 D101, and D101A results in an approximately 30% drop in runoff product. This suggests that the conserved NT +1-adenine in y-mt promoters is primarily essential at the later stages of transcription initiation. In general, the impact of a substitution is stronger at an early stage than at later stages of initiation. In 11 y-mt promoters, the +1 to +8 NT sequence is rich in adenines[33], which stack better than thymines[34]. Substituting +3 to +8 NT adenines

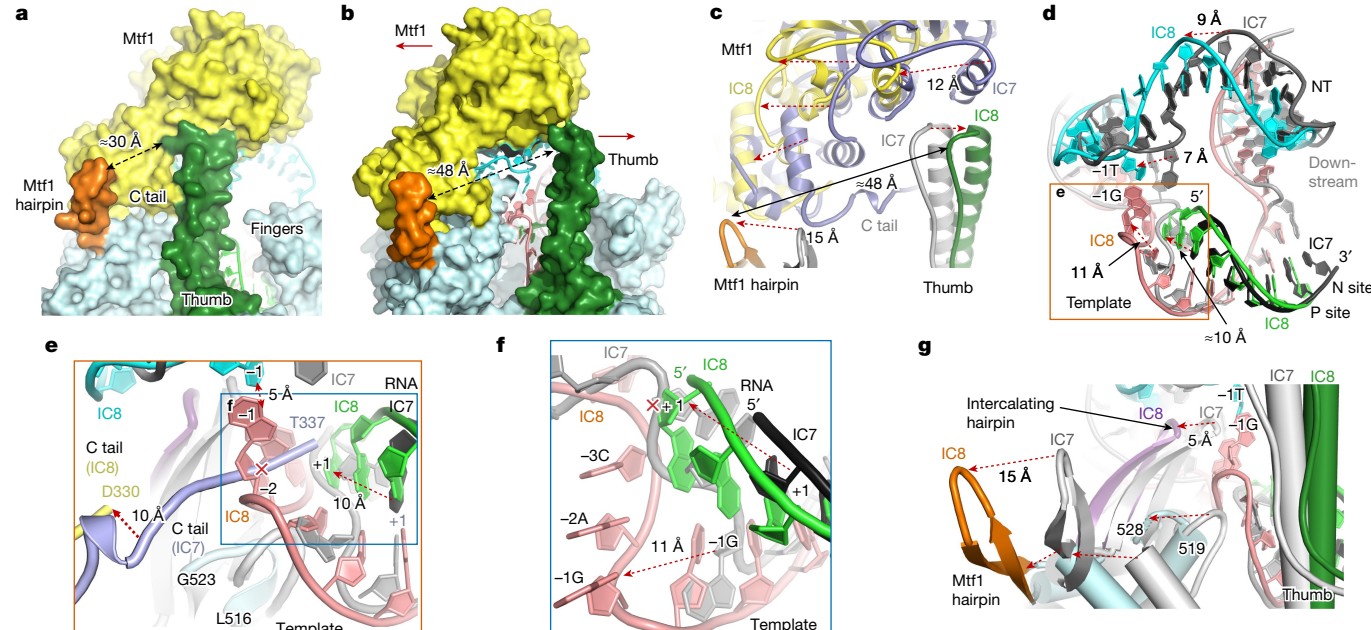

**Fig. 4 | Key structural rearrangements in IC7-to-IC8 transition. a**, The Mtf1 interacts with the y-mtRNAP thumb and Mtf1 hairpin in IC7 (yellow, Mtf1; light blue, y-mtRNAP with thumb (green) and Mtf1 hairpin (orange); and cyan, NT); these interactions are conserved from PmIC to IC7. **b**, In IC8, the Mtf1 and thumb move away from each other; the arrows indicate the shift directions (colour scheme as in **a**; and pink, template). The gap between the tip of thumb and the Mtf1 hairpin widens from about 30 Å in IC7 to about 48 Å in IC8. **c**, A zoomed view showing the movements in Mtf1, thumb and the Mtf1 hairpin from IC7 (blue, Mtf1; grey, thumb) to IC8 (yellow, Mtf1; green, thumb). **d**, The NT staircase shifts by about 9 Å towards Mtf1 from IC7 (grey) to IC8 (cyan). The 3′ end of the RNA is in the N site in IC7 (dark grey) and in the P site in IC8 (green).

The −1 template base shifts by 11 Å from IC7 (grey) to IC8 (pink), and the −1 NT base shifts by 7 Å, bringing the upstream template and NT within base-pairing distance in IC8. **e**, A zoomed view of the region outlined in **d** showing key structural changes in the template and the surrounding protein region during the IC7-to-IC8 transition. Notably, the C tail in IC7 (blue) is pushed out and disordered after D330 in IC8 (yellow), and the NT and template −1 bases are in the proximity for base-pairing in IC8. **f**, Potential clash of the RNA–DNA of IC8 with the template in IC7 (grey) explains the repositioning of the −1G nucleotide by about 11 Å in the IC7-to-IC8 transition. **g**, Positional shifts of the Mtf1 hairpin, thumb, intercalating hairpin, and y-mtRNAP 519–528 region in the IC7-to-IC8 transition.

with a string of thymine bases impairs runoff synthesis, whereas adding 6–12 successive thymines beyond the NT +12 position has little effect[35]. Notably, a nick between NT −1/+1 or +1/+2 does not reduce runoff synthesis (Fig. 2f,g); that is, the nick may not perturb the Mtf1-supported NT backbone and the base stacking (Extended Data Fig. 6b).

## Increased template scrunching in IC7 and IC8

We optimized the promoter sequence to successfully produce the structures of IC7 and IC8 states (Extended Data Fig. 1a). Initially, a G:C base pair at the +7 position prevented elongation beyond IC6, and substitution with T:A allowed for IC7 and IC8 elongation in our cryo-EM samples. The IC7 structure aligns well with that of IC6, except the RNA extension further scrunches the template and expands the NT strand stacking downstream to the +7 thymine base (Extended Data Fig. 7a,b). Template scrunching moves the −1 base by about 8 Å to a region adjacent to the thumb helix, 519–528 moiety and C tail of Mtf1 (Extended Data Fig. 7c).

The cryo-EM data of IC8 yield two closely related conformations (Extended Data Fig. 7d). IC8′ is like IC8 except for a wider opening between Mtf1 and thumb. The resolution of IC8′ is marginally lower, 3.68 versus 3.62 Å (Extended Data Table 2); thus, only the higher-resolution IC8 conformation is discussed. IC8 shows major structural rearrangements in the DNA and proteins from IC7 that are probably linked with IC-to-EC transition at 8-nt RNA synthesis[10]. Interaction of the thumb with Mtf1 stabilizes the IC states up to IC7, and notably this interaction is lost in IC8 (Fig. 4a,b). The expanded NT staircase pushes the Mtf1 by about 12 Å in one direction, and the scrunched template pushes the thumb by about 7 Å in the opposite direction (Fig. 4c), triggering the dissociation of the Mtf1–thumb interactions. The RNA 3′ end occupies

the P site in IC8 and the N site in IC7; consequently, the 5′ RNA end of the RNA–DNA extends by 2-nt length (Fig. 4d), about 10 Å, and pushes the template and the C tail of Mtf1 to new locations in IC8. The 5′ end of the RNA in IC8 occupies the space of the C tail of Mtf1 in IC7, and the C tail is pushed out of the polymerase cleft (Fig. 4e and Extended Data Fig. 5); the C tail is disordered after D330 in IC8. The 8-nt RNA–DNA in IC8 also pushes the single-stranded template to a highly scrunched state (Fig. 4f). The severely scrunched template in IC8 repositions the y-mtRNAP N-terminal 519–528 region, located between the thumb and the intercalating hairpin, by about 8 Å (Fig. 4g), triggering a series of structural rearrangements that include shifts of about 15 Å and about 5 Å of the Mtf1 hairpin and intercalating hairpin from their respective positions in IC7. Consequently, the gap between the tips of the Mtf1 hairpin and thumb is extended by about 18 Å in IC8 compared to IC7 (Fig. 4a–c). The Mtf1 hairpin interacts with the C-terminal domain of Mtf1, and the intercalating hairpin stabilizes the initiation bubble. The thumb, Mtf1 hairpin and intercalating hairpin must dissociate from Mtf1 for upstream promoter release.

## IC8 is poised for bubble collapse

As increasing RNA–DNA length scrunches the template; the bulging −1 template nucleotide shifts to new locations in the polymerase cleft (Extended Data Fig. 7e–g). The −1 base stacks with the RNA–DNA duplex in IC2 and IC3, presumably to stabilize the +1 base pair for initial RNA synthesis. In later ICs, the −1 base unstacks and shifts towards the thumb. Between IC2 and IC6, the template −1 nucleotide shifts about 21 Å along the direction of the extending RNA–DNA (Extended Data Fig. 7f), and between IC6 and IC8, the −1 base shifts about 19 Å in a direction about 95° to the initial shift (Extended Data Fig. 7g). In IC8,

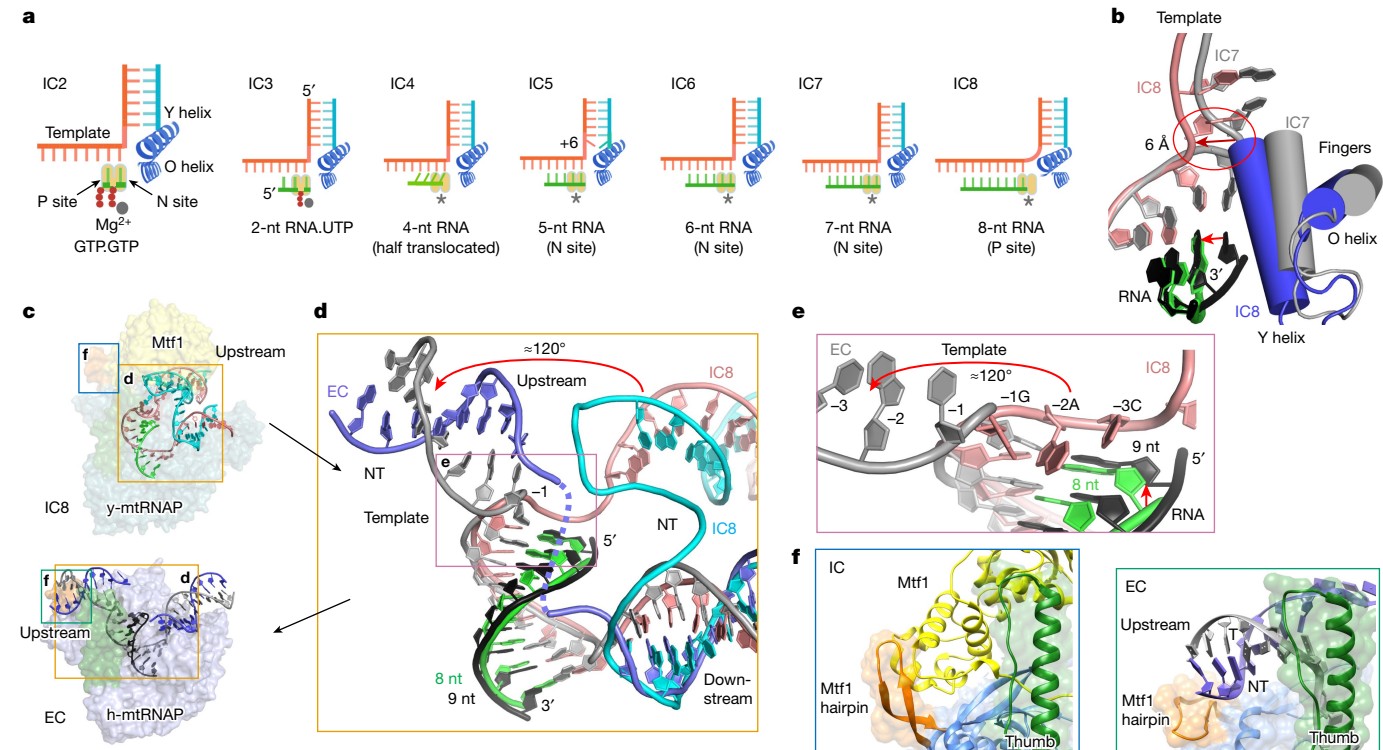

**Fig. 5 | Structural changes in the active site and the transition from IC8 to EC. a**, A schematic of the growing RNA–DNA (or NTP) from IC2 to IC8 and the translocation state of the 3′ end of the RNA (green, RNA; cyan, NT; orange, template; asterisk, active site). Both IC2 and IC3 have their respective incoming NTPs bound at the N site prior to incorporation; IC4 has 4-nt RNA in a half-translocated state between the N and P sites; IC5 has RNA 3′ end at the N site in a pre-translocated state, but the +6 base pair is melted downstream; the 3′-end of the 6- and 7-nt RNAs in IC6 and IC7 are in the N site; IC8 has 8-nt RNA 3′ end positioned at the P site, but the +9 template base has not moved to the N site. RNA.UTP, a 2-mer RNA and a UTP. **b**, The kinked template between the RNA–DNA and downstream DNA in IC7 (grey) flattens out in IC8 (pink) accompanied with the movement of the Y and O helices in IC8 (blue). **c**, The structures of the

y-mtRNAP IC8 (top) and the h-mtRNAP EC (4BOC; bottom)[24] showing the different tracks of the upstream DNA. **d**, The areas in the orange boxes in **c** are aligned on the basis of RNAP Cα superposition. The RNA–DNA and downstream DNA of IC8 and EC superimpose (IC8: green, RNA; pink, template, and cyan, NT; h-mtRNAP EC: dark grey, RNA; grey template; and blue, NT), whereas the upstream DNAs have different orientations indicating a flip by about 120° for the IC8-to-EC transition. The scrunched −1 template position is the pivotal point for upstream DNA flipping. **e**, A zoomed view of the region outlined in **d** showing flipping of the template strand in IC8 (pink) to EC (grey). **f**, Zoomed views of the Mtf1 hairpin region in IC8 (left) and EC (right), marked in **c**, showing the Mtf1 hairpin supporting the C-terminal domain of Mtf1 in the IC (left) and the upstream DNA in the EC (right).

template scrunching moves the −1 base by 11 Å, stacking it with the −2 and −3 template bases (Fig. 4f) to point them towards the complementary NT bases. The −1 template and NT bases are within a distance of about 5 Å from each other (Fig. 4e) and poised to initiate the bubble collapse process. The −1 base-pair reannealing probably triggers the bubble collapse process, facilitating the release of −2 to −4 NT bases from Mtf1. The mismatched pre-melted bubble promoter in our study prevents annealing of the −1 to −4 NT:template bases and thus enabled us to capture the IC8 transient state before the bubble collapse. We propose that the observed positional changes for Mtf1 and promoter release would occur concomitantly with bubble collapse to facilitate the IC-to-EC transition.

## Active site adapts to IC transitions

During each nucleotide addition process, the active site of RNAP cycles over multiple conformational states allowing NTP binding, catalytic incorporation and translocation involving the active participation of the O and Y helices in single-subunit RNAPs[1,2,21] or the bridge helix in bacterial RNAPs and PolII (refs. 6,36). The IC states of y-mtRNAP are captured in various conformational states of the active site (Fig. 5a). A template kink between the RNA–DNA and downstream DNA duplexes exists in all T7 RNAP, h-mtRNAP and y-mtRNAP IC structures. Notably,

this region is stretched in IC8 (Fig. 5b) and then returns to the characteristic template kink in the h-mtRNAP EC structure. The RNA 3′ end in IC8 is positioned at the P site, but the +9 template base has not moved to the N site. The unusually stretched template in IC8 apparently captures a transient state of translocation because the pre-melted bubble used in the structure fails to anneal upstream DNA and consequently stalled IC8 before the transition of the next template nucleotide to the N site.

## Modelling the transition from IC8 to EC

Previous studies of single-subunit RNAPs compared structures of the EC with early IC states to deduce conformational changes accompanying the IC-to-EC transition, such as the reorganization of the upstream DNA to a new location for the formation of an RNA-exit channel[10,24,29,37]. In our array of intermediate structures to the point of the IC-to-EC transition at IC8, we visualize the specific conformational changes required for bubble collapse and Mtf1 release. The IC8 structure (Fig. 4) shows how the expanded NT pushes Mtf1 and highly scrunched template: pushes the thumb away from Mtf1; repositions the Mtf1 hairpin and intercalating hairpin; removes the C tail of Mtf1 from the polymerase cleft; and repositions the template −1 to −3 bases adjacent to the complementing NT strand for bubble collapse. To understand the final steps of the IC8-to-EC transition, we aligned the RNAPs of the y-mtRNAP IC8

to the h-mtRNAP EC[24] (Fig. 5c,d and Extended Data Fig. 7h). The RNAP superposition shows that the downstream DNA and RNA–DNA tracks in IC8 and EC structures align, indicating that these elements remain relatively unaffected in the transition. However, the upstream DNA undergoes a large swivel motion of about 120° from the severely bent structure in IC8 to a relaxed conformation in the EC[2]. The potential clash between the template in IC8 and the 5′ end of the RNA in the EC probably contributes to the template reorganization (Fig. 5e). Notably, the Mtf1 hairpin (or lever loop/B2 hairpin) of y-mtRNAP (or h-mtRNAP), supporting the C-terminal domain of Mtf1 (or TFB2M) in the IC states, is engaged with the upstream DNA in the EC state (Fig. 5f).

## Transcription initiation in other systems

The general principles of promoter-specific transcription initiation and transition to elongation are conserved in single- and multi-subunit RNAPs; however, the detailed mechanisms are system specific. Single-subunit T7 RNAP and mtRNAPs form a smaller initiation bubble spanning from −4 to +2 in contrast to a larger bubble in the multi-subunit RNAPs of bacteria (−11 to +2) and eukaryotes. The RNAP and transcription initiation factor(s) remain bound to the promoter throughout initiation and dissociate only following transition to the EC. The y-mtRNAP IC structures in this study show how DNA scrunching driven by the growing RNA–DNA reorganizes the template strand, NT strand and the flexible C tail of Mtf1 to support RNA synthesis during initial transcription and later promoter escape for the EC transition (Supplementary Video 1). The transitions associated with the discrete structural states can be further explored by time-resolved and molecular dynamics simulation studies. In y-mtRNAP, NT scrunching contributes to abortive synthesis in IC2 and IC3 states, and the expanded NT strand and tightly scrunched template in IC8 trigger structural changes to enter the EC. Evidently, transcription initiation involves DNA scrunching[7–10]. DNA scrunching has been observed during abortive synthesis in the bacterial RNAP system by single-molecule studies[30]; however, the structural snapshots have not been captured. Hence, it remains to be determined whether the specific IC states observed in y-mtRNAP have functional and/or conformational analogues in other RNAP systems.

The RNAPs from phage T7, bacteria and PolII are the best-characterized systems thus far (Extended Data Fig. 8). Existing structures show similarities in the promoter escape mechanism involving clashes of RNA–DNA with transcription factors and/or structural elements of RNAP. In T7 RNAP, the growing RNA–DNA clashes with the N-terminal domain, which interacts with the promoter, causing the domain to rotate and ultimately dissociate from the promoter[8,37,38]. For PolII, TFIIB and its B-reader element involved in promoter recognition are displaced when 12- to 13-nt RNA is synthesized[39]. In bacterial RNAP ICs, clashes of the σ-finger with RNA–DNA[31] and template scrunching trigger the EC transition[14]. As many key cellular processes evolve from simpler single-cell species to multicellular systems with added complexity, we expect a similar evolution process for transcription initiation; however, this has not been explored in detail.

The transcription initiation mechanism of y-mtRNAP is most similar to that of h-mtRNAP. A major difference is that h-mtRNAP requires TFAM to position and engage the h-mtRNAP–TFB2M at the promoter start site[19,40], whereas y-mtRNAP–Mtf1 catalyses promoter-specific initiation without needing Abf2, the TFAM homologue. This is presumably because the y-mt promoter consensus sequence is specifically recognized by Mtf1. The Y103-X-W105 region of Mtf1, essential for promoter melting in yeast[1], is not conserved in TFB2M, in which the corresponding region has a large loop structure (residue L154 to S168)[2]. This difference suggests that the homologous factors in y-mtRNAP and h-mtRNAP ICs may use distinct promoter melting mechanisms. Although y- and h-mtRNAPs are expected to follow a similar initiation path, the involvement of TFAM versus y-ins might create noticeable structural differences between the two in initial RNA synthesis and transition into

elongation. The y-ins domain essential for transcription initiation by y-mtRNAP is absent in h-mtRNAP, which recruits TFAM for initiation. Some of the tasks of y-ins are delegated to TFAM as both stabilize the promoter. It also remains to be seen whether the NT scrunching and progressive rearrangements of the transcription bubble in y-mtRNAP are followed in h-mtRNAP. The findings of the current study can guide in designing experiments to obtain the structures of h-mtRNAP IC intermediates. Such insights will enhance our understanding of the mechanisms involved in mitochondrial DNA transcription initiation in the human system.

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

## Methods

### Expression and purification of WT and mutant Mtf1

The pTrcHisC plasmid encoding His6-tagged *S. cerevisiae* Mtf1 was transformed into *Escherichia coli* BL21 competent cells and expressed and purified as described previously[22,41]. Cells were grown at 37 °C in LB medium supplemented with 100 µg ml⁻¹ ampicillin to an optical density of 0.8 at 600 nm before induction with 1 mM isopropyl β-D-1-thiogalactopyranoside. After 16 h at 16 °C, cells were collected and the cell pellet was lysed by sonication in the presence of protease inhibitor and lysozyme in lysis buffer (50 mM sodium phosphate pH 8, 300 mM NaCl, 10% glycerol, 0.1 mM phenylmethylsulfonyl fluoride (PMSF)), and subsequently subjected to polyethylenimine (10%) and ammonium sulfate (55%) precipitation. The dissolved ammonium sulfate pellet was loaded on a 5-ml DEAE Sepharose and 5 ml Ni-Sepharose cartridge (GE Healthcare Life Sciences) connected in tandem. After sample loading, the DEAE Sepharose cartridge was detached and the Ni-Sepharose cartridge was washed with 50 ml wash buffer (50 mM sodium phosphate buffer pH 8, 300 mM NaCl, 10% glycerol, 1 mM PMSF, 20 mM imidazole). The Mtf1 protein was eluted with a 70-ml gradient of 20 mM to 500 mM imidazole in wash buffer. The Mtf1 peak eluent was collected and loaded on two 1-ml heparin-Sepharose columns (GE Healthcare Life Sciences) connected in tandem. Subsequently, the columns were washed with 20 ml heparin wash buffer (50 mM sodium phosphate buffer pH 8, 150 mM NaCl, 10% glycerol, 1 mM EDTA, 1 mM dithiothreitol (DTT), 1 mM PMSF) before elution with a 50-ml gradient of 150 mM to 1 M NaCl in heparin wash buffer. The eluted Mtf1 protein was collected and concentrated using a 10-kDa-cutoff Amicon ultra-centrifugal filter (Merck Millipore) and stored at −80 °C. The Mtf1 mutant proteins were purified using the same protocol (Supplementary Fig. 1).

### Expression and purification of ΔN100 and Δy-ins (1237–1321 deleted) y-mtRNAPs

The ProEXHTB plasmid encoding His6-tagged *S. cerevisiae* ΔN100 y-mtRNAP was transformed into *E. coli* BL21 RIL Codon Plus competent cells and expressed and purified as described previously[22,23,41]. Cell growth conditions, lysis, polyethylenimine and ammonium sulphate precipitation, tandem DEAE and Ni column, and heparin column chromatography steps were identical to those for Mtf1 described above. The ΔN100 y-mtRNAP protein eluent from the heparin column was treated with TEV protease at 100:1 (w:w) ΔN100 y-mtRNAP to TEV protease for 16 h at 4 °C. The cleaved protein was loaded on a 5 ml Ni-Sepharose cartridge and the flow-through was collected and concentrated using a 10-kDa-cutoff Amicon ultra-centrifugal filter (Merck Millipore) and stored at −80 °C. The y-ins-deleted y-mtRNAP was purified using the same protocol (Supplementary Fig. 1).

### In vitro transcription initiation assay to measure runoff and abortive RNA synthesis

In vitro transcription initiation assays were carried out as discussed previously[22]. Transcription reactions with the pre-melted initiation bubble promoter (2 µM) were carried out using y-mtRNAP (1 µM) and Mtf1 (2 µM) in reaction buffer (50 mM Bis-tris propane pH 7.0, 100 mM NaCl, 5 mM MgCl₂, 1 mM EDTA, 2 mM DTT) and the indicated combinations of 100 µM of ATP, GTP and UTP, and 200 µM of 3′-dCTP spiked with [α-³²P]GTP. The RNA synthesis reaction was conducted at 4 °C for 2 h or at 25 °C for 15 min and terminated using 0.5 M EDTA and formamide dye mixture (98% formamide, 0.025% bromophenol blue, 10 mM EDTA). RNA products were resolved on a 24% polyacrylamide, 4 M urea gel. Visualization, runoff RNA product quantification and analysis of the gel were accomplished with ImageQuant software. The quantified runoff products were graphed using GraphPad Prism 9 software.

Transcription reactions on duplex promoters (2 µM) were carried out with y-mtRNAP or mutant (1 µM) and Mtf1 or mutants (2 µM) and 100 µM of ATP, GTP and UTP, and 200 µM of 3′ dCTP spiked with [γ-³²P]ATP in reaction buffer (50 mM Tris-acetate pH 7.5, 100 mM potassium glutamate, 10 mM magnesium acetate, 1 mM DTT, 0.01% Tween 20) at 25 °C for 15 min. Transcription reactions on nicked and nucleotide-deleted promoters were carried out similarly but were monitored using [α-³²P] GTP. Reactions were terminated using 0.5 M EDTA and formamide dye mixture (98% formamide, 0.025% bromophenol blue, 10 mM EDTA); uncropped gels are in Supplementary Fig. 1.

### Assembly and characterization of y-mtRNAP IC2, IC4, IC5, IC6, IC7 and IC8

Complex assembly and characterization were carried out as described previously[1,22]. Three y-mtRNAP PmICs were prepared by incubating ΔN100 y-mtRNAP, Mtf1 and a promoter DNA (Extended Data Fig. 1a) in a molar ratio of 1:1.2:1.2 for 2 h at 4 °C. The bubble-1 PmIC with a 33-bp bubble promoter was used for the y-mtRNAP IC2 and IC4 structures, and the bubble-2 PmIC with a 36-bp bubble promoter was used for IC5 and IC6 structures. The bubble-3 PmIC with a 37-bp bubble promoter was used to capture the IC7 and IC8 states. Each PmIC, at a starting concentration of 6 mg ml⁻¹ in buffer A (50 mM Bis-tris propane pH 7.0, 100 mM NaCl, 5 mM MgCl₂, 1 mM EDTA, 2 mM DTT), was purified by size-exclusion chromatography using a Superdex 200 Increase 10/300 GL column connected to an AKTA PURE 25 FPLC system (GE Healthcare Life Sciences) maintained at 6 °C. The column was in-line with a multi-angle light-scattering device (mini-DAWN), a differential refractive index measuring device (Optilab) and a dynamic light-scattering device (DynaPro Nanostar) from Wyatt Technology for measuring the molecular mass and hydrodynamic radius of the eluted samples from the size-exclusion column. After purification, each PmIC was concentrated to about 3 mg ml⁻¹ (about 15 µM), aliquoted and stored at −80 °C before further use. The concentrations of the PmICs were determined through NanoDrop One UV-VIS Spectrophotometer (Thermo Fisher) measurements with buffer A as the blank.

The PmIC on the 33-bp bubble promoter was incubated with GTP at a molar ratio of 1:50 to generate IC2, and with pppGpG (TriLink Bio-Technologies), UTP and ATP at a molar ratio of 1:3:25:25 to prepare IC4. The PmIC on the 36-bp bubble promoter was incubated with pppGpG and ATP at a molar ratio of 1:3:50 to generate IC5, and pppGpG, ATP, UTP and GTP at a molar ratio of 1:3:50:20:20 to generate IC6. The 37-bp bubble promoter PmIC was incubating with pppGpG, ATP and UTP at a molar ratio of 1:10:50:40 for IC7; and with GTP, ATP and UTP at a molar ratio of 1:40:40:30 for IC8. All of the ICs were incubated for 2 h at 4 °C before grid preparation.

The complexes were characterized by stand-alone cuvette mode DLS experiments using a DynaPro Nanostar (Wyatt Technology) to confirm similar hydrodynamic radius and polydispersity values for all IC samples. For the DLS experiments, 8 µl of IC samples at a concentration of 0.5 mg ml⁻¹ was loaded into disposable cuvettes (Wyatt Technologies) and placed in the sample chamber maintained at 4 °C. DLS experiments consisted of 30 acquisitions for each sample, and data were analysed using Dynamics software (Version 7.10.0.23; Wyatt Technology).

### Cryo-EM grid preparation and data collection

The workflows for IC2–IC8 structure determination are shown in Supplementary Fig. 2. Vitreous grids of y-mtRNAP ICs were prepared on Quantifoil R 1.2/1.3 holey carbon grids (Cu300 or Au300) using a Leica EM GP (Leica Microsystems). The grids were glow-discharged for 45 s at 25 mA current with the chamber pressure set at 0.3 mbar (PELCO easi-Glow; Ted Pella). Glow-discharged grids were mounted in the sample chamber of a Leica EM GP at 8 °C and 95% relative humidity. Optimized grids for IC2 and IC5 were obtained by spotting 3 µl of the sample at 0.8 mg ml⁻¹ (4 µM) and 0.9 mg ml⁻¹ (4.5 µM), respectively, on Cu300 grids. The sample (in 50 mM Bis-tris propane, pH 7.0; 100 mM NaCl, 5 mM MgCl₂, 1 mM EDTA and 2 mM DTT) was incubated on the grids for

30 s before back-blotting for 12 s using two pieces of Whatman Grade 1 filter paper, and the plunge-freezing was carried out by dipping the grids in liquid ethane at a temperature of −172 °C. Optimized grids for IC4, IC6, IC7 and IC8 were obtained by spotting 3 µl of the sample at 0.7 (3.5 µM of IC4 and IC7), 0.8 (4 µM of IC6) and 0.5 mg ml⁻¹ (2.5 µM of IC8) on Au300 grids, before back-blotting for 12 s.

The grids were clipped and mounted on a 200-keV Glacios cryo-transmission electron microscope (Thermo Fisher) with an autoloader and Falcon 3 direct electron detector as installed in our laboratory. High-resolution datasets for γ-mtRNAP IC2 and γ-mtRNAP IC4 to IC8 were collected on the Glacios using EPU software version 2.9.0 (Thermo Fisher).

Electron microscopy data were recorded as movies in counting mode at a nominal magnification of ×190.000 for IC2 yielding a pixel size of 0.76 Å, and at a nominal magnification of ×150.000 for IC4 to IC8, yielding a pixel size of 0.97 Å. The total exposure time was 25.15 s for IC2, 38.41 s for IC4 to IC6, and 37.06 s for IC7 and IC8 with a total dose of 40 electrons per square angstrom for all datasets. All videos were recorded as gain-corrected MRC files. The data collection parameters for all structures are listed in Extended Data Tables 1 and 2.

### Cryo-EM data processing

For each dataset, individual video frames were motion-corrected and aligned using MotionCor2 (ref. 42) as implemented in the Relion 3.1 package[43] and the contrast transfer function parameters were estimated by CTFFIND-4 (ref. 44). The particles were automatically picked using the reference-free Laplacian-of-Gaussian routine in Relion 3.1. The picked particles were cleaned by cycles of two-dimensional (2D) and 3D classifications. The previously obtained density map for IC3 (Electron Microscopy Data Bank ID: EMD-10846; Protein Data Bank (PDB) ID: 6YMW) was blurred to 40 Å resolution and used as the template for initial 3D classifications. The 3D classifications eliminated partially disordered and incomplete particles. The particles forming lower-resolution density maps (Supplementary Fig. 2) were eliminated by manually visualizing in Chimera; no lower-resolution map for an IC state defined a conformation distinct from the accepted one. However, two distinct classes, IC8 and IC8′, were detected in the 3D classification stage and were separated as two independent datasets for further refinement. The homogeneous particle sets from the best 3D class were used to calculate gold-standard 3D auto-refined maps and corresponding masks. For each structure, the particles were repicked at the 3D auto-refined positions and classified further within its mask. The particles were extracted with a box size of 256³ pixels for IC2 and 192³ for the remaining datasets. Final 3D classification generated a distinct class, and no additional class such as PmIC was detected in processing any of the datasets (Extended Data Tables 1 and 2). The final set of particles for each IC was used to calculate gold-standard auto-refined maps, further improved by Bayesian polishing and contrast transfer function refinement. All data processing steps were carried out using Relion 3.1 (Supplementary Fig. 2). The local-resolution maps were calculated using ResMap and the orientation plots were generated by Relion 3.1. Supplementary Fig. 3 shows the 3D-FSC plots for all IC structures calculated on the 3D-FSC server (https://3dfsc.salk.edu/).

### Model building

Data processing yielded 3.47-Å, 3.44-Å, 3.39-Å, 3.62-Å, 3.75-Å and 3.62-Å density maps for IC2, IC4, IC5, IC6, IC7 and IC8, respectively, and these were used to fit the atomic models for the respective structures. Previously published PmIC (PDB ID: 6YMV) and IC3 (PDB ID: 6YMW) structures were used as references for building the models of the γ-mtRNAP IC2 and IC4-IC8 structures. Furthermore, the AlphaFold prediction of γ-mtRNAP (UniProt ID: P13433) was used for modelling the insertion region (γ-ins, residues 1232 to 1328). In addition, the γ-ins region of previously published PmIC and IC3 structures have been corrected on the basis of the AlphaFold model. All model building were carried out manually using COOT[45]. Model building was coupled with iterative rounds of real-space structure refinement using Phenix 1.19.2 (ref. 46). In the IC2 final structure, Mtf1 is traced from S2 to S340 and γ-mtRNAP is traced from I386 to the end residue S1351 (with the stretches of residues 442–447, 559–588 and 1309–1320 missing). In the IC4 structure, Mtf1 is traced from S2 to G341 (end residue) and γ-mtRNAP is traced from A407 to S1351 (with the stretches of residues 559–588 and 1311–1319 missing). In the IC5 structure, Mtf1 is traced from S2 to Y335 and γ-mtRNAP is traced from I386 to S1351 (with the stretches of residues 556–588 and 1310–1319 missing). In the IC6 structure, Mtf1 is traced from S2 to T337 and γ-mtRNAP is traced from I386 to S1351 (with the stretches of residues 554–588 and 1311–1319 missing). In the final IC7 structure, Mtf1 is resolved from residues S2 to T337 and γ-mtRNAP is traced from I386 to S1351 (with the stretches of residues 554–588 and 1312–1318 missing). In the IC8 structure, Mtf1 is traced from S2 to D330 and γ-mtRNAP is traced from I386 to S1351 (with the stretches of residues 524–526, 554–588 and 1311–1319 missing). The DNA and RNA chains are different for each structure and were modelled independently into the experimental density maps. In IC2, the base of template nucleotide 22 has been removed owing to weak density. In IC4 the bases of NT nucleotide 123 and template nucleotides 22 and 24 were removed, and template nucleotide 23 was omitted. All structure figures were generated using PyMol (https://pymol.org/2/), Chimera[47] and ChimeraX[48]. Videos were generated using Chimera.

### Reporting summary

Further information on research design is available in the Nature Portfolio Reporting Summary linked to this article.

### Data availability

The coordinates and cryo-EM density maps for IC2, IC4, IC5, IC6, IC7, IC8 and IC8′ have been deposited under PDB and Electron Microscopy Data Bank accession codes 8AP1 and EMD-15556; 8ATT and EMD-15662; 8ATV and EMD-15664; 8ATW and EMD-15665; 8C5S and EMD-16442; 8C5U and EMD-16443; and 8Q63 and EMD-18183, respectively. For model building, the following models were used: 6YMV and 6YMW. The AlphaFold prediction of γ-mtRNAP (UniProt ID: P13433) was used for modelling the insertion region. Uncropped gels are available in Supplementary Fig. 1. Source data are provided with this paper.

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

**Acknowledgements** We thank A. K. Singh for the helpful discussion and N. R. Shewakramani for help making the videos. The study was supported by Rega Virology and Chemotherapy internal grants to K.D. and National Institutes of Health grant GM118086 to S.S.P. Q.G. acknowledges financial support from the Research Foundation – Flanders (FWO-Vlaanderen) through doctoral fellowship grant number 1162823N.

**Author contributions** Conceptualization: S.S.P., K.D. Methodology: J.S., protein purification and assay development; B.D.W., protein purification, grid optimization and data collection; U.B. protein purification and expression construct preparation. Investigation: Q.G., model building and structure analysis; J.S., biochemical assays; S.S.P., model building and structure analysis; K.D., cryo-EM data processing and model building, and structure analysis. Visualization (data interpretation and graphical representations): Q.G., S.S.P., K.D. Funding acquisition: S.S.P., K.D. Project administration: S.S.P., K.D. Supervision: S.S.P., K.D. Writing (original draft): S.S.P., K.D. Writing (review and editing): all authors.

**Competing interests** The authors declare no competing interests.

**Additional information**

**Correspondence and requests for materials** should be addressed to Smita S. Patel or Kalyan Das.

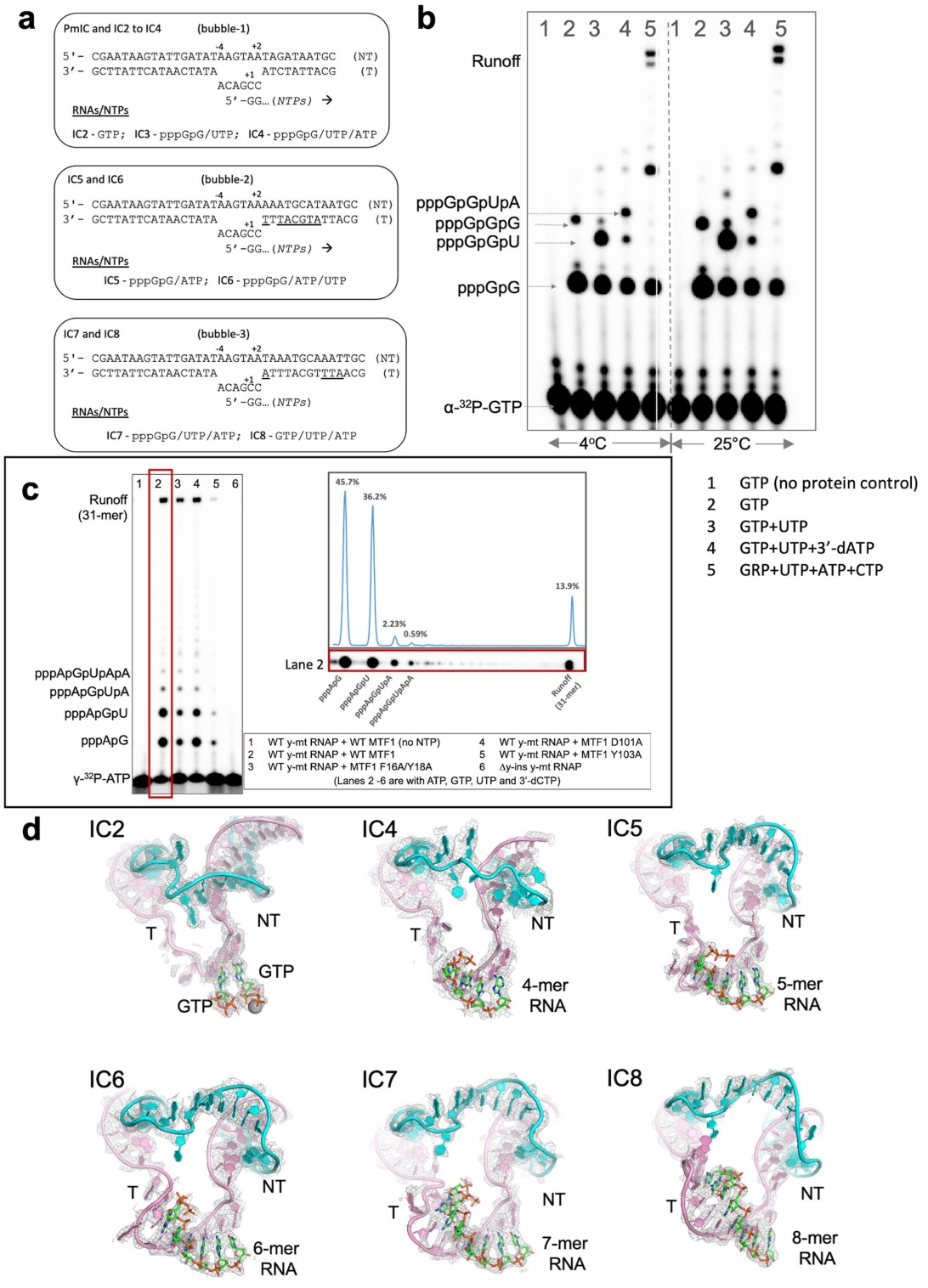

**a**

PmIC and IC2 to IC4        (bubble-1)
```
                    -4      +2
5'- CGAATAAGTATTGATATAAGTAATAGATAATGC (NT)
3'- GCTTATTCATAACTATA   +1 ATCTATTACG (T)
                 ACAGCC
                    5'-GG...(NTPs) →
```
RNAs/NTPs

IC2 - GTP;   IC3 - pppGpG/UTP;   IC4 - pppGpG/UTP/ATP

IC5 and IC6        (bubble-2)
```
                    -4      +2
5'- CGAATAAGTATTGATATAAGTAAAAATGCATAATGC (NT)
3'- GCTTATTCATAACTATA   +1 TTTACGTATTACG (T)
                 ACAGCC
                    5'-GG...(NTPs) →
```
RNAs/NTPs

IC5 - pppGpG/ATP;   IC6 - pppGpG/ATP/UTP

IC7 and IC8        (bubble-3)
```
                    -4      +2
5'- CGAATAAGTATTGATATAAGTAATAAATGCAAATTGC (NT)
3'- GCTTATTCATAACTATA   ATTTACGTTTAAACG (T)
                 ACAGCC  +1
                    5'-GG...(NTPs)
```
RNAs/NTPs

IC7 - pppGpG/UTP/ATP;   IC8 - GTP/UTP/ATP

**b**

Runoff

pppGpGpUpA
pppGpGpG
pppGpGpU

pppGpG

α-$^{32}$P-GTP

←—— 4°C ——→  ←—— 25°C ——→

1  GTP (no protein control)
2  GTP
3  GTP+UTP
4  GTP+UTP+3'-dATP
5  GRP+UTP+ATP+CTP

**c**

Runoff
(31-mer)

pppApGpUpApA
pppApGpUpA
pppApGpU
pppApG

γ-$^{32}$P-ATP

45.7%
36.2%
2.23%  0.59%
13.9%

Lane 2

pppApG  pppApGpU  pppApGpUpA  pppApGpUpApA   Runoff (31-mer)

| 1 | WT y-mt RNAP + WT MTF1 (no NTP) | 4 | WT y-mt RNAP + MTF1 D101A |
| 2 | WT y-mt RNAP + WT MTF1 | 5 | WT y-mt RNAP + MTF1 Y103A |
| 3 | WT y-mt RNAP + MTF1 F16A/Y18A | 6 | Δy-ins y-mt RNAP |

(Lanes 2-6 are with ATP, GTP, UTP and 3'-dCTP)

**d**

IC2    GTP  GTP

IC4    4-mer RNA   T   NT

IC5    5-mer RNA   T   NT

IC6    6-mer RNA   T   NT

IC7    7-mer RNA   T   NT

IC8    8-mer RNA   T   NT

**Extended Data Fig. 1** | See next page for caption.

**Extended Data Fig. 1 | The DNA scaffolds for structural studies of PmIC to IC8 complexes. a**. Three bubble promoter DNA scaffolds were used to synthesize RNAs for trapping the IC2 to IC8 complexes in situ; the nucleotides and a 2-mer RNA used for obtaining each IC are indicated. The inserted or modified sequences in bubble−2 and bubble-3 compared to the preceding bubbles are underlined. **b**. The gel image shows the transcription reaction on the IC2 to IC4 bubble under the cryo-EM experimental condition (4 °C, left), and at room temperature (25 °C; right) monitored using α-$^{32}$P-GTP; the buffer conditions are as for the cryo-EM sample. The raw gel image is in Supplementary Fig. 1d. **c**. The transcription assay on a duplex 21S γ-mtRNAP promoter. Transcription reactions were carried out for 15 min in 50 mM Tris acetate pH 7.5, 100 mM potassium glutamate, 10 mM magnesium acetate, 1 mM DTT, 0.01% Tween 20 buffer at 25 °C. The reactions contained 100 µM of each ATP, UTP, and GTP, 200 µM of 3′-dCTP spiked with γ-$^{32}$P-ATP. Lane-2 in the gel image is quantified to determine the relative percentages of 2-, 3-, 4-mer, and runoff RNAs (source data provided). Representative of three biological repeats is shown in Supplementary Fig. 1d. Note that the pre-melted bubble and duplex promoters show a different pattern of abortives attributed to the pre-melted initiation region and the inability of the bubble promoter to reanneal for the transition into EC. The raw gel images are in Supplementary Fig. 1b and source data provided. **d**. Experimental density maps define the transcription bubbles in IC2, IC4, IC5, IC6, IC7, and IC8 (NT, cyan; T, pink; GTP and RNA, green). The contour levels of the density maps are 1.75, 1.3, 2, 2, 2, and 3σ for IC2, IC4, IC5, IC6, IC7, and IC8, respectively. The density map for IC4 is unsharpened and the remaining maps are B-sharpened.

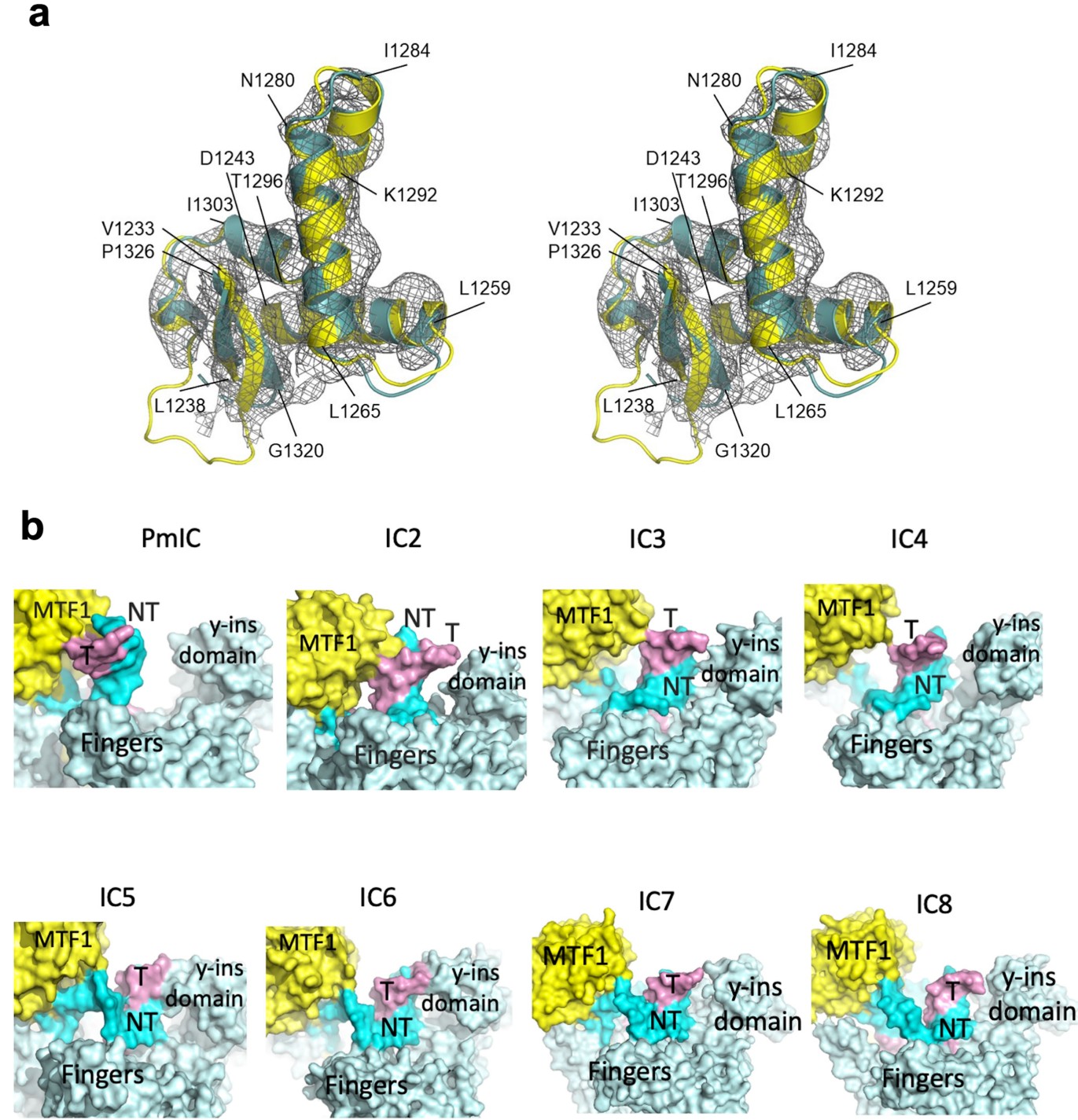

**Extended Data Fig. 2 | The y-ins region and its interactions with downstream DNA in IC states. a**. Wall-eyed stereo view showing the starting y-ins AlphaFold model (UniProt ID: P13433, yellow) that was fitted to the density (turquoise model); the density for y-ins domain in the IC6 structure is contoured at 2.5σ. **b**. Relative positioning and interactions of y-ins domain with the downstream DNA in PmIC and all IC structures; yellow MTF1, light blue y-mtRNAP, cyan non-template, pink template DNA. The y-ins and MTF1 N-terminal domains define the downstream cleft, which interacts with the downstream DNA duplex. The interactions between the DNA and the downstream cleft are rearranged in IC structures. The DNA interacts only with MTF1 in PmIC, moves closer to y-ins in IC2, and interacts extensively with y-ins in IC5 to IC8, where the NT staircase structure is walled against MTF1. The structures are oriented as Fig. 1a (left panel).

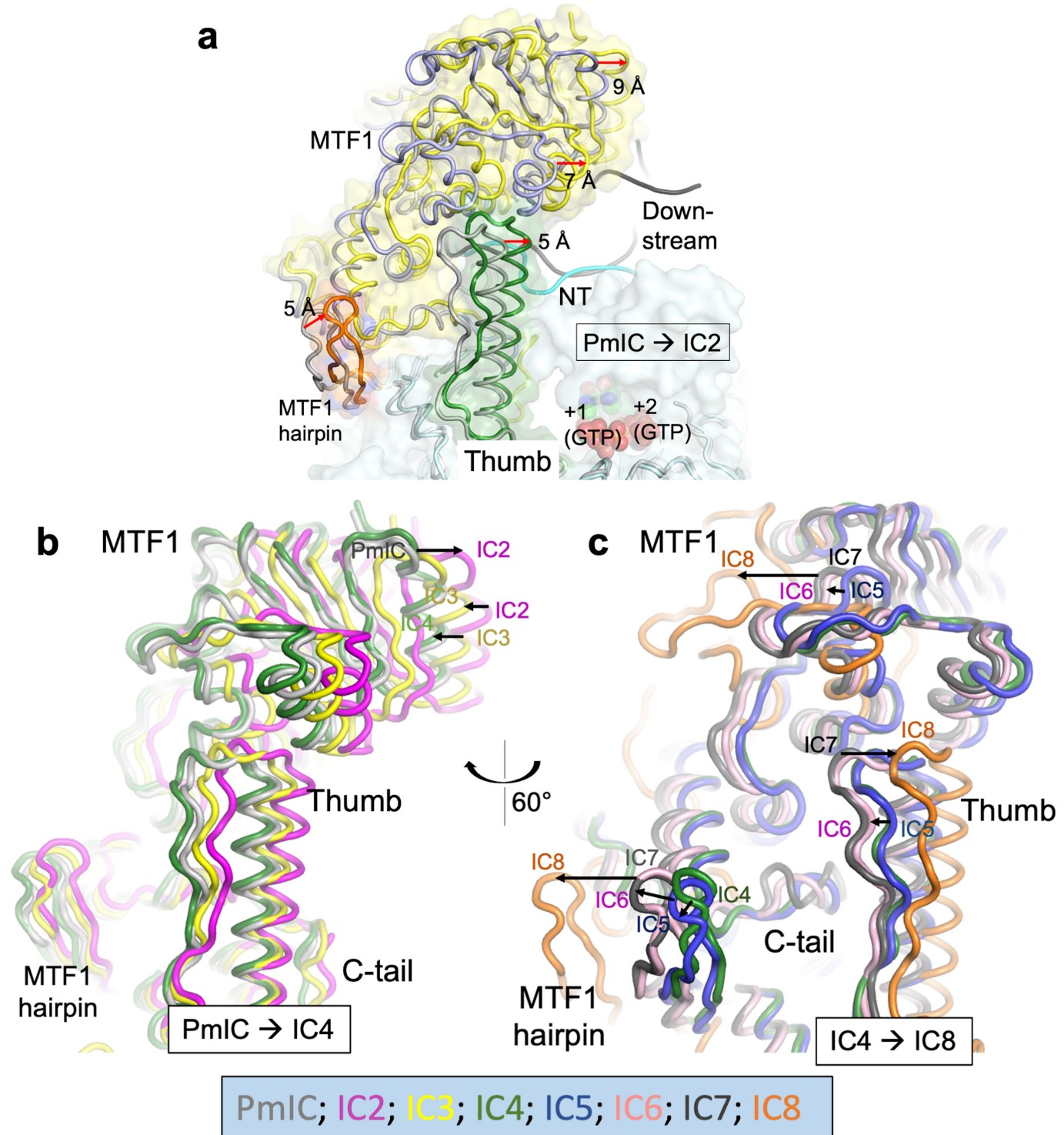

**Extended Data Fig. 3 | Relative positioning of MTF1 and thumb in different IC states. a.** Superposition of IC2 (yellow MTF1, orange MTF1 hairpin, green thumb, cyan NT, and GTP colored by heteroatom in space-filling representation) on PmIC (blue MTF1 and gray mtRNAP) show correlated motions of MTF1 and thumb subdomain/MTF1 hairpin (red arrows) to secure the downstream cleft upon binding of the initiating GTP in IC2. Positioning of MTF1 and y-mtRNAP (thumb and MTF1-hairpin) in different IC states; from PmIC to IC4 (**b**) and IC4 to IC8 (**c**). The structures are aligned based on y-mtRNAP Cα superposition. The structures are colored - PmIC (gray), IC2 (magenta), IC3 (yellow), IC4 (green), IC5 (blue), IC6 (pink), IC7 (dark gray), and IC8 (orange).

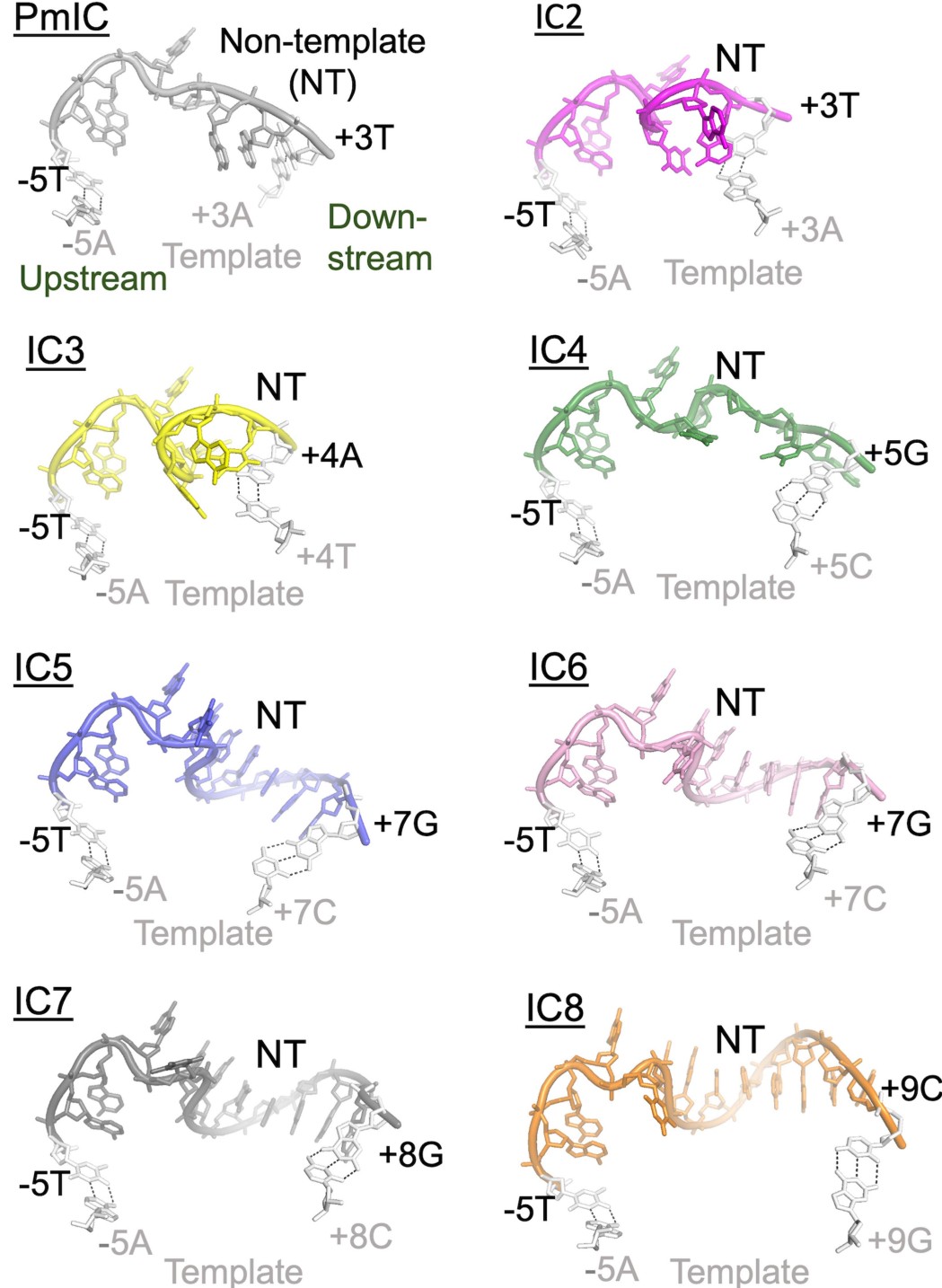

**Extended Data Fig. 4 | The conformations of the NT single-stranded region in PmIC to IC8.** The NT strands in transcription bubbles for the IC states are shown in color: PmIC, gray; IC2, magenta; IC3, yellow; IC4, green; IC5, blue; IC6, pink; IC7, dark gray; IC8, orange. The NT:template base pairs at the upstream and downstream ends of each NT strand in the transcription bubble are in light gray. The NT strands are visibly scrunched in IC2 and IC3, and forming a stacked spiral staircase structure in the complexes from IC5 to IC8.

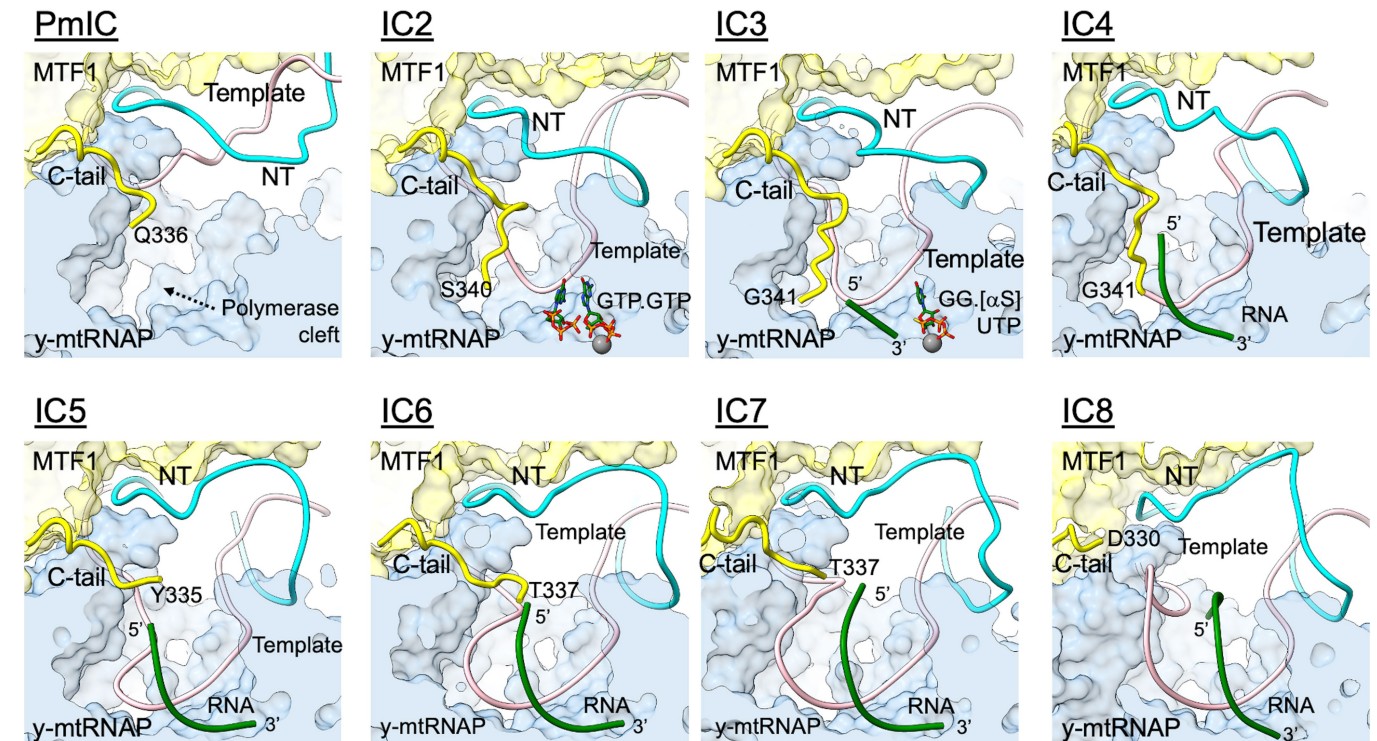

**Extended Data Fig. 5 | Structural rearrangements of RNA:DNA, template, NT, and C-tail with each nucleotide addition from PmIC to IC8.** The NT (cyan), template (pink), RNA (green), and MTF1 C-tail (yellow) in the bubble region are shown. The C-tail relocates as RNA:DNA grows with each nucleotide addition.

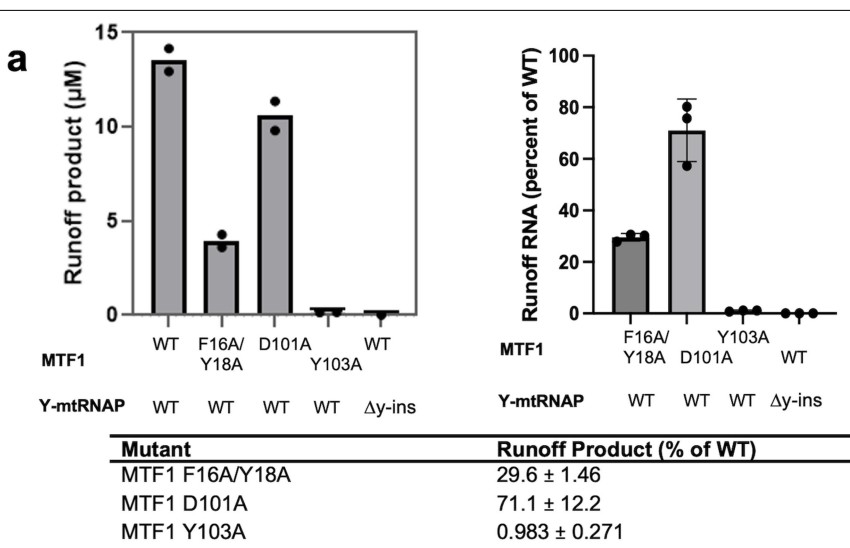

| Mutant | Runoff Product (% of WT) |
|---|---|
| MTF1 F16A/Y18A | 29.6 ± 1.46 |
| MTF1 D101A | 71.1 ± 12.2 |
| MTF1 Y103A | 0.983 ± 0.271 |
| Δy-ins y-mtRNAP | 0.00889 ± 0.00476 |

MTF1, yellow; NT, cyan; template, pink

**Extended Data Fig. 6 | Effects of MTF1 and y-mtRNAP mutations on transcription and their structural impacts. a.** The bar chart and table show the amount (in µM) of runoff RNA and percentages by the mutants compared to that by WT proteins; the gel is in Extended Data Fig. 1c and the raw gel images in Supplementary Fig. 1. The error bars (mean±SD) are from three biological repeats (source data). **b.** The sugar-phosphate backbone of the stacked NT bases is walled against MTF1 N-terminal domain. **c.** -180° rotation view to panel **b** shows the interaction of MTF1 F16/Y18 with +3 to +5 NT backbone and D101 with +1 NT base in IC5. **d.** Interactions of the template strand +3 and +4 nucleotide backbone with F16/Y18 groove of MTF1 in PmIC. **e.** Interactions of the template strand +8 and +9 backbone with F16/Y18 groove of MTF1 in IC2.

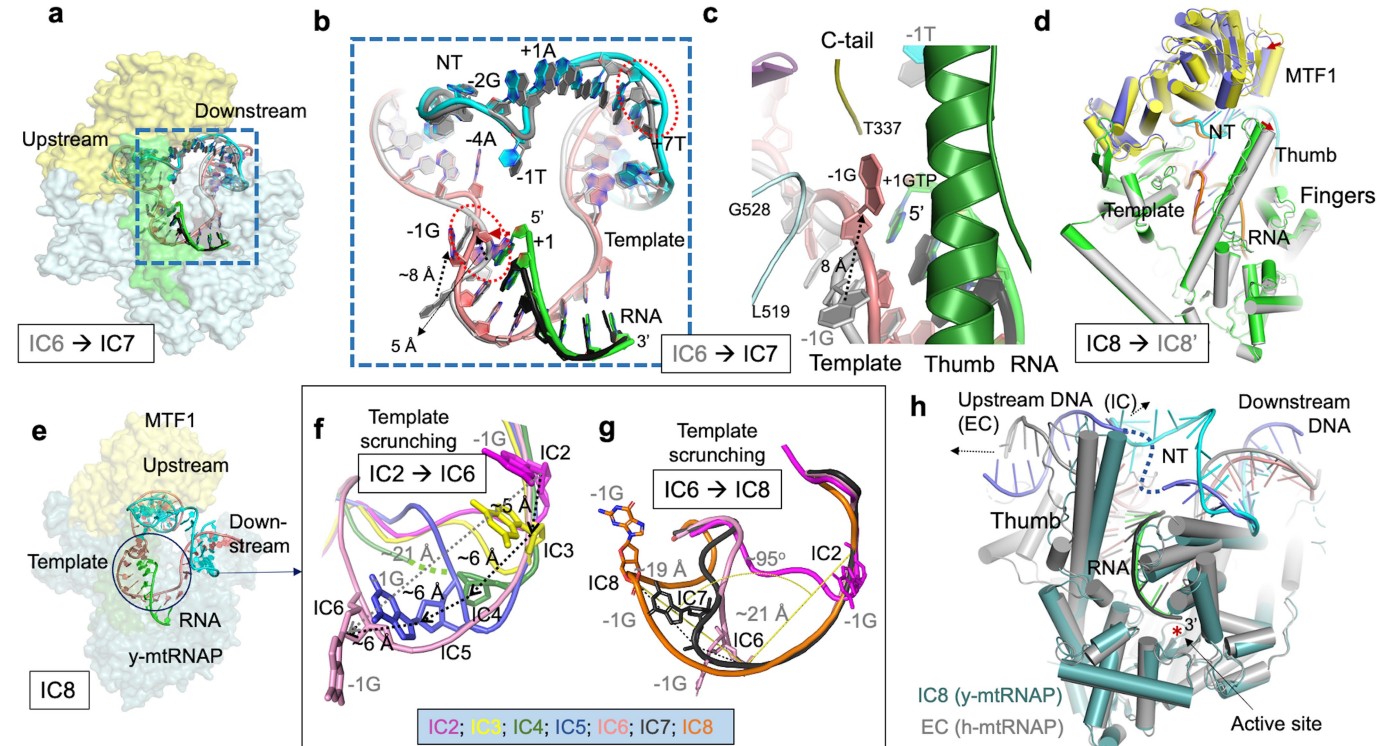

**Extended Data Fig. 7 | Structural changes from IC6 to IC8 states and between IC8 and EC states. a.** Superposition of IC6 and IC7; 809 Cα atoms are aligned with an rmsd of 0.63 Å. MTF1 and y-mtRNAP of IC7 are in yellow and light blue surface representations, respectively, and the transcription bubble in IC6 is gray and colored in IC7 (pink template; cyan non-template; green RNA). **b.** A zoomed view of the transcription bubbles in IC6 and IC7 shows small changes. The RNA extends by one nucleotide towards the 5′-end in IC7 and the template and non-template (NT) bulge, as highlighted by red circles. **c.** A zoomed view showing the template −1G nucleotide is shifted by ~8 Å towards Thumb in the IC6→IC7 transition. **d.** Superposition of two experimentally observed IC8 conformations; 817 Cα atoms of IC8 and IC8′ aligned with an rmsd of 0.67 Å; IC8 (green y-mtRNAP, yellow MTF1, pink template, cyan NT, and green RNA) and IC8′ (gray y-mtRNAP, blue MTF1, orange template, NT and RNA). **e.** IC8 structure defining the position and orientation of the template (pink) in ICs. Relative positions of the template −1G nucleotide from IC2 to IC6 (**f**) and IC6 to IC8 (**g**) show how the template is increasingly scrunched as the transcription bubble expands; the templates are aligned based on RNAP Cα superpositions. **h.** Superposition of y-mtRNAP IC8 and h-mtRNAP EC (PDB ID: 4BOC); 486 Cα atoms are aligned with an rmsd of 1.4 Å. The y-mtRNAP IC8 structure is colored: steel blue y-mtRNAP, green RNA, pink template, cyan non-template, and h-mtRNAP EC is colored: blue NT, dark gray RNA, and gray template and h-mtRNAP. The missing NT strand in EC is represented by the blue dotted line.

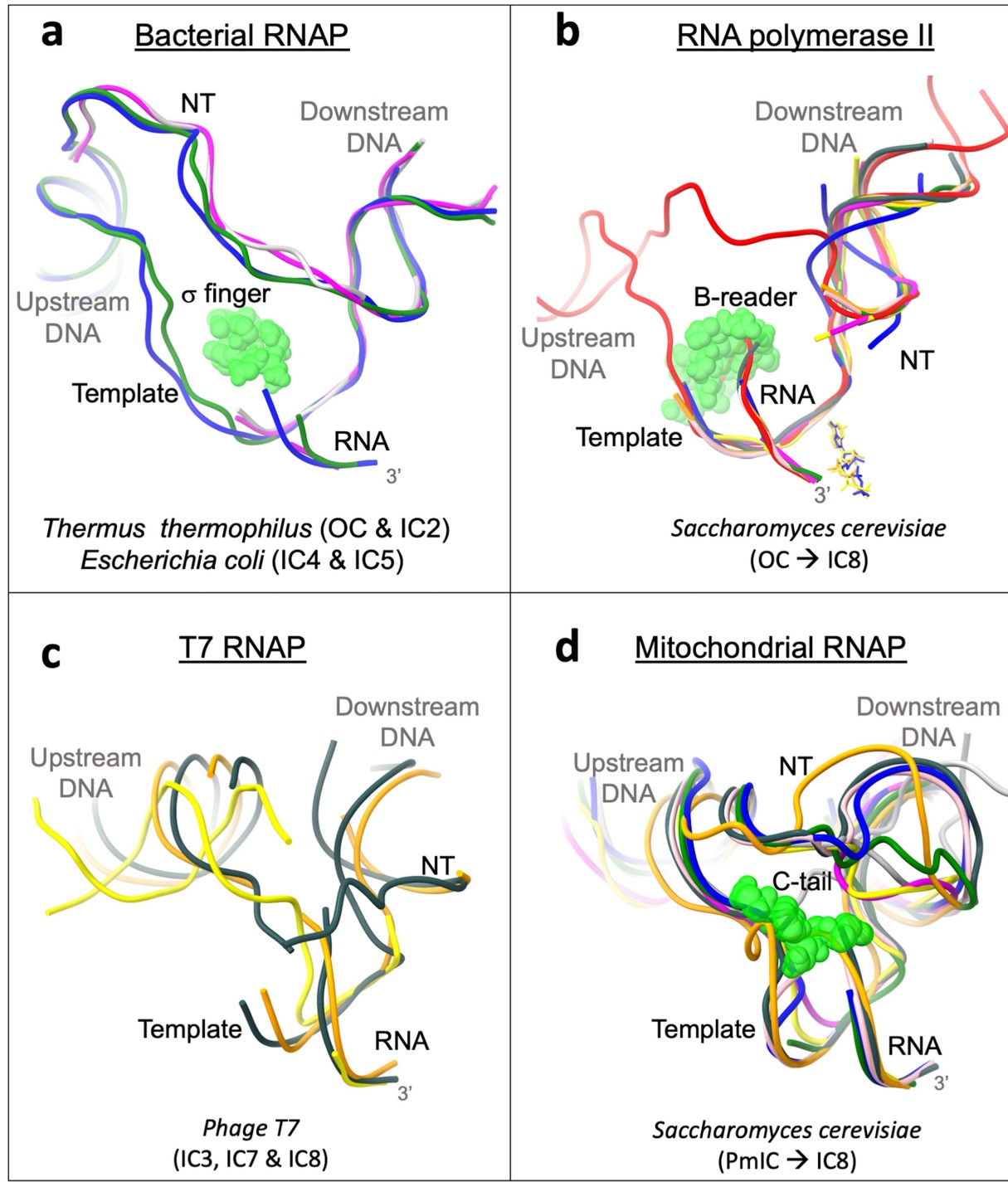

**a** Bacterial RNAP

NT
Downstream DNA
Upstream DNA
σ finger
Template
RNA
3'

*Thermus thermophilus* (OC & IC2)
*Escherichia coli* (IC4 & IC5)

**b** RNA polymerase II

Downstream DNA
Upstream DNA
B-reader
RNA
NT
Template
3'

*Saccharomyces cerevisiae*
(OC → IC8)

**c** T7 RNAP

Downstream DNA
Upstream DNA
NT
Template
RNA
3'

*Phage T7*
(IC3, IC7 & IC8)

**d** Mitochondrial RNAP

Downstream DNA
Upstream DNA
NT
C-tail
Template
RNA
3'

*Saccharomyces cerevisiae*
(PmIC → IC8)

Color codes: PmIC/OC; IC2; IC3; IC4; IC5; IC6; IC7; IC8

**Extended Data Fig. 8 | Structural states of the transcription bubbles in ICs of bacterial RNAP (a), RNA Pol II (b), T7 RNAP (c) and y-mtRNAP (d).** The IC structures of bacterial RNAP, RNA Pol II, T7 RNAP, and y-mtRNAP are superimposed in respective figures. **a**. *Thermus thermophilus* open complex (OC) (PDB ID: 4G7H – light gray), *Thermus thermophilus* IC2 (PDB ID: 4G7O – magenta), *E. coli* IC4 (PDB ID: 4YLO – green) and *E coli* IC5 (PDB ID: 4YLP – blue) are aligned. The σ finger is shown in lime, space filling representation. **b**. RNA Pol II (*S. Cerevisiae*) OC (PDB ID: 4A3I – light gray), IC2 (PDB ID: 4A3G – magenta), IC3 (PDB ID: 4A3J – yellow), IC4 (PDB ID: 4A3M – green), IC5 (PDB ID: 4A3E –

blue), IC6 (PDB ID: 4A3F – pink), IC7 (PDB ID: 4A3K – dark gray), IC8 (PDB ID: 4A3L – orange), and IC9 (PDB ID: 5C4J – red), The B-reader is shown in lime, space-filling representation (PDB ID: 4BBS). **c**. T7 RNAP IC3 (PDB ID: 1QLN – yellow), IC7 (PDB ID: 3E2E – dark gray) and IC8 (PDB ID: 3E2J). **d**. y-mtRNAP PmIC (PDB ID: 6YMV – light gray), IC2 (PDB ID: 8AP1 – magenta), IC3 (PDB ID: 6YMW – yellow), IC4 (PDB ID: 8ATT – green), IC5 (PDB ID: 8ATV – blue), IC6 (PDB ID: 8ATW – pink), IC7 (PDB ID: 8C5S – dark gray) and IC8 (PDB ID: 8C5U – orange). The C-tail is shown in lime, space-filling representation.

**Extended Data Table 1 | Cryo-EM data collection, refinement, and validation statistics for IC2, IC4, IC5, and IC6 structures**

| | IC2 (y-mtRNAP; MTF1; dsDNA; 2GTP) | IC4 (y-mtRNAP; MTF1; dsDNA; pppGpGpUpA) | IC5 (y-mtRNAP; MTF1; dsDNA; pppGpGpApApA) | IC6 (y-mtRNAP; MTF1; dsDNA; pppGpGpApApApU) |
|---|---|---|---|---|
| | (EMD-15556) (PDB 8AP1) | (EMD-15662) (PDB 8ATT) | (EMD-15664) (PDB 8ATV) | (EMD-15665) (PDB 8ATW) |
| **Data collection and processing** | | | | |
| Magnification | 190,000x | 150,000x | 150,000x | 150,000x |
| Voltage (kV) | 200 | 200 | 200 | 200 |
| Electron exposure (e–/Å$^2$) | 40 | 40 | 40 | 40 |
| Defocus range (μm) | -0.8 to -2.2 | -0.8 to -1.8 | -0.8 to -2.2 | -0.8 to -1.8 |
| Pixel size (Å) | 0.76 | 0.97 | 0.97 | 0.97 |
| Symmetry imposed | C1 | C1 | C1 | C1 |
| Initial particle images (no.) | 1,149,870 | 671,969 | 549,988 | 940,973 |
| Final particle images (no.) | 137,631 | 138,730 | 91,298 | 144,092 |
| Map resolution (Å) | 3.47 | 3.44 | 3.39 | 3.62 |
| FSC threshold | 0.143 | 0.143 | 0.143 | 0.143 |
| Map resolution range (Å) | 2.7 to 3.5 | 3.1 to 4.5 | 3.1 to 4.5 | 3.1 to 5 |
| | | | | |
| **Refinement** | | | | |
| Initial model used (PDB code) | 6YMV/6YMW | 6YMV/6YMW | 6YMV/6YMW | 6YMV/6YMW |
| Model resolution (Å) | 3.0 | 3.3 | 2.9 | 3.3 |
| FSC threshold | 0.143 | 0.143 | 0.143 | 0.143 |
| Model resolution range (Å) | 100 to 3.0 | 100 to 3.3 | 100 to 2.9 | 100 to 3.3 |
| Map sharpening $B$ factor (Å$^2$) | -144 | -80 | -140 | -125 |
| Model composition | | | | |
| Non-hydrogen atoms | 11,221 | 11,124 | 11,161 | 11,283 |
| Protein residues | 1,257 | 1,246 | 1,257 | 1,258 |
| Ligands | N/A | N/A | N/A | N/A |
| $B$ factors (Å$^2$) | | | | |
| Protein | 22.34 | 88.32 | 28.60 | 75.27 |
| Ligand | N/A | N/A | N/A | N/A |
| R.m.s. deviations | | | | |
| Bond lengths (Å) | 0.006 | 0.006 | 0.005 | 0.006 |
| Bond angles (°) | 1.057 | 1.076 | 1.006 | 1.023 |
| Validation | | | | |
| MolProbity score | 1.59 | 1.61 | 1.60 | 1.60 |
| Clashscore | 5.85 | 10.00 | 6.95 | 8.42 |
| Poor rotamers (%) | 0.27 | 0.18 | 0.00 | 0.00 |
| Ramachandran plot | | | | |
| Favored (%) | 96.07 | 97.58 | 96.64 | 97.20 |
| Allowed (%) | 3.93 | 2.42 | 3.36 | 2.80 |
| Disallowed (%) | 0.00 | 0.00 | 0.00 | 0.00 |

**Extended Data Table 2 | Cryo-EM data collection, refinement, and validation statistics for IC7, IC8, and IC8′ structures**

| | IC7 (y-mtRNAP; MTF1; dsDNA; pppGpGpUpApApApU) | IC8 (y-mtRNAP; MTF1; dsDNA; pppGpGpUpApApApUpG) | IC8′ (y-mtRNAP; MTF1; dsDNA; pppGpGpUpApApApUpG) |
|---|---|---|---|
| | (EMD-16442) (PDB 8C5S) | (EMD-16443) (PDB 8C5U) | (EMD-18183) (PDB 8Q63) |
| **Data collection and processing** | | | |
| Magnification | 150,000x | 150,000x | 150,000x |
| Voltage (kV) | 200 | 200 | 200 |
| Electron exposure (e–/Å²) | 40 | 40 | 40 |
| Defocus range (µm) | -0.8 to -1.8 | -0.8 to -1.8 | -0.8 to -1.8 |
| Pixel size (Å) | 0.97 | 0.97 | 0.97 |
| Symmetry imposed | C1 | C1 | C1 |
| Initial particle images (no.) | 641,911 | 877,163 | 877,163 |
| Final particle images (no.) | 96,741 | 119,532 | 129,204 |
| Map resolution (Å) | 3.75 | 3.62 | 3.68 |
| FSC threshold | 0.143 | 0.143 | 0.143 |
| Map resolution range (Å) | 3.5 to 5.0 | 3.5 to 5.0 | 3.5 to 5.0 |
| | | | |
| **Refinement** | | | |
| Initial model used (PDB code) | 6YMV/6YMW | 6YMV/6YMW | 6YMV/6YMW |
| Model resolution (Å) | 3.4 | 3.3 | 3.3 |
| FSC threshold | 0.143 | 0.143 | 0.143 |
| Model resolution range (Å) | 100 to 3.4 | 100 to 3.3 | 100 to 3.3 |
| Map sharpening $B$ factor (Å²) | -144 | -80 | -80 |
| Model composition | | | |
| Non-hydrogen atoms | 11,312 | 11,419 | 11,346 |
| Protein residues | 1,260 | 1,248 | 1,244 |
| Ligands | N/A | N/A | N/A |
| $B$ factors (Å²) | | | |
| Protein | 31.28 | 90.67 | 100.52 |
| Ligand | N/A | N/A | N/A |
| R.m.s. deviations | | | |
| Bond lengths (Å) | 0.007 | 0.006 | 0.007 |
| Bond angles (°) | 1.261 | 1.044 | 1.242 |
| Validation | | | |
| MolProbity score | 1.72 | 1.59 | 1.91 |
| Clashscore | 7.99 | 7.66 | 12.26 |
| Poor rotamers (%) | 0.00 | 0.00 | 0.09 |
| Ramachandran plot | | | |
| Favored (%) | 95.93 | 97.01 | 95.54 |
| Allowed (%) | 4.07 | 2.99 | 4.46 |
| Disallowed (%) | 0.00 | 0.00 | 0.00 |

# Reporting Summary

## Statistics

For all statistical analyses, confirm that the following items are present in the figure legend, table legend, main text, or Methods section.

| n/a | Confirmed | |
|---|---|---|
| ☐ | ☒ | The exact sample size (*n*) for each experimental group/condition, given as a discrete number and unit of measurement |
| ☐ | ☒ | A statement on whether measurements were taken from distinct samples or whether the same sample was measured repeatedly |
| ☒ | ☐ | The statistical test(s) used AND whether they are one- or two-sided *Only common tests should be described solely by name; describe more complex techniques in the Methods section.* |
| ☒ | ☐ | A description of all covariates tested |
| ☒ | ☐ | A description of any assumptions or corrections, such as tests of normality and adjustment for multiple comparisons |
| ☐ | ☒ | A full description of the statistical parameters including central tendency (e.g. means) or other basic estimates (e.g. regression coefficient) AND variation (e.g. standard deviation) or associated estimates of uncertainty (e.g. confidence intervals) |
| ☒ | ☐ | For null hypothesis testing, the test statistic (e.g. *F*, *t*, *r*) with confidence intervals, effect sizes, degrees of freedom and *P* value noted *Give P values as exact values whenever suitable.* |
| ☒ | ☐ | For Bayesian analysis, information on the choice of priors and Markov chain Monte Carlo settings |
| ☒ | ☐ | For hierarchical and complex designs, identification of the appropriate level for tests and full reporting of outcomes |
| ☒ | ☐ | Estimates of effect sizes (e.g. Cohen's *d*, Pearson's *r*), indicating how they were calculated |

*Our web collection on statistics for biologists contains articles on many of the points above.*

## Software and code

Policy information about availability of computer code

| Data collection | EPU 2.9.0 (for single-particle cryo-EM data collection on Glacios) |
|---|---|
| Data analysis | Relion 3.1, CTFFIND-4 and ResMap 1.1.5 for data-processing; Coot 0.9.4 and Phenix 1.19.2-4158 for model building and structure refinement; PyMol 2.4.1, Chimera 1.15rc, ChimeraX 1.2.5 to generate structural figures and movies. ImageQuant version 7 for visualization, quantification and analysis of transcription assays. GraphPad Prism 9.5.1 for visualization and plot presentation of transcription assays. Dynamics software version 7.10.0.23 to confirm hydrodynamic radius and polydispersity. 3DFSC webserver for 3D-FSC cryo-EM plots. |

For manuscripts utilizing custom algorithms or software that are central to the research but not yet described in published literature, software must be made available to editors and reviewers. We strongly encourage code deposition in a community repository (e.g. GitHub). See the Nature Portfolio guidelines for submitting code & software for further information.

## Data

Policy information about availability of data

All manuscripts must include a data availability statement. This statement should provide the following information, where applicable:
- Accession codes, unique identifiers, or web links for publicly available datasets
- A description of any restrictions on data availability
- For clinical datasets or third party data, please ensure that the statement adheres to our policy

The coordinates and cryo-EM density maps for IC2, IC4, IC5, IC6, IC7, IC8 and IC8' have been deposited under PDB and EMDataBank accession codes 8AP1, 8ATT,

8ATV, 8ATW, 8C5S, 8C5U and 8Q63, and EMD-15556, EMD-15662, EMD-15664, EMD-15665, EMD-16442, EMD-16443 and EMD-18183, respectively. For model building, the following models from PDB were used: 6YMV and 6YMW. AlphaFold accession number P13433 was used for γ-ins region model building. All other related data generated and/or analyzed during this research, such as raw cryo-EM images, are available from the corresponding authors upon reasonable request. Source data and raw gels are also available.

# Research involving human participants, their data, or biological material

Policy information about studies with human participants or human data. See also policy information about sex, gender (identity/presentation), and sexual orientation and race, ethnicity and racism.

| Reporting on sex and gender | N/A |
|---|---|
| Reporting on race, ethnicity, or other socially relevant groupings | N/A |
| Population characteristics | N/A |
| Recruitment | N/A |
| Ethics oversight | N/A |

Note that full information on the approval of the study protocol must also be provided in the manuscript.

# Field-specific reporting

Please select the one below that is the best fit for your research. If you are not sure, read the appropriate sections before making your selection.

☒ Life sciences ☐ Behavioural & social sciences ☐ Ecological, evolutionary & environmental sciences

For a reference copy of the document with all sections, see nature.com/documents/nr-reporting-summary-flat.pdf

# Life sciences study design

All studies must disclose on these points even when the disclosure is negative.

| Sample size | No statistical methods were used for sample size calculations. Six cryo-EM data sets were collected on the yeast mitochondrial RNA polymerase transcription initiation complex and all data sets yielded sufficient amount of good particles after classification to obtain high-resolution density maps. Biochemical and functional assays have been repeated 2 to 3 times independently and showed to have excellent reproducibility. |
|---|---|
| Data exclusions | No data was excluded for this study. Only poor particles were discarded during the cryo-EM data processing as it is a vital part of the workflow. |
| Replication | All biochemical and functional assays have been repeated 2 to 3 times independently and indicated reproducibility. Cryo-EM images were reproducibly obtained on different days, collections and with different protein purifications. |
| Randomization | Randomization was not performed as experimental groups are not needed for this work and thus it is not applicable for this current study. |
| Blinding | No experimental groups or randomization is required for this work, hence blinding is not applicable to this study. |

# Reporting for specific materials, systems and methods

We require information from authors about some types of materials, experimental systems and methods used in many studies. Here, indicate whether each material, system or method listed is relevant to your study. If you are not sure if a list item applies to your research, read the appropriate section before selecting a response.

## Materials & experimental systems

| n/a | Involved in the study |
|-----|----------------------|
| ☒ | Antibodies |
| ☒ | Eukaryotic cell lines |
| ☒ | Palaeontology and archaeology |
| ☒ | Animals and other organisms |
| ☒ | Clinical data |
| ☒ | Dual use research of concern |
| ☒ | Plants |

## Methods

| n/a | Involved in the study |
|-----|----------------------|
| ☒ | ChIP-seq |
| ☒ | Flow cytometry |
| ☒ | MRI-based neuroimaging |

nature portfolio | reporting summary

April 2023

