## [Peer Review File · Nature]

Manuscript Title: Cryo-EM structures illustrate step-by-step yeast mitochondrial transcription initiation

Reviewer Comments & Author Rebuttals

Reviewer Reports on the Initial Version:

Referees' comments:

Referee #1 (Remarks to the Author):

In this manuscript, Goovaerts, Shen et al. report cryo-EM structures of several yeast mitochondrial RNA polymerase transcription initiation complexes. By using a pre-melted DNA scaffold and different combinations of ribonucleotides and/or dinucleotides, they stabilize and resolve initially transcribing complexes (ICs) with 2mer, 3mer, 4mer, 5mer and 6mer RNAs.

This study is technically well carried out, and the structural data is of high quality. It provides detailed new insights into the mechanism of transcription initiation in yeast mitochondria. In combination with a previous study from the same corresponding authors (PMID 33278362) it for the first time reveals the distinct steps from initial RNA synthesis to synthesis of a 6mer RNA by γ -mtRNAP. These structures show how γ -mtRNAP progressively accommodates a growing RNA chain while stabilizing the non-template and template strands in surprising conformations. In contrast to most previous studies on mitochondrial and other RNA polymerases, the non-template strand is essentially entirely resolved in these structures, which is likely due to the fact that it is stabilized in a unique fashion by the transcription initiation factor MTF1 in the mitochondrial system.

My major criticism is that the authors miss the chance to assess how generalizable the observed mechanisms of transcription initiation are, and instead make some strong claims about the conservation of initiation steps in T7 as well as multisubunit RNA polymerases. However, there are no structural comparisons to available data from the prokaryotic or the Pol II RNAP system given. For example, structures of initially transcribing complexes of Pol II with 2, 4, 5, 6, and 7 nt RNAs have been reported (Sainsbury et al., 2011). However, this is neither cited nor are these structures compared to the ICs reported here. In the absence of this, it is unclear whether the structural states and rearrangements here are specific to the (yeast) mitochondrial system, or reveal general principles of transcription initiation. While the authors hypothesize such generalizability, they do not provide compelling analyses that would allow the reader to follow this reasoning. In contrast, I think many features could equally well be argued to likely be mitochondria-specific. For example, the hypothesis that NT strand stacking and stabilization may be a general mechanism during RNAP initiation is speculative in my opinion, as NT strand stabilization is achieved by the mitochondria-specific initiation factor MTF1. Similarly, I think the assumption that most steps are conserved in single-subunit RNAP initiation is at least questionable, as T7 RNAP undergoes very different conformational states during the transition from IC to EC than mtRNAP.

Overall, I think the results reported here are of high quality and provide exciting new insights into the mechanisms of transcription initiation by the unique mitochondrial transcription apparatus. However, I believe this manuscript would benefit from a more thorough structural comparison to other transcription systems in order to clarify which aspects are specific to the mitochondrial system and which may be more general. Specifically, I suggest the authors address the following points:

Major points

- Regarding the structural data:

IC2: Chain T, residue 22 has virtually no density and should be omitted from the model in my opinion.

IC4: Chain T, residues 22-24 have very poor density and should be omitted from the model in my opinion.

The authors should supply unsharpened half-maps and masks used for post-processing with the EMDB entry.

- A general point that is not addressed in the manuscript is what effect the DNA scaffold used may have on the complexes observed by structural analysis. The scaffold contains a 6nt mismatch bubble, which is a classical approach to stabilize transcribing RNA polymerase complexes. However, this scaffold may pose a barrier to polymerase progression, as the separated strands cannot re-anneal upstream of the transcription bubble, which could in principle favor formation or stabilization of unnatural (scrunched) states in which the polymerase synthesizes RNA but is hindered from promoter escape. For example, Fig 2b lane 5 shows accumulation of an intermediate product smaller than the run-off product, which could indicate a barrier to promoter escape. Have the authors addressed this possibility, for example by comparing longer run-off transcription between the bubble scaffold used for structure determination and an identical but fully complementary promoter DNA strand? I think this possible limitation should at least be discussed and if possible addressed experimentally.

- Line 35 -36: In light of the points below, I think this should read "...regulating transcription initiation in yeast mitochondria".

- Line 358 – 361: The authors hypothesize that NT strand stacking may be a general mechanism of stabilizing transition states during initiation, but do not provide a compelling rationale for this assumption. As they correctly state, the NT strand of transcription bubbles is poorly resolved in most structures of other transcription systems, including bacterial RNAP and Pol II. In my opinion, this argues against an ordered, stacked structure of the NT strand as observed here, as the ordering in the γ -mtRNAP complexes is the reason it becomes visible in the first place.

- Lines 381 – 395: The assumption that the mechanism of transcription initiation observed here for γ -mtRNAP is generally conserved in all RNAPs (or even T7 RNAPs) is in my opinion speculative and not supported by data. While it may very well be true for the very first steps of RNA synthesis, both γ -mtRNAP and h-mtRNAP undergo a rearrangement of the upstream DNA during the transition from IC to EC (statement iii). Already the closely homologous T7 RNAP does not follow the same mechanism and instead undergoes a large conformational rearrangement of the polymerase (Tahirov 2002, Yin & Steitz 2002). This should at least be mentioned here, and it should be clarified what limitations the generalization has based on available data.

- Please provide a (supplementary) figure that highlights the cryo-EM workflow and data quality, in particular a directional FSC plot for each structure and viewing angle distribution.

Minor points

- Figure 1a, upper panel: IC4: Was GTP added or ATP? I do not see how GTP would form IC4, and in Figure 1d IC4 is depicted with an RNA sequence of GGUA.

- Lines 238 – 240: This is very good hypothesis, but not experimentally proven or modeled anywhere. Please tone down (for example "may create space...").

- Line 332 – 333: The authors should note here that this movement has been proposed previously for the h-mtRNAP (ref 20).

- Line 372: This statement requires citations, which should include PMID 19945377.

- Figure 5d: How was this EC model obtained?

Referee #2 (Remarks to the Author):

Goovaerts et al. determine four cryo-EM structures of yeast mitochondrial RNAP initial transcribing complexes by using scaffolds containing a -4 to +2 mismatched region to mimic the transcription bubble. Supplementing different combinations of NTPs and/or 2mer initiator RNA the authors obtain complexes using cryo-EM at different stages of initial transcription, specifically 2-mer, 4-mer, 5-mer, and 6-mer RNA (IC2, IC4, IC5, IC6). Together with their previously determined 3-mer (IC3) and partially melted initiation complexes (PmIC), they structurally describe a step-by-step mechanism of yeast mitochondrial RNAP initial RNA synthesis. The authors observe the formation of an extended base stacking ('spiral staircase') in the 5-mer containing complex. Furthermore, they provide evidence for the importance of the -1 base identity during late stages of initial transcription. Overall, this manuscript describes an extension of what the authors published in 2021 (PmIC and IC3); given all the issues (described below), these results may not meet the high standards of a Nature publication.

These are certainly interesting structures to think about, but there are a number of issues the authors need to address:

Major issues

1. The normal cryo-EM quality control figures, particularly the particle orientation distributions (extremely important), the FSCs, and the local resolution maps, are not to be found. Also, the processing pipelines for the cryo-EM data (also extremely important) detailing the processing steps showing the number of particles in each 3D class and what particles were kept and which were thrown out explicitly, along with corresponding map volumes are essential.

2. In the abstract (lines 18-29) and in lines 198-209, the authors discuss the idea of 'stress' induced by scrunching, which is supposed to 'drive abortive RNA synthesis during early transcription initiation steps and promoter release later'. In the text: 'A scrunched state is known to hold energy and has a tendency to unscrunch to release the stored energy, which destabilizes the RNA:DNA hybrid and releases short RNAs'. No reference(s) are provided for this... This seems like a lot of handwaving/speculation; I would argue that this is not known, and that it is unlikely to be true [see Vahia and Martin, 2011 (doi: 10.1021/bi200620q) and Samanta and Martin, 2013; 10.1074/jbc.m113.497669]. Vahia & Martin studied T7 RNAP (a homolog of the mitochondrial RNAP studied here) and found that nicking the DNA strands (which would presumably relieve 'scrunching stress') did not have any effect on abortive initiation. Furthermore, this 'scrunching stress' hypothesis is eminently testable (Vahia & Martin tested it...).

3. The complexes for cryo-EM were prepared at 4°C and each incubation (with MTF1 and subsequently with the NTPs) was for several hours. The protein is apparently active at 4°C (since the different RNAs were synthesized by the RNAP), but still, I wonder if things are really the same at 4°C? Most importantly, is the pattern of abortive products the same at 4°C than at 25°C? In the cases where in vitro transcription results are shown (Figs. 2b, 3a, 4d, 4g, Supplementary Fig. 5), the experiments were all done at 25°C.

4. The authors use a variant of γ -mtRNAP where the 100 residues at the N-terminus are missing. However, the main text refers to this variant simply as γ -mtRNAP, and no further explanation is given in the text as to why this deletion was made. The referenced functional studies which form the basis for most of this study's conclusions seem to use the full-length protein, and deletion of the N-terminal sequence was shown to impact transcription initiation efficiency and the degree of abortive RNA synthesis (ref 46). The authors must explain how their conclusions are impacted by this deletion and take it into consideration when discussing the transitions of the IC->EC states.

5. The authors observe a "spiral staircase" stacking of the scrunched NT strand in the IC5 structure. The scaffold used for IC5 consists only of adenine bases in the staircase region, while the staircase region in the scaffold used for IC2 and IC4 is disrupted by a T. Single-stranded homopolymers of deoxy-adenine are known to exhibit strong stacking preference, therefore naturally organizing into a spiral staircase. The authors conclude that stabilization of IC5 attained via the staircase facilitates processivity (lines 355f). However, as the authors argue in lines 265ff, this "staircase stabilization" of IC states may only hold true for certain sequences. How does the conclusion of the authors relate to natural mitochondrial promoter sequences?

6. Also, obviously I don't expect the authors to do time-resolved cryo-EM, but they should address the issue that these structures are at equilibrium (at 4°C) and may not reflect the actual dynamics of initiation. For example, regarding the nt-strand 'spiral staircase' observed in IC5, what if the rate of formation of that structure is slow compared to the rate of aborting the RNA product or binding the next NTP and elongating to the next step? Then in the dynamic situation of initiation, that spiral staircase structure would never form and its properties would be irrelevant...

7. There is also a lot of conjecture regarding the IC to EC transition (instead of actual structural information). On these constructs (pre-melted), the IC to EC transition is greatly disfavored at 25°C; Tang et al. JBC 2009 Fig 2 showed that the lengths and amounts of abortive vs productive products differ between transcribing from a duplex promoter or a pre-melted bubble (-4 to +2) construct (as used in this study). For the pre-melted promoter: 5- and 6-mers pile up versus very few from duplex promoter template (predominantly 2-mers and some 3-mers). Relatively little full length transcript is produced from a mismatched bubble, suggesting that the IC to EC conversion is greatly disfavored on this construct. Why even do this study on a pre-melted bubble?

8. Is the partially-melted initiation complex (PmIC) a significant intermediate on the initiation pathway? Previous studies done at 25°C at equilibrium adding ATP (can bind +1) does not change the size of the bubble (deduced is -4 to +2 based on 2-AP fluorescence). From this data, the authors deduced that NTP binding is not needed to form the full bubble. What evidence exists to support the PmIC being populated at room temperature or physiological temperatures with normal constructs (i.e. not pre-melted)?

9. If the IC complexes were on pathway and were stable,

i. wouldn't the authors have expected to see a mixture of intermediates (classes) in the IC4, IC5, IC6 mixes?

ii. It is truly surprising that the IC2 population is not detected to be heterogeneous given the model and observation that release of 2-mers in vitro dominates the start of initiation. To support the arguments on lines 144-145, the authors need to perform transcription experiments to demonstrate that dimers form at 40°C.

iii. Note that in Fig 2b (cryo-EM construct), there are no 2-mers when GTP and UTP are present. Why?

iv. Fig 2b: when GTP, UTP and ATP are present, see 3-mers and 4-mers. In cryoEM (Fig 1a) experiment, only see a 4 mer (IC4). Why?

v. Why did the authors use pppGpG instead of GTP for the IC4-IC6 experiments?

10. Only model with a bound Mg²⁺ is IC2 (the i+1 NTP is bound (triphosphate is still present)). The authors say IC2 is catalytically poised for bond synthesis but they did not model a second Mg²⁺?

11. Re IC to EC conversion: How can the bubble collapse (e.g. rewind) when has been engineered to not rewind? Isn't that possibly why IC7 did not form?

12. What do the authors mean on lines 323-326 that a 6-mer is the last stable state? When duplex promoter DNA is used, the only significant products formed are 2,3 mer and runoff (e.g. Fig. 4g), indicating that any RNA longer than a 3-mer is rapidly extended and that $n > 3$ products are only transiently populated.

13. Fig 2b, none of the lanes corresponds to the nucleotide combinations used in the cryo-EM experiments on this construct (Fig. 1a), nor the condition of the cryo-EM experiment (different buffer, 40C). Does a 12-mer represent a product from an elongation complex when there is no remaining downstream duplex DNA?

Other issues

7. The authors show that variation of the -1 base identity severely affects transcription efficiency. What is the sequence conservation of the base and the interacting residues? How would efficient transcription be explained if a different base in that position occurs in other promoters?

8. In the IC 6 state, the -1 base inserts into a pocket at the thumb-domain. However, when considering the transition to the EC state later in the text (line 306ff), the -1 base capture is not mentioned although the preceding section highlights its importance through structural and biochemical evidence. What is the role of this pocket during the transition to EC?

9. Fig 1a: indicated GTP for IC4 but should be ATP (in addition to pppGpG /UTP).

10. line 209: "relative amount of 3-mer abortive" is not defined and confusing (relative to other abortive products, relative to runoff?).

11. In general, some of the figures are pretty difficult to digest...

A lot of the structural figures need an 'inset' of the overall structure to help orient the viewer to the zoomed-in views in the context of the entire complex.

Is the active site Mg²⁺ visible - I don't see any density figures that show that clearly

Fig. 1e - why not show where the active site Mg²⁺'s are

Fig. 1f - orientation is difficult to figure out

Fig. 2d is very difficult to digest

Figs. 3b, c, d, f, g - it would be helpful to show the RNA somehow (I understand all of them can't be shown because it would be way too busy, but perhaps show the longest RNA in each panel...

Fig. 4e - the important point here is what is happening with the -1 base, but this is obscured by the thick, densely colored backbone worms. One could make the backbone worms much thinner and a very light color.

Fig. 5c and 5d is supposed to illustrate the large change in the upstream DNA on the transition from IC6 to EC, but the view in 5C doesn't show the upstream DNA...

12. Fig 4g: The underlined bases in the sequence are not defined in the figure legend.

13. Fig 4g/Supplement Fig 5: How often has the experiment been repeated? If it was $n \geq 3$ it would be useful to add error bars to the quantifications shown in Supplement Fig 5 a-d.

14. Fig 3a: Legend is missing information on the quantification shown on the right-hand side as well as some explanation on the sequence (what are the red bases? What does the underline mean?)

15. The included movies are a little misleading because they show the NTPs flying in a way that would be impossible (i.e. through the protein). The authors should either show the NTPs coming in the way we know they do (this RNAP has an NTP entry tunnel), or at least don't show the NTPs flying in, just have them 'appear' in their binding sites

16. Lines 461-464: total final concentrations of NTPs used should also be given. Would be helpful if the authors' converted 3 mg/mL of the PmlC to molar concentration as well.

17. How were the concentrations of the PmlCs determined (e.g. lines 453, 459)?

18. Please give origins of the NTPs used? E.g. where did pppGpG come from?

19. Incorrect or incomplete set of NTPs given in Fig 1a for IC4 and IC6 re NTPs added, please correct.

Referee #3 (Remarks to the Author):

This manuscript by Goovaerts and colleagues is an expansion of previous work by the same group on the structural characterization of the initiation complex of the yeast mitochondrial RNA polymerase. Whereas previously the authors had only been able to obtain two snapshots of the process, the current manuscript completes a detailed characterization of the first six nucleotide additions during initiation. Given that transition to elongation mode for this polymerase appears to take place after the eight incorporation (based on previous work from some of the same authors), this study provides a very complete picture of the initiation process.

Although the general characteristics of the yeast mitochondrial transcription initiation process had already been established, including the fact that it involves template scrunching, the current manuscript is nevertheless a breakthrough in that it provides mechanistic understanding on how the mtRNAP is able to progressively accommodate more and more scrunched nucleotides as the initiation process proceeds. Furthermore, it reveals surprising structural differences between the complexes, including evidence of scrunching in the non-template strand in the early steps of the process, that provide some insight into the factors that determine the balance between abortive synthesis and further elongation. The authors have further extended the insight provided by their structures by carrying out modeling of the process of transition to the elongation phase, and the movies provided will ensure that this work will reach a wide audience.

In my opinion, the manuscript could be strengthened by polishing some of the illustrations and clarifying some of the wording. The four structures provide a clear picture of the dynamics of the scrunching process and reveal some interesting hints about the factors that result in abortive termination. However, these aspects of the manuscript seem fairly speculative in the absence of additional experimental evidence. Nevertheless, this is a body of work that will be of immense interest to the field.

I propose the following points for consideration by the authors.

Major points

-The authors propose that the conformations of the non-template (NT) strand contribute to abortive synthesis. This is because they observe a different conformation of the non-template strand in the IC2 and IC3 complexes that is suggestive of scrunching, and that this conformation disappears in IC4 and subsequent complexes. As the authors point out in the discussion, there is an apparent correlation of this conformation with a higher likelihood of abortive termination, but in my opinion the manuscript does not present any evidence that this conformation is in fact related to the biochemical behavior. To support the importance of NT scrunching, the authors point out that it has been previously shown that deletions of the C-terminal tail of MTF1 result in a reduction in abortive products, arguing that this evidence supports their claim since the C-terminal tail is located in the proximity of the relevant NT nucleotides. However, deleting the last 4 nucleotides of this tail (D4 deletion) in MTF1 is sufficient to lead to a drastic decrease in abortive transcript production, and neither of these four residues appear to interact with the non-template strand in IC2. (In this respect, the authors might have noticed that E338, the last residue deleted in the A4 deletion, does appear to stabilize one of the non-templated nucleotides in IC3, and that the biochemical phenotype in Basu et al. might be consistent with this). Further complicating the issue, the NT conformations in IC2 and IC3 are not identical, and the density clearly indicates that there is some conformational variability even in IC2 (although not to the same extent as in the other structures). Moreover, in some cases, the difference in sequence (purine vs pyrimidine)

might affect the specific conformation adopted by the NT strand, perhaps accounting for some of these differences. As things stand, this conclusion appears highly speculative. Perhaps the authors could strengthen their claim by providing additional evidence based on mutagenesis. Unfortunately given that the observed conformations differ, it does not appear that there is a single interaction that might account for the IC2 and IC3 conformations, but the authors might be able to gain some insight by characterizing the phenotype of additional side chain mutations. For instance, have the authors attempted to substitute the Arg533 side chain?

- That the authors obtained exclusively the GTP-bound IC2 state is surprising. The explanation provided in the manuscript is convincing, but it would necessitate that catalysis be orders of magnitude slower than nucleotide binding (which seems likely) and 2-mer dissociation (which is perhaps less likely). In their previous work, the absence of complexes containing the 2-mer but no nucleotide can be explained by a very fast rate of nucleotide binding. In the present work, the absence of 2-mer complexes would imply equally fast dissociation. Moreover, if catalysis is that slow, this seems perhaps at odds with the inability to observe nucleotide-containing intermediates in the subsequent steps. An alternative explanation is that the IC2 complex is not catalytically competent. It is not entirely clear from the methods if the purified PmICs contain Mg, or if Mg is added together with the NTPs (does the buffer used for size-exclusion chromatography contain Mg?). Moreover, while elongation of the longer complexes can evidently take place at 4C, could that initial reaction be very inefficient at that temperature? Have the authors been able to observe initiation in reactions carried out at 4C in the conditions used to incubate their complexes?

Minor points

-The first paragraph of the introduction focuses on DNA scrunching, but while it appears to provide a general introduction to the process, the authors mention details that are specific to particular systems (e.g. 8-12 nt to transition to elongation mode, which is the case for T7 RNAP, when the authors have demonstrated that this is not the case for yeast mtRNAP). I would suggest rewording this paragraph to either clarify that some of the statements are applicable to specific polymerases or to make the statements more general.

-“The transition from PmIC to IC2 occurs with a large conformational change that moves the template strand by $\sim 22 \text{ \AA}$ ” – This is not an informative statement unless the reference points are also indicated, and I find that it might be misleading. What moves by 22A? Most of the template strand atoms are in fact within 1A of their original position.

-In Figure 1a, it would be helpful to color-code the differences between the two substrates.

-I wonder if the left image in Figure 2b could be made more clear.

-Figures 2e and 2f are not very helpful to illustrate the points made in the text. Out of context, the shift from +3/+4 to +8/+9 can only be grasped from the annotations, which are no better than what is stated in the text. Moreover, no interactions are shown, so it is not easy to determine whether indeed there is a loss of DNA interactions.

-It would be useful to provide a supplementary figure that clearly shows the conformation of all the atoms in the variable region of the NT strand in each of the complexes (including PmIN and IC3), and not just the backbone.

-It might make more sense to present the quantification of Figure 3a in the same figure and not in Figure 4. What are the dots in Figure 4d? Have these experiments been repeated? If not, they should be, and means +/- SE should be reported in the figure.

-Line 437, it should be “were”, no “was”.

-Line 224, “next”, not “net”.

Author Rebuttals to Initial Comments:

Response to Reviewers

We thank the reviewers for their constructive comments and suggestions that have been very helpful in preparing this revised manuscript. Overall, the reviewers found the structures to be of high quality, providing new insights into the mechanism of transcription initiation. A careful reading of the critiques reveals three major issues raised by the reviewers: 1) lacking a description of the generality of our findings through a comparison of the yeast mtRNAP structures to other known IC intermediates like in Pol II and bacterial RNAP, 2) requiring the biochemical evidence for our model that non-template scrunching contributes to abortive synthesis of short RNAs, and 3) lacking evidence that template scrunching and non-template platform contribute to IC→EC transition. We are happy to report that all three of these major concerns have been addressed through new biochemical experiments (suggested by reviewer 2) and two new structures of IC7 and IC8 states. In particular, IC8 structure traps a transient state between IC → EC transition and add valuable insights.

Comparing existing structures of Pol II and bacterial transcription bubbles in various IC states to y-mtRNAP bubble has been instructive in revealing similarities and differences that we have briefly discuss in pages 12 – 13 and show in Supplementary Fig. 11. The new IC8 structure clearly shows that the non-template staircase and scrunched template participate in dissociating the transcription factor MTF1 to facilitate EC transition. The new biochemical experiments show that relieving the stress from non-template strand by introducing nicks significantly decreases abortive synthesis (Fig. 2g and Supplementary Fig. 6). These are strong structural and biochemical evidences confirming that the non-template scrunching plays a role in abortive synthesis of short RNAs and scrunched template in the transition from transcription initiation to elongation.

We believe that the revised manuscript is greatly strengthened with the addition of new findings. Below we provide a point-by-point response to the reviewers' comments. The new results /discussions and specific modifications in response to the reviewers' comments are in blue in the updated manuscript.

Referee #1 (Remarks to the Author):

In this manuscript, Goovaerts, Shen et al. report cryo-EM structures of several yeast mitochondrial RNA polymerase transcription initiation complexes. By using a pre-melted DNA scaffold and different combinations of ribonucleotides and/or dinucleotides, they stabilize and resolve initially transcribing complexes (ICs) with 2mer, 3mer, 4mer, 5mer and 6mer RNAs.

This study is technically well carried out, and the structural data is of high quality. It provides detailed new insights into the mechanism of transcription initiation in yeast mitochondria. In combination with a previous study from the same corresponding authors (PMID 33278362) it for the first time reveals the distinct steps from initial RNA synthesis to synthesis of a 6mer RNA by y-mtRNAP. These structures show how y-mtRNAP progressively accommodates a growing RNA chain while stabilizing the non-template and template strands in surprising conformations. In contrast to most previous studies on mitochondrial and other RNA

polymerases, the non-template strand is essentially entirely resolved in these structures, which is likely due to the fact that it is stabilized in a unique fashion by the transcription initiation factor MTF1 in the mitochondrial system.

We appreciate the positive comments. We have added two new structures and biochemical experiments to the revised manuscript addressing the reviewer's concerns. This includes a comparison of existing IC intermediate bubbles of Pol II, bacterial RNAP, T7 RNAP with *y*-mtRNAP (Supplementary Fig. 11). After several trials, now we could obtain the IC7 and IC8 structures that have added significant insights into IC → EC transition. We now show that the template scrunching, expanding NT staircase, and clash of growing RNA:DNA hybrid contribute to destabilize the IC8 state for EC transition.

My major criticism is that the authors miss the chance to assess how generalizable the observed mechanisms of transcription initiation are, and instead make some strong claims about the conservation of initiation steps in T7 as well as multisubunit RNA polymerases. However, there are no structural comparisons to available data from the prokaryotic or the Pol II RNAP system given. For example, structures of initially transcribing complexes of Pol II with 2, 4, 5, 6, and 7 nt RNAs have been reported (Sainsbury et al., 2011). However, this is neither cited nor are these structures compared to the ICs reported here. In the absence of this, it is unclear whether the structural states and rearrangements here are specific to the (yeast) mitochondrial system or reveal general principles of transcription initiation. While the authors hypothesize such generalizability, they do not provide compelling analyses that would allow the reader to follow this reasoning. In contrast, I think many features could equally well be argued to likely be mitochondria-specific. For example, the hypothesis that NT strand stacking and stabilization may be a general mechanism during RNAP initiation is speculative in my opinion, as NT strand stabilization is achieved by the mitochondria-specific initiation factor MTF1. Similarly, I think the assumption that most steps are conserved in single-subunit RNAP initiation is at least questionable, as T7 RNAP undergoes very different conformational states during the transition from IC to EC than mtRNAP.

Overall, I think the results reported here are of high quality and provide exciting new insights into the mechanisms of transcription initiation by the unique mitochondrial transcription apparatus. However, I believe this manuscript would benefit from a more thorough structural comparison to other transcription systems in order to clarify which aspects are specific to the mitochondrial system and which may be more general.

Thank you for the encouraging comments on the high quality and exciting new insights. As pointed out by the reviewer, IC structures of Pol II are characterized and can be compared to the *y*-mtRNAP. Since the protein components are vastly different, we make a comparison of the transcription bubbles in different IC states of pol II structures (Cheung, 2011), bacterial RNAP structures (Zuo & Steitz, Mol Cell 2015; Zhang et al. Science 2012) in Supplementary Fig 11 and also discuss the finding of (Sainsbury et al., 2013) on page 12 (ref 40). We agree with the reviewer that the basic transcription steps are conserved in all RNAPs, but the specific mechanisms may differ between single-subunit and multi-subunit RNAPs. We have removed

general statements pointed out by the reviewer and now we compare only the features that are structurally or functionally related. We now briefly discuss known commonality and differences of the y-mtRNAP ICs with other single-stranded RNAP, bacterial RNAP, and pol II transcription initiation systems (Page 12).

Specifically, I suggest the authors address the following points:

Major points

- Regarding the structural data:

IC2: Chain T, residue 22 has virtually no density and should be omitted from the model in my opinion.

IC4: Chain T, residues 22-24 have very poor density and should be omitted from the model in my opinion.

The authors should supply unsharpened half-maps and masks used for post-processing with the EMDB entry.

We have deposited the B-sharpened, unsharpened, both half maps, and FSC plots in EMDB. As per the reviewer's suggestion, we will also deposit the masks for all structures.

We rechecked the regions of the IC2 and IC4 structures.

In IC2 - We have now removed the base of the template nucleotide 22, and the map-to-model correlation is now 0.44.

In IC4 – The template nucleotides 23 has been removed, and 22 and 24 are modeled to the unsharpened density map after removing the bases. Then carried out the real-space structure refinement using the B-sharpened map. The map-to-model correlations for the sugar-phosphate backbone are 0.51 and 0.40 for the nucleotides 22 and 24, respectively. Although, the position and conformation of the nucleotides are less reliable, we think the sugar-phosphate backbone with reasonable model-to-map correlation will be informative.

- A general point that is not addressed in the manuscript is what effect the DNA scaffold used may have on the complexes observed by structural analysis. The scaffold contains a 6nt mismatch bubble, which is a classical approach to stabilize transcribing RNA polymerase complexes. However, this scaffold may pose a barrier to polymerase progression, as the separated strands cannot re-anneal upstream of the transcription bubble, which could in principle favor formation or stabilization of unnatural (scrunched) states in which the polymerase synthesizes RNA but is hindered from promoter escape. For example, Fig 2b lane 5 shows accumulation of an intermediate product smaller than the run-off product, which could indicate a barrier to promoter escape. Have the authors addressed this possibility, for example by comparing longer run-off transcription between the bubble scaffold used for structure determination and an identical but fully complementary promoter DNA strand? I think this possible limitation should at least be discussed and if possible addressed experimentally.

We completely agree with the reviewer's insightful statement that the bubble would not support reannealing of the initially melted bases upstream of the start site; hence, will be defective in

promoter escape and transition into elongation. The suggested experiment with longer bubble and duplex promoters was reported in an earlier study published by Patel lab (Paratkar et al., JBC 286, pp16109-120, 2011). The long bubble promoter is active in making RNAs up to about 8-mer but poor at making runoff products (as suggested by the reviewer). Our new IC8 structure also trapped a transient state prior to upstream reannealing because of the mismatch bubble used for the structural study.

In general, the IC intermediates on a regular duplex promoter are not stable enough to capture for high resolution structural studies, most likely due to transcriptional bubble collapse, different start sites etc. Our previous single-molecule FRET studies showed that *y*-mtRNAP undergoes IC to EC transition at 8-nt synthesis step (Sohn et al, Nature comm. 2020, ref. 8). We have now trapped a transient IC8 state prior to upstream reannealing rather than the expected promoter-released states because of the mismatch bubble used for the structural study.

We have addressed above points raised here on page 10, paragraph 1 of the revised manuscript.

- Line 35 -36: In light of the points below, I think this should read "...regulating transcription initiation in yeast mitochondria".

We have revised the abstract and removed this line

- Line 358 – 361: The authors hypothesize that NT strand stacking may be a general mechanism of stabilizing transition states during initiation, but do not provide a compelling rationale for this assumption. As they correctly state, the NT strand of transcription bubbles is poorly resolved in most structures of other transcription systems, including bacterial RNAP and Pol II. In my opinion, this argues against an ordered, stacked structure of the NT strand as observed here, as the ordering in the *y*-mtRNAP complexes is the reason it becomes visible in the first place.

As pointed out by the reviewer, unlike in *y*-mtRNAP ICs, the NT strands in other RNAP IC structure are not well resolved. Hence, we cannot generalize that NT strand stacking would be used by other RNAPs to stabilize the IC intermediates. Accordingly, we have removed the line stating that hypothesis.

- Lines 381 – 395: The assumption that the mechanism of transcription initiation observed here for *y*-mtRNAP is generally conserved in all RNAPs (or even T7 RNAPs) is in my opinion speculative and not supported by data. While it may very well be true for the very first steps of RNA synthesis, both *y*-mtRNAP and *h*-mtRNAP undergo a rearrangement of the upstream DNA during the transition from IC to EC (statement iii). Already the closely homologous T7 RNAP does not follow the same mechanism and instead undergoes a large conformational rearrangement of the polymerase (Tahirov 2002, Yin & Steitz 2002). This should at least be mentioned here, and it should be clarified what limitations the generalization has based on available data.

Thank you for the suggestions. The mechanism of transcription initiation and transition by h- and y-mtRNAPs are expected to have significant similarities than those with T7 RNAP. The mtRNAPs use transcription factors to bind and melt the promoter DNA whereas T7 RNAP relies on its extensive base specific interactions with the promoter to catalyze these steps. Both MTF1 and TFB2M contain a C-tail that allows template alignment in the active site (Basu et al., 2020, ref. 29), whereas T7 RNAP uses -1 base unstacking to position the TSS in the active site (Ref. 38, 39). The mtRNAPs undergo IC to EC transition by dissociating/displacing the transcription factor from the promoter whereas T7 RNAP undergoes a large conformational change in its N-terminal domain to dissociate promoter interactions for transition to EC. Comparison shows that the upstream DNA in the T7 RNAP transcription bubbles shift in different IC states, unlike in y-mtRNAP as discussed in page 12 (paragraph 2) and shown in Supplementary Fig. 11.

- Please provide a (supplementary) figure that highlights the cryo-EM workflow and data quality, in particular a directional FSC plot for each structure and viewing angle distribution.

We have now included the workflow of the data processing for all structures in Supplementary Fig. 13 and 14.

Minor points

- Figure 1a, upper panel: IC4: Was GTP added or ATP? I do not see how GTP would form IC4, and in Figure 1d IC4 is depicted with an RNA sequence of GGUA.

The error has been corrected in Supplementary Fig. 1a of the revised manuscript.

- Lines 238 – 240: This is very good hypothesis, but not experimentally proven or modeled anywhere. Please tone down (for example “may create space...”).

The C-tail is constantly rearranged with each nucleotide addition. Large structural rearrangement in our new IC8 structure has pushed C-tail away from the transcription bubble region (Fig. 4b). Now this has been discussed in page 10, paragraph 1.

- Line 332 – 333: The authors should note here that this movement has been proposed previously for the h-mtRNAP (ref 20).

We have added the reference in page 11, last paragraph. “.. as proposed earlier for h-mtRNAP²⁰”

- Line 372: This statement requires citations, which should include PMID 19945377.

This part has been removed as we have shortened the discussion on C-tail and focus more on the findings from transient IC8 state.

- Figure 5d: How was this EC model obtained?

The EC model was obtained from the crystal structure of h-mtRNAP EC with a 9-mer RNA:DNA hybrid (Schwinghammer et al. 2013; ref. 24 in the manuscript). We clarify this in the revised manuscript under Materials and Methods, and added Supplementary Fig. 10 showing the superposition of h-mtRNAP EC on y-mtRNAP IC8.

Referee #2 (Remarks to the Author):

Goovaerts et al. determine four cryo-EM structures of yeast mitochondrial RNAP initial transcribing complexes by using scaffolds containing a -4 to +2 mismatched region to mimic the transcription bubble. Supplementing different combinations of NTPs and/or 2mer initiator RNA the authors obtain complexes using cryo-EM at different stages of initial transcription, specifically 2-mer, 4-mer, 5-mer, and 6-mer RNA (IC2, IC4, IC5, IC6). Together with their previously determined 3-mer (IC3) and partially melted initiation complexes (PmIC), they structurally describe a step-by-step mechanism of yeast mitochondrial RNAP initial RNA synthesis. The authors observe the formation of an extended base stacking ('spiral staircase') in the 5-mer containing complex. Furthermore, they provide evidence for the importance of the -1 base identity during late stages of initial transcription. Overall, this manuscript describes an extension of what the authors published in 2021 (PmIC and IC3); given all the issues (described below), these results may not meet the high standards of a Nature publication.

We thank the reviewer for the careful inspection of our manuscript and making constructive suggestions to rigorously test our models. DNA scrunching is observed now in all DNA-dependent RNAPs as part of the initiation mechanism and suggested to play a role in abortive synthesis and transition to elongation. However, high-resolution structures and supporting biochemical experiments are important for establishing the role of scrunching in each RNAP system. We have added two new structures of IC7 and IC8 intermediates that now provide structural evidence for template scrunching and non-template staircase in promoting the dissociation of the transcription factor MTF1 to facilitate the steps leading to EC transition, including promoter escape.

Additionally, we have conducted new experiments, as suggested by the reviewer, on nicked and nucleotide-deleted promoters. The experimental results are in agreement with the scrunching model derived from structures and clearly show significant drop in -2 and -3 RNA abortives when the NT is nicked and unable to scrunch, as discussed in page 7-8, and shown in Fig 2g and Supplementary Fig. 6. We have replaced the speculative description of -1 template role in IC to EC transition with a detailed description of the new biochemical data and IC8 structure, and plan to publish the -1 template data as a separate study.

These are certainly interesting structures to think about, but there are a number of issues the authors need to address:

Major issues

1. The normal cryo-EM quality control figures, particularly the particle orientation distributions (extremely important), the FSCs, and the local resolution maps, are not to be

found. Also, the processing pipelines for the cryo-EM data (also extremely important) detailing the processing steps showing the number of particles in each 3D class and what particles were kept and which were thrown out explicitly, along with corresponding map volumes are essential.

We have now added the flowcharts of data processing, FSC curves, and angular distribution of particles for each structure in Supplementary Figs. 13 and 14.

2. In the abstract (lines 18-29) and in lines 198-209, the authors discuss the idea of 'stress' induced by scrunching, which is supposed to 'drive abortive RNA synthesis during early transcription initiation steps and promoter release later'. In the text: 'A scrunched state is known to hold energy and has a tendency to unscrunch to release the stored energy, which destabilizes the RNA:DNA hybrid and releases short RNAs'. No reference(s) are provided for this...

This seems like a lot of handwaving/speculation; I would argue that this is not known, and that it is unlikely to be true [see Vahia and Martin, 2011 (doi: 10.1021/bi200620q) and Samanta and Martin, 2013; 10.1074/jbc.m113.497669]. Vahia & Martin studied T7 RNAP (a homolog of the mitochondrial RNAP studied here) and found that nicking the DNA strands (which would presumably relieve 'scrunching stress') did not have any effect on abortive initiation. Furthermore, this 'scrunching stress' hypothesis is eminently testable (Vahia & Martin tested it...).

Thank you for making these suggestions. We have now carried out the suggested transcription runoff assays on nicked and nucleotide-deleted DNA promoters to test the scrunching abortive synthesis model. The runoff assay results are shown in Fig. 2g and Supplementary Fig. 6, and discussed on page 7- 8. The results are very clear - nicking the NT strand between -1/+1 or +1/+2, the positions that are involved in NT loop formation in IC2 and IC3, drastically reduced 2-mer (20-fold) and 3-mer abortive (about 6-fold) compared to the intact promoter. The same effect is observed when the NT nucleotides +1 or +1+2 are deleted. Interestingly, a template nick had a drastic effect on transcription initiation and runoff synthesis. We speculate that the template nick prevents an efficient IC2 formation, but the mechanism is not known and will require further investigation.

These new biochemical experiments provide direct proof that NT strand scrunching in IC2 and IC3 create stressed intermediates that significantly contribute to the abortive synthesis. We have also included the Samanta and Martin, 2013 reference (Ref. 31).

3. The complexes for cryo-EM were prepared at 4°C and each incubation (with MTF1 and subsequently with the NTPs) was for several hours. The protein is apparently active at 4°C (since the different RNAs were synthesized by the RNAP), but still, I wonder if things are really the same at 4°C? Most importantly, is the pattern of abortive products the same at 4°C than at 25°C? In the cases where in vitro transcription results are shown (Figs. 2b, 3a, 4d, 4g, Supplementary Fig. 5), the experiments were all done at 25°C.

As suggested, we have carried out a side-by-side transcription walking experiment with the bubble promoter at 4°C for 2 h and 25°C for 15 min using the cryo-EM buffer. Supplementary Fig 1b shows the new results. The γ -mtRNAP is active at 4°C, and the assays show the same RNA synthesis pattern at the two temperature settings. We also see 2-mer synthesis with GTP alone at 4°C. Therefore, the 2xGTP bound IC2 observed in the cryo-EM experiment must be due to a combination of the faster rate of 2-mer RNA dissociation and the slower rate of the nucleotide incorporation.

Because we did not observe any noticeable difference in the RNA synthesis pattern at 4°C versus 25°C, we did not repeat the rest of the 25°C experiments at 4°C.

4. The authors use a variant of γ -mtRNAP where the 100 residues at the N-terminus are missing. However, the main text refers to this variant simply as γ -mtRNAP, and no further explanation is given in the text as to why this deletion was made. The referenced functional studies which form the basis for most of this study's conclusions seem to use the full-length protein, and deletion of the N-terminal sequence was shown to impact transcription initiation efficiency and the degree of abortive RNA synthesis (ref 46). The authors must explain how their conclusions are impacted by this deletion and take it into consideration when discussing the transitions of the IC- \rightarrow EC states.

The text now explicitly mentions that γ -mtRNAP used in the study lacks the N-terminal 100 aa in the first paragraph of the Result section. In ref 23, Patel's lab has demonstrated that the enzymatic properties of Δ 100 and full-length γ -mtRNAP are not very different. As correctly stated by the reviewer, Δ 100 makes relatively less abortives: Full-length protein undergoes \sim 18 abortive events per productive, and Δ 100 undergoes \sim 13. However, the abortive patterns of the two proteins are very similar, suggesting that 100 aa deletion does not significantly impact the overall mechanism of DNA scrunching, RNA elongation, or transition to EC.

We used Δ 100 variant because in our initial crystallography efforts DN100 successfully produced diffracting crystals suggesting that the construct is more stable for structural studies compared to the full-length γ -mtRNAP. The crystallography path was not successful due to severe twinning, and we switched to cryo-EM and continued our experiments with Δ 100 construct.

5. The authors observe a "spiral staircase" stacking of the scrunched NT strand in the IC5 structure. The scaffold used for IC5 consists only of adenine bases in the staircase region, while the staircase region in the scaffold used for IC2 and IC4 is disrupted by a T. Single-stranded homopolymers of deoxy-adenine are known to exhibit strong stacking preference, therefore naturally organizing into a spiral staircase. The authors conclude that stabilization of IC5 attained via the staircase facilitates processivity (lines 355f). However, as the authors argue in lines 265ff, this "staircase stabilization" of IC states may only hold true for certain sequences. How does the conclusion of the authors relate to natural mitochondrial promoter sequences?

The 11 yeast mitochondrial DNA promoters have an A-rich sequence between +1 to +8 NT bases (ref. 34, page 9, first paragraph). E.g., positions +1, +2, +4: 100% A, +3: 100% T, +5:

64% A, +6: 70% A, +7: 57% A, +8: 70% A. The occasional T's in the NT sequence could affect the stacking interactions. As stated in the manuscript (Page 9), substituting NT strand adenines between positions +3 to +8 with a string of thymine bases impairs runoff synthesis, whereas adding 6-12 successive thymines beyond the NT +12 position has little effect (ref. 36).

6. Also, obviously I don't expect the authors to do time-resolved cryo-EM, but they should address the issue that these structures are at equilibrium (at 4°C) and may not reflect the actual dynamics of initiation. For example, regarding the nt-strand 'spiral staircase' observed in IC5, what if the rate of formation of that structure is slow compared to the rate of aborting the RNA product or binding the next NTP and elongating to the next step? Then in the dynamic situation of initiation, that spiral staircase structure would never form and its properties would be irrelevant...

The reviewer raises interesting ideas, but as the reviewer also points out, these questions can be resolved only through future time-resolved structural studies. The NT staircase is a prominent element in all the structures after IC4. It is formed at IC5, and we now show that the structure is stable and expands in IC7 and IC8 states. The expanded NT staircase pushes MTF1 and dissociate it from the thumb in IC8. Hence, the NT staircase has multiple roles – it facilitates RNA synthesis after 4-mer by engaging the newly unwound NT base to the stack, and facilitates EC transition by breaking the thumb:MTF1 interactions in IC8.

7. There is also a lot of conjecture regarding the IC to EC transition (instead of actual structural information). On these constructs (pre-melted), the IC to EC transition is greatly disfavored at 25°C; Tang et al. JBC 2009 Fig 2 showed that the lengths and amounts of abortive vs productive products differ between transcribing from a duplex promoter or a pre-melted bubble (-4 to +2) construct (as used in this study). For the pre-melted promoter: 5- and 6-mers pile up versus very few from duplex promoter template (predominantly 2-mers and some 3-mers). Relatively little full length transcript is produced from a mismatched bubble, suggesting that the IC to EC conversion is greatly disfavored on this construct. Why even do this study on a pre-melted bubble?

In the revised manuscript, we have added two new structures including IC7 and IC8 that captured the conformational changes expected to occur for the IC to EC transition. Our single molecule FRET studies (Sohn et al Nat. comm. 2020; ref. 8 in the manuscript) indicated that EC transition occurs at 8-nt synthesis. Hence, the IC8 structure, which we could obtain by using a pre-melted promoter that cannot undergo bubble collapse, turned out to be a key design for capturing a transient intermediate state just prior to bubble collapse and promoter release. Fig 5 of the revised manuscript is dedicated to these structures.

Why study pre-melted bubbles? Thus far all the structures of transcribing complexes obtained with T7, bacterial, or Pol II system have been possible with bubble or fork promoters only. A potential problem of structural study with fully duplex promoter is the instability of the IC intermediates and trapping the various states at each nucleotide position, or multiple start sites. We and others are attempting structures with no bubble or a smaller bubble by cryo-EM, but experimental protocols are yet to be established.

8. Is the partially-melted initiation complex (PmIC) a significant intermediate on the initiation pathway? Previous studies done at 25°C at equilibrium adding ATP (can bind +1) does not change the size of the bubble (deduced is -4 to +2 based on 2-AP fluorescence). From this data, the authors deduced that NTP binding is not needed to form the full bubble. What evidence exists to support the PmIC being populated at room temperature or physiological temperatures with normal constructs (i.e. not pre-melted)?

The reviewer correctly points out that our previous 2AP studies reported fluorescence increase in NT bases at +1 and +2 in the absence of the initiating ATP. This can be explained by the similar conformations of the +1 and +2 NT bases in PmIC and IC2 structures. The 2AP fluorescence is dependent on base stacking interactions with neighboring bases. The major change during PmIC to IC2 transition occurs in the template strand that undergoes a large movement to align the +1 and +2 template bases with the NTPs at the active site. Accordingly, we reported that position -1 2AP undergoes a fluorescence change upon NTP binding.

The evidence for the existence of PmIC on duplexed promoter comes from our previous single-molecule FRET experiments (Sohn et al., Nature comm. 2020, ref. 8). Structurally, we show that the DNA in PmIC is less bent than in IC2. In single-molecule experiments, we labeled the duplex promoter with donor and acceptor fluorophores at positions +16 and -16. In the presence of γ -mtRNAP and MTF1, but in the absence of initiating NTPs, we observed a low FRET state at 0.135 and a midFRET state at 0.376 (see figure below). The low FRET state is consistent with a closed state and the midFRET state with the PmIC. Upon addition of initiating NTPs, the midFRET state dynamically transitions to a high FRET state at 0.563, consistent with the more bent DNA in IC2. The structure of PmIC revealed this previously uncharacterized state and explained the FRET result. The FRET continues to increase with the addition of nucleotides until +8 when it abruptly decreases due to unbending of DNA during EC transition. We paste Figure 1 from Sohn et al. below and highlight the indicated data in red. The transition to EC at IC8 state is indicated by the blue box.

Fig. 1 Transcription initiation occurs through dynamic conformational changes. **a** Single-molecule measurements of transcription initiation dynamics. The dual-labeled DNA template complexed with Rpo41 and Mtf1 was observed using a total internal reflection fluorescence microscope. **b** DNA templates used in base-pair-wise measurements of initiation complex dynamics. DNA template I could be stalled at positions +2, +3, +5, and +6, while DNA template II, which differed from DNA template I by four base-pairs (blue), could be stalled at positions +7 and +8. Both templates were labeled with Cy5 at position -16 of the non-template strand (magenta) and Cy3 at position +16 of template strand (green). The transcription promoter (underscored) and start site (arrow) are indicated. **c** FRET histograms from single-molecule traces with colocalized Cy3 and Cy5 signals at each stalling position. Histograms were fit to single, double, or triple Gaussian peaks (Supplementary Table 1). The brown, green, and magenta curves represent low, mid, and high FRET populations, respectively. Dashed vertical lines mark the major FRET peaks of DNA only, DNA + Rpo41/Mtf1, and the complex at position +7. Dashed lines in magenta mark the major FRET peaks at positions +2, +3, +5, and +6. **d** Representative smFRET traces at positions 0, +2, and +6 showing the Cy3 (green) and Cy5 (magenta) signals, and the FRET efficiency traces (navy). **e** The FRET level of the major population in **c** shown for each stalling position as the center of the major Gaussian peak. The error bars represent the error in finding the peak center position from Gaussian fitting. **f** The Cy3-Cy5 distance at each stalling position calculated as the average between those obtained from DNA templates I/II (**e**) and I/II NT (Supplementary Fig. 5). Error bars represent the propagation of the errors in FRET levels.

9. If the IC complexes were on pathway and were stable,
 i. wouldn't the authors have expected to see a mixture of intermediates (classes) in the IC4, IC5, IC6 mixes?

We clearly see preferred structural state for each IC, which moves to the next IC state upon the availability of the required nucleotide. Only IC3 had a significant population of PmIC in the sample. Also, we observed existence of two closely related conformations of the IC8 states. However, we do not rule out the existence of lower states associated with each IC state. If they

exist, then such potential states are not populated in significant amounts or not highly stable, and therefore, not detected in data processing.

ii. It is truly surprising that the IC2 population is not detected to be heterogeneous given the model and observation that release of 2-mers in vitro dominates the start of initiation. To support the arguments on lines 144-145, the authors need to perform transcription experiments to demonstrate that dimers form at 40C.

We were also surprised to see this result, given that in our previous IC3, we observed a mixture of PmlC and IC3, which we did not see in the IC2 cryo-EM experiment. As suggested, we show in Supplementary Fig. 1b that GTP addition under the cryo-EM experimental condition (2h, 4°C) produces 2-mer RNA just as we see in a parallel 15 min set at 25°C for comparison. Our structural and biochemical experimental show that the 2-mer RNA is produced, however, dissociated rapidly, and IC2 with two GTPs is structurally trapped presumably because the catalytic phosphodiester bond formation is rate limiting in the IC2 state.

iii. Note that in Fig 2b (cryo-EM construct), there are no 2-mers when GTP and UTP are present. Why?

Our parallel set of transcription reactions at 4°C and 25°C (in the new Supplementary Fig. 1b) show the presence of 2-mers when GTP and UTP are present.

iv. Fig 2b: when GTP, UTP and ATP are present, see 3-mers and 4-mers. In cryoEM (Fig 1a) experiment, only see a 4 mer (IC4). Why?

Indeed, the IC2 and IC3 states are highly abortive and could only be trapped prior to the nucleotide incorporation. The observed RNAs in the transcription experiments are most likely abortive RNAs and not the IC2 and IC3 stalled complexes. All remaining IC states with larger RNAs are stabilized with more interactions as observed in the cryo-EM structures. This could be the reason why a sizeable population of prior IC states were not observed co-existing with IC4 state. Also, please see our response to (i) above.

v. Why did the authors use pppGpG instead of GTP for the IC4-IC6 experiments?

Using pppGpG instead of GTP allowed us flexibility in terms of using a minimal number of scaffolds to capture maximum number of intermediate states. Use of pppGpG may prevent multiple start sites. However, we have successfully used GTP (instead of pppGpG) for forming IC8.

10. Only model with a bound Mg²⁺ is IC2 (the i+1 NTP is bound (triphosphate is still present). The authors say IC2 is catalytically poised for bond synthesis but they did not model a second Mg²⁺?

Two metal ions are involved in the catalytic addition of a nucleotide by a polymerase. The ion B chelates all three phosphates of an incoming NTP/dNTP, and the ion B is always present with an NTP/dNTP bound at the active site. In contrast, the ion A is engaged to complete the

reaction. Observing the ion B in structures is not unusual. For example, the structural studies of HIV-1 reverse transcriptase often observed the presence of only ion B and not A.

11. Re IC to EC conversion: How can the bubble collapse (e.g. rewind) when has been engineered to not rewind? Isn't that possibly why IC7 did not form?

We are now able to capture the IC7 and IC8 intermediate with a slightly modified sequence of the bubble promoter (Supplementary Fig 1a). It seems that non G:C sequence is required at the +7 position for the formation of IC7.

12. What do the authors mean on lines 323-326 that a 6-mer is the last stable state? When duplex promoter DNA is used, the only significant products formed are 2,3 mer and runoff (e.g. Fig. 4g), indicating that any RNA longer than a 3-mer is rapidly extended and that $n > 3$ products are only transiently populated.

We agree with the reviewer, we have removed this text.

13. Fig 2b, none of the lanes corresponds to the nucleotide combinations used in the cryo-EM experiments on this construct (Fig. 1a), nor the condition of the cryo-EM experiment (different buffer, 40C). Does a 12-mer represent a product from an elongation complex when there is no remaining downstream duplex DNA?

The polymerase runoff assay is repeated with cryo-EM buffer at 4°C and 25°C (now Supplementary Fig 1b) and with a non-bubble promoter (now Supplementary Fig 1c). The 12-mer is a runoff product when there is no remaining downstream duplex, but we cannot be sure whether it is from an elongation complex or an extended RNA:DNA hybrid. Our IC8 structure and EC model with 9-mer RNA:DNA (Fig. 5), however, suggest that the template should be switched at IC8/IC9 state to form the RNA exit channel.

Other issues

7. The authors show that variation of the -1 base identity severely affects transcription efficiency. What is the sequence conservation of the base and the interacting residues? How would efficient transcription be explained if a different base in that position occurs in other promoters?

The -1 template base is an adenine in all γ -mtRNAP promoters. The -1 template base changes its location in the polymerase cleft at each nucleotide addition step (Fig. 5a-b); hence, it is unlikely to have an adenine specific interaction; we have observed that a purine is important at this position.

8. In the IC 6 state, the -1 base inserts into a pocket at the thumb-domain. However, when considering the transition to the EC state later in the text (line 306ff), the -1 base capture is not mentioned although the preceding section highlights its importance through structural and biochemical evidence. What is the role of this pocket during the transition to EC?

These are good questions. We now have more information about IC to EC transition from the new IC7 and IC8 structures. Template scrunching affects the position of the -1 base the most. In IC8, the -1 base is poised to reanneal with the complementary NT base. Because we used a pre-melted promoter, the -1 NT:template bases were not paired, and thereby we were able to trap the IC8 intermediate just before the bubble collapse. The revised manuscript describes the new structures and the model for EC transition.

9. Fig 1a: indicated GTP for IC4 but should be ATP (in addition to pppGpG /UTP).

Thank you! This is corrected.

10. line 209: "relative amount of 3-mer abortive" is not defined and confusing (relative to other abortive products, relative to runoff?).

The 3-mer abortive from the C-tail mutant reaction is relative to the wild type MTF1. However, our discussion of C-tail is now shorten on Page 7 to "Previous single-molecule FRET and biochemical studies of γ -mtRNAP showed that partial deletion of MTF1 C-tail relaxes the crunched IC2 and reduces abortive synthesis²⁹".

11. In general, some of the figures are pretty difficult to digest...

A lot of the structural figures need an 'inset' of the overall structure to help orient the viewer to the zoomed-in views in the context of the entire complex.

Is the active site Mg²⁺ visible - I don't see any density figures that show that clearly

Fig. 1e - why not show where the active site Mg²⁺'s are

Fig. 1f - orientation is difficult to figure out

Fig. 2d is very difficult to digest

Figs. 3b, c, d, f, g - it would be helpful to show the RNA somehow (I understand all of them can't be shown because it would be way too busy, but perhaps show the longest RNA in each panel...

Fig. 4e - the important point here is what is happening with the -1 base, but this is obscured by the thick, densely colored backbone worms. One could make the backbone worms much thinner and a very light color.

Fig. 5c and 5d is supposed to illustrate the large change in the upstream DNA on the transition from IC6 to EC, but the view in 5C doesn't show the upstream DNA...

We have now modified the figures by taking the reviewers suggestions into account. Also, with new information from the IC7 and IC8 structure, we have added multiple new figures in Fig. 3 – 5.

12. Fig 4g: The underlined bases in the sequence are not defined in the figure legend.

We have now shortened the discussion on template -1 nucleotide, and the old Fig. 4g showing an elaborated study on the impacts template -1 modifications has been removed. However, the underline sequences in other figures are now defined in respective figure legends.

13. Fig 4g/Supplement Fig 5: How often has the experiment been repeated? If it was $n \geq 3$ it would be useful to add error bars to the quantifications shown in Supplement Fig 5 a-d.

The experiments were done in duplicate.

14. Fig 3a: Legend is missing information on the quantification shown on the right-hand side as well as some explanation on the sequence (what are the red bases? What does the underline mean?)

Now the figure is Supplementary Fig 1c, and we have removed the red letters and modified the figure legend.

15. The included movies are a little misleading because they show the NTPs flying in a way that would be impossible (i.e. through the protein). The authors should either show the NTPs coming in the way we know they do (this RNAP has an NTP entry tunnel), or at least don't show the NTPs flying in, just have them 'appear' in their binding sites

We appreciate the comment on the NTP reaching the active site. We have now modified the movie that avoids the flying of the NTPs.

16. Lines 461-464: total final concentrations of NTPs used should also be given. Would be helpful if the authors' converted 3 mg/mL of the PmlC to molar concentration as well.

3 mg/ml of the PmlC corresponds to $\sim 15 \mu\text{M}$ and has been added to the materials and methods subsection "Assembly and characterization of γ -mtRNAP IC2, IC4, IC5, IC6, IC7 and IC8", page 15.

17. How were the concentrations of the PmlCs determined (e.g. lines 453, 459)?

PmlC concentrations for storage and grid preparation were determined via NanoDrop™. The buffer without protein was used for the blank reading. This information is included in page 15.

18. Please give origins of the NTPs used? E.g. where did pppGpG come from?

The 2-mer RNA pppGpG was purchased from TriLink BioTechnologies. Now we have added this to the experimental section in page 15.

19. Incorrect or incomplete set of NTPs given in Fig 1a for IC4 and IC6 re NTPs added, please correct.

The scaffold sequences are moved to Supplementary Fig. 1 and the errors are corrected.

Referee #3 (Remarks to the Author):

This manuscript by Goovaerts and colleagues is an expansion of previous work by the same group on the structural characterization of the initiation complex of the yeast mitochondrial RNA polymerase. Whereas previously the authors had only been able to obtain two snapshots of the process, the current manuscript completes a detailed characterization of the first six nucleotide additions during initiation. Given that transition to elongation mode for this polymerase appears to take place after the eighth incorporation (based on previous work from some of the same authors), this study provides a very complete picture of the initiation process.

Although the general characteristics of the yeast mitochondrial transcription initiation process had already been established, including the fact that it involves template scrunching, the current manuscript is nevertheless a breakthrough in that it provides mechanistic understanding on how the mtRNAP is able to progressively accommodate more and more scrunched nucleotides as the initiation process proceeds. Furthermore, it reveals surprising structural differences between the complexes, including evidence of scrunching in the non-template strand in the early steps of the process, that provide some insight into the factors that determine the balance between abortive synthesis and further elongation. The authors have further extended the insight provided by their structures by carrying out modeling of the process of transition to the elongation phase, and the movies provided will ensure that this work will reach a wide audience.

In my opinion, the manuscript could be strengthened by polishing some of the illustrations and clarifying some of the wording. The four structures provide a clear picture of the dynamics of the scrunching process and reveal some interesting hints about the factors that result in abortive termination. However, these aspects of the manuscript seem fairly speculative in the absence of additional experimental evidence. Nevertheless, this is a body of work that will be of immense interest to the field.

We appreciate the reviewer for the comments and suggestions. Our new structures of IC7 and IC8, the modeled EC with 9-mer RNA:DNA duplex, and biochemical experiments with nicked DNA provide strong evidence for the proposed mechanism associated with abortive synthesis of 2- and 3-mer RNA and the IC to EC transition.

I propose the following points for consideration by the authors.

Major points

-The authors propose that the conformations of the non-template (NT) strand contribute to abortive synthesis. This is because they observe a different conformation of the non-template strand in the IC2 and IC3 complexes that is suggestive of scrunching, and that this

conformation disappears in IC4 and subsequent complexes. As the authors point out in the discussion, there is an apparent correlation of this conformation with a higher likelihood of abortive termination, but in my opinion the manuscript does not present any evidence that this conformation is in fact related to the biochemical behavior.

To support the importance of NT scrunching, the authors point out that it has been previously shown that deletions of the C-terminal tail of MTF1 result in a reduction in abortive products, arguing that this evidence supports their claim since the C-terminal tail is located in the proximity of the relevant NT nucleotides. However, deleting the last 4 nucleotides of this tail (D4 deletion) in MTF1 is sufficient to lead to a drastic decrease in abortive transcript production, and neither of these four residues appear to interact with the non-template strand in IC2. (In this respect, the authors might have noticed that E338, the last residue deleted in the A4 deletion, does appear to stabilize one of the non-templated nucleotides in IC3, and that the biochemical phenotype in Basu et al. might be consistent with this). Further complicating the issue, the NT conformations in IC2 and IC3 are not identical, and the density clearly indicates that there is some conformational variability even in IC2 (although not to the same extent as in the other structures). Moreover, in some cases, the difference in sequence (purine vs pyrimidine) might affect the specific conformation adopted by the NT strand, perhaps accounting for some of these differences. As things stand, this conclusion appears highly speculative. Perhaps the authors could strengthen their claim by providing additional evidence based on mutagenesis. Unfortunately given that the observed conformations differ, it does not appear that there is a single interaction that might account for the IC2 and IC3 conformations, but the authors might be able to gain some insight by characterizing the phenotype of additional side chain mutations. For instance, have the authors attempted to substitute the Arg533 side chain?

The concern about lacking evidence for NT scrunching in IC2 and IC3 contributing to abortive synthesis was brought up by reviewer 2 as well. The reviewer suggested introducing nicks to reduce stress and we have now carried out the suggested transcription runoff assays with nicked and nucleotide-deleted DNA promoters to test the scrunching abortive synthesis model. These results are shown in Figure 2g and Supplementary Fig. 6. The results are very clear - nicking the NT strand between -1/+1 or +1/+2, positions involved in NT loop formation in IC2 and IC3, drastically reduced 2-mer and 3-mer RNA abortives compared to the intact NT. The same effect is observed when the NT nucleotides +1 or +1+2 are deleted. Interestingly, a nick in the template between -1 and +1 drastically reduced transcription initiation and runoff synthesis.

We believe that these new biochemical experiments add strong support to our proposed model that NT strand scrunching in IC2 and IC3 creates stressed intermediates that carry out abortive synthesis.

- That the authors obtained exclusively the GTP-bound IC2 state is surprising. The explanation provided in the manuscript is convincing, but it would necessitate that catalysis

be orders of magnitude slower than nucleotide binding (which seems likely) and 2-mer dissociation (which is perhaps less likely). In their previous work, the absence of complexes containing the 2-mer but no nucleotide can be explained by a very fast rate of nucleotide binding. In the present work, the absence of 2-mer complexes would imply equally fast dissociation. Moreover, if catalysis is that slow, this seems perhaps at odds with the inability to observe nucleotide-containing intermediates in the subsequent steps. An alternative explanation is that the IC2 complex is not catalytically competent. It is not entirely clear from the methods if the purified PmlCs contain Mg, or if Mg is added together with the NTPs (does the buffer used for size-exclusion chromatography contain Mg?). Moreover, while elongation of the longer complexes can evidently take place at 4C, could that initial reaction be very inefficient at that temperature? Have the authors been able to observe initiation in reactions carried out at 4C in the conditions used to incubate their complexes?

We have now carried out transcription reactions at 4°C and 25°C using the cryo-EM buffer conditions that contains Mg²⁺ ions (Supplementary Fig. 1b) and these experiments show the mismatched 6-mer bubble promoter used in the structural study catalytically synthesizes RNAs, including the 2-mer. The only reasonable explanation for seeing the IC2 intermediate with two GTPs bound instead of 2-mer RNA is that the 2-mer dissociates quickly and PmlC is replaced with GTPs rapidly compared to the phosphodiester bond formation for IC2.

Minor points

-The first paragraph of the introduction focuses on DNA scrunching, but while it appears to provide a general introduction to the process, the authors mention details that are specific to particular systems (e.g. 8-12 nt to transition to elongation mode, which is the case for T7 RNAP, when the authors have demonstrated that this is not the case for yeast mtRNAP). I would suggest rewording this paragraph to either clarify that some of the statements are applicable to specific polymerases or to make the statements more general.

We have modified the statement to make it more general.

-“The transition from PmlC to IC2 occurs with a large conformational change that moves the template strand by ~22 Å” – This is not an informative statement unless the reference points are also indicated, and I find that it might be misleading. What moves by 22A? Most of the template strand atoms are in fact within 1A of their original position.

The distance between PmlC and IC2 at their N1 positions are ~2 Å, 27 Å, 22 Å respectively for -1, +1, and +2 template nucleotides. This information is now added in page 6. Once the template is aligned with the initiating nucleotides/RNA, the template follows a very similar track except scrunching to generate various forms of bulges of the single-strand part of the template (between -4 and -1 positions).

-In Figure 1a, it would be helpful to color-code the differences between the two substrates.

We have underlined the inserted or modified sequences in bubble-2 (compared to bubble-1) and in bubble-3 (compared to bubble-2) in Supplementary Fig. 1a.

-I wonder if the left image in Figure 2b could be made more clear.

We have new gels in Supplementary Fig. 1b, c that show the transcription assays on a bubble promoter and a regular promoter.

-Figures 2e and 2f are not very helpful to illustrate the points made in the text. Out of context, the shift from +3/+4 to +8/+9 can only be grasped from the annotations, which are no better than what is stated in the text. Moreover, no interactions are shown, so it is not easy to determine whether indeed there is a loss of DNA interactions.

Now the old Fig. 2e and 2f are Supplementary Fig. 7b-d that show the close proximity of F16/Y18 with different parts of downstream DNA in different structures. The F16A/Y18A double mutant MTF1 drops the runoff product to 30% of the wild-type MTF1. The MTF1 surface at F16/Y18 region non-specifically supports downstream DNA at different IC states, and the F16A/Y18A mutation impacts by altering the interacting surface rather than loss of specific interactions.

-It would be useful to provide a supplementary figure that clearly shows the conformation of all the atoms in the variable region of the NT strand in each of the complexes (including PmIN and IC3), and not just the backbone.

Now we have shown the conformations of NT strands in all 8 structures from PmIC to IC8 in Supplementary Fig. 5, and we agree that this figure is very informative which clearly shows the scrunching of NT strand in IC2 and IC3, and the stacked NT staircase structure in IC5 to IC8.

-It might make more sense to present the quantification of Figure 3a in the same figure and not in Figure 4. What are the dots in Figure 4d? Have these experiments been repeated? If not, they should be, and means +/- SE should be reported in the figure.

The information in old Figures 3a and 4d is now combined as a part of Supplementary Fig. 7. The experiments were done in duplicates; each dot represents an experiment. The runoff products are quantified with % error.

-Line 437, it should be "were", no "was".

Sorry for the typo. We have now corrected this.

-Line 224, "next", not "net".

Sorry for the typo.

Reviewer Reports on the First Revision:

Referees' comments:

Referee #1 (Remarks to the Author):

In their revised manuscript, Goovaerts et al. have largely addressed my previous comments and concerns. In addition, they provide new data in the form of two additional structures, which now result in a complete picture of transcription initiation in yeast mitochondria. In particular IC8 is an important addition, as it provides insights into how the transition from IC to EC is triggered. This is a big advancement for the field, as it provides unprecedented detail of the stepwise synthesis of RNA during transcription initiation in yeast mitochondria.

The authors now also provide a conceptual comparison to transcription initiation in other systems, such as bacterial and eukaryotic nuclear transcription. This comparison clearly highlights that in all these systems clashes between the growing RNA and protein elements eventually trigger loss of initiation factors and promoter escape, and that DNA scrunching may play a conceptually similar role in promoting abortive transcription. However, to me it still leaves a question mark on how universal the mechanisms observed here are. I think it should be clearly noted that the mechanisms of NT scrunching and progressive movements of the transcription bubble observed in this work are likely to be highly specific for the yeast mitochondrial system, as they involve extensive interactions with the mitochondria-specific initiation factor MTF1. Thus, it remains open whether conceptually similar NT scrunching will be observed in other systems. In order to highlight whether the NT scrunching observed here is likely to be a general mechanism of mitochondrial transcription initiation, a comparison of conservation between MTF1 and TFB2M would be helpful (see my final comment).

Overall, I think this manuscript has been much improved, but needs some more work on details (see below). In general, I would recommend the authors try to simplify and unify the figures a bit more. I find them rather hard to follow, because they are not consistent with regards to coloring (for example of DNA/RNA stands: TS is sometimes blue, sometimes pink, sometimes orange; likewise, protein elements are often colored throughout figures differently than in the scheme in Figure 1b). I think by simplifying and unifying the figures, the data could be depicted in a much more obvious way, which would help the reader grasp the points made much more easily.

Comments:

- Line 90: "was" should be "were".
- Line 154: Is this shift what is shown in Fig. 2e? Then it should be referenced here.
- Line 193: "Relatively" is a vague term – please quantify here.
- Line 201: From this description, it remains unclear to me why stress in the NT strand would lead the complex to revert to PmIC, instead of promoting it to progress to IC4 quickly?
- Lines 208-209: The authors should also quantify the total amount of run-off transcription compared in these different conditions, i.e. do the nicks / deletions impair overall transcription initiation?
- Fig 3a: I would suggest to indicate the active site by modeling metal A.
- Line 218: What's the evidence for this favoring? This assumption is logical, but speculative.
- Line 221 and Figure 3b: It is very confusing that the MTF1 C-tail is displayed here in exactly the same way as the nucleic acid backbone in Figure 2e,f and in this panel. I suggest to clearly differentiate the depiction of protein and DNA/RNA.
- Line 228: This base is occluded by another strand in the figure, and it's not clearly visible what "pre-unwound" means in this context. Is it still base-paired? Then the pairing base should be shown.
- Suppl. Fig 7d,e: The figures have low quality and are hard to see.
- Line 280: The highly scrunched what? This sentence seems to be missing something.

- Line 282: C-tail not shown in referenced figure panel.
- Line 307: What makes this shift significant?
- Lines 312-327: In this paragraph, the authors discuss the conformations of the fingers domain and RNA 3' end, and compare IC8 to the human mtRNAP EC structure. For the human mtRNAP EC, two states of RNA and fingers domain have been observed: pre-translocated (PDB 4BOC) and post-translocated (PDB 5OLA). From this paragraph, it is not clear how the states observed by the authors relate to these known conformations, which are typical intermediates for RNAPs. Do the authors observe novel conformations, or are the states observed alterations of known pre- and post-translocated conformations?
- Lines 329-344: This paragraph discusses the rearrangements that occur between IC and EC. While the authors now correctly cite the previous work on the human mtRNAP IC (ref 20), I think it should be clearly stated that essentially all of the observations described in this paragraph have previously been described for the human system. In my opinion, the comparison of IC8 and EC9 of γ -mtRNAP described here is largely confirmatory of what has been previously proposed regarding the IC to EC transition in the human system, and does not provide major new insights into this part of mitochondrial transcription initiation.
- Line 378-379: What makes the authors speculate that γ -ins and TFAM have similar roles? Is this based on structural data presented here? To my understanding, γ -ins is involved in downstream DNA interactions, while TFAM has been shown to bind to the upstream promoter and tether h-mtRNAP (ref 20).
- Lines 381-383: The authors should mention that another open question regarding initiation by γ -mtRNAP is the role of the NTE, which in the human system forms a PPR domain and has been shown to interact with TFAM. Predictions show that the γ -mtRNAP may also form a PPR fold, and the Patel lab has shown previously that residues that are not visible in any of the currently available structures are necessary for promoter-specific initiation (ref 23).
- Line 373-383: The detailed structural insights obtained in this work on the interactions between MTF1 and the NT strand would possibly allow the authors to draw further comparisons to the human system: Are the residues in MTF1 that form stacking and base-specific interactions conserved in TFB2M? If so, a strong argument could be made that scrunching proceeds by a similar mechanism in the human system.

Referee #2 (Remarks to the Author):

The authors have submitted an improved manuscript:

- crucial new structures (IC7 and IC8)
- new biochemical data

But issues raised in the previous review have not been addressed:

- Overall, the figures are still very difficult to digest. Insets of the overall structure need to be provided (in the same orientation as the zoom-in, with the region of the zoom-in indicated). 'Standard orientations' need to be defined so that the viewer understands what they are looking at. For some figures, only extreme zoom-ins are provided without context of the overall structure (Figs. 3b-f, 4a-c, pretty much all the views of Fig. 5, many supplemental Figs.). The description of Fig. 3a in the text focuses on what is happening at the downstream fork, i.e. the premature melting of +6, but this is shown at the far right edge of the figure and much of the relevant structure is cut off. Some Figs. provide a title that explains which structures are being viewed (for example, Fig. 2d 'IC3 -> IC4'), while other figures do not (Fig. 2c, 3c, 3d, 3f, 4b, 4c, 4e, 4f). The Figs. have very complex color-coding; sometimes the color-key is given (for example, Fig. 2f), many figures an overall color-key is not given (Fig. 3e, 3f, all panels of Fig. 4 and 5). Some Figs. seem to be redundant, for example, Fig. 4e shows the opening of the polymerase cleft/beginning

of Mtf1 detachment, but this is already seen much more clearly in Figs. 3g, 3h. Fig. 5f is still confusing - since the DNA is not shown in the IC view, it is not possible to follow the conformational reposition of the DNA. It may be necessary to provide a simple schematic/model summarizing the mechanism deduced from the finding of this manuscript, otherwise it is extremely difficult for a reader to come to an overall understanding of the mechanism described here.

- Lines 90-91: Although it is explained in the rebuttal, it is not explained in the manuscript text why the DeltaN100 deletion is suitable for the present study.
- The description of the cryoEM data processing (lines 506-515) is still poorly documented and lacks crucial details. For example: How many classes were used for the 3D classification? Were any masks used? What were the input models for reconstruction and where did they come from (ab initio/template). The FSC graphs and local resolution map depictions in the cryoEM pipelines, specifically the font size, is too small to read. The FSC plots are missing a legend to identify the differently colored curves. Without being able to tell what is blue and what is red in the angular distribution depictions, it seems they show notable biases. The authors should include a legend, calculate 3D-FSCs (<https://3dfsc.salk.edu/>) and at least comment on the impact of these observations for their structural interpretations.

- New biochemical experiments (Figs. 2g and Supp Fig. 6) support that authors argument that nt-strand scrunching creates stressed intermediates that significantly contribute to abortive synthesis. This finding needs to be placed in context of previous work, however: As mentioned in the previous review, the experiments testing the 'scrunching stress' idea was first described by Vahia & Martin (Biochemistry, 2011) with T7 RNAP (this reference needs to be included, more importantly than the Samanta & Martin reference which did these experiments on the unrelated E. coli RNAP). The results of Vahia & Martin showed that knicking the template-strand upstream of the RNA/DNA hybrid did not have the expected effects on abortive initiation, contrary to the authors results with the mt-RNAP (Fig. 2g and Supp Fig. 6). The previous findings of Vahia & Martin need to be discussed, and the seeming contradiction needs to be addressed.

- In the previous review, we made the point that these structures are at equilibrium at 4°C and may not reflect the actual dynamics of initiation. There are prominent structural characteristics that could arise due to this issue (Nt-strand staircase, pre-melting of +6 in IC5...) and others are also possible. As mentioned previously, we do not expect the authors to address this experimentally at this time, but this issue was not addressed at all in the revised manuscript.

Referee #3 (Remarks to the Author):

The authors have addressed all my concerns with new experimental data and modifications to the manuscript. The new figures are quite helpful and I believe that the manuscript has substantially improved.

Author Rebuttals to First Revision:

Response to reviewers

We thank the reviewers for their helpful comments. We have now addressed all the concerns raised by the reviewers to improve the quality of the manuscript. The figures are now updated by taking the suggestions from Reviewers 1 and 2.

Referees' comments:

Referee #1 (Remarks to the Author):

In their revised manuscript, Goovaerts et al. have largely addressed my previous comments and concerns. In addition, they provide new data in the form of two additional structures, which now result in a complete picture of transcription initiation in yeast mitochondria. In particular IC8 is an important addition, as it provides insights into how the transition from IC to EC is triggered. This is a big advancement for the field, as it provides unprecedented detail of the stepwise synthesis of RNA during transcription initiation in yeast mitochondria.

The authors now also provide a conceptual comparison to transcription initiation in other systems, such as bacterial and eukaryotic nuclear transcription. This comparison clearly highlights that in all these systems clashes between the growing RNA and protein elements eventually trigger loss of initiation factors and promoter escape, and that DNA scrunching may play a conceptually similar role in promoting abortive transcription. However, to me it still leaves a question mark on how universal the mechanisms observed here are. I think it should be clearly noted that the mechanisms of NT scrunching and progressive movements of the transcription bubble observed in this work are likely to be highly specific for the yeast mitochondrial system, as they involve extensive interactions with the mitochondria-specific initiation factor MTF1. Thus, it remains open whether conceptually similar NT scrunching will be observed in other systems. In order to highlight whether the NT scrunching observed here is likely to be a general mechanism of mitochondrial transcription initiation, a comparison of conservation between MTF1 and TFB2M would be helpful (see my final comment).

Overall, I think this manuscript has been much improved, but needs some more work on details (see below). In general, I would recommend the authors try to simplify and unify the figures a bit more. I find them rather hard to follow, because they are not consistent with regards to coloring (for example of DNA/RNA stands: TS is sometimes blue, sometimes pink, sometimes orange; likewise, protein elements are often colored throughout figures differently than in the scheme in Figure 1b). I think by simplifying and unifying the figures, the data could be depicted in a much more obvious way, which would help the reader grasp the points made much more easily.

The reviewer asks us to address the generality of the observed conformations of γ -mtRNAP ICs to other RNAPs and h-mtRNAP. As suggested by the reviewer, we have added the following texts on pages 11-12:

Page 11 – “DNA scrunching has been observed during abortive synthesis in the bacterial RNAP system by single molecule studies³⁰; however, the structural snapshots have not been captured. Hence, it remains to be determined if the specific IC states observed in γ -mtRNAP have functional and/or conformational analogs in other RNAP systems.”

Page 11 – “As many key cellular processes evolve from simpler single-cell species to multicellular systems with added complexity, we expect a similar evolution process for transcription initiation; however, this has not been explored in detail.”

Page 12 – The Y103-X-W105 region of MTF1, essential for promoter melting in yeast¹⁹, is not conserved in TFB2M, in which the corresponding region has a large loop structure (residue L154 to S168)²⁰. This

difference suggests that the homologous factors in *y*-mtRNAP and *h*-mtRNAP may use distinct promoter melting mechanisms.

The Figures are simplified, and the coloring is now consistent.

Comments:

- Line 90: “was” should be “were”.

corrected

- Line 154: Is this shift what is shown in Fig. 2e? Then it should be referenced here.

Yes. Now it is Fig. 2c and referred to in the text.

- Line 193: “Relatively” is a vague term – please quantify here.

The sentence “The B-factor 126 \AA^2 of the NT region is $\sim 1.6x$ of the average (80 \AA^2) for the structure; IC3 has $\sim 0.9x$ average B-factor for the region.” is now added to paragraph 2, page 7.

- Line 201: From this description, it remains unclear to me why stress in the NT strand would lead the complex to revert to PmIC, instead of promoting it to progress to IC4 quickly?

We hypothesize that NT scrunching in IC2 and IC3 states and fewer interactions of short 2- and 3-mer RNAs increase the probability of aborting the short RNAs in early transcription steps, as observed in runoff assays (Extended Data Fig. 1C). We have added the following sentence at the end of Paragraph 2, page 6.

“Additionally, fewer interactions of short RNAs with *y*-mtRNAP and template make the IC2/IC3 states less stable, increasing the probability of dissociating the short RNAs.”

- Lines 208-209: The authors should also quantify the total amount of run-off transcription compared in these different conditions, i.e. do the nicks / deletions impair overall transcription initiation?

The following panel is added as Fig. 2h.

- Fig 3a: I would suggest to indicate the active site by modeling metal A.

The metal A is added, and Fig. 3a is modified.

- Line 218: What's the evidence for this favoring? This assumption is logical, but speculative.

For clarity, we have modified the sentence on page 7 (last paragraph):

“Thereby, IC5→IC6 transition would require less energy to flip the unpaired +6 template nucleotide into the polymerase cleft, position the next NTP, and form 6-bp RNA:DNA.”

- Line 221 and Figure 3b: It is very confusing that the MTF1 C-tail is displayed here in exactly the same way as the nucleic acid backbone in Figure 2e,f and in this panel. I suggest to clearly differentiate the depiction of protein and DNA/RNA. Also, Fig.

We have added new Extended Data Fig. 6 showing the C-tail conformations in 8 individual structures. Also, the new Fig.3c is modified for clarity.

- Line 228: This base is occluded by another strand in the figure, and it's not clearly visible what “pre-unwound” means in this context. Is it still base-paired? Then the pairing base should be shown.

We have added a new Fig. 3b showing the pre-melted +6 base-pair in IC5.

- Suppl. Fig 7d,e: The figures have low quality and are hard to see.

The figures are modified in Extended Data Figure 7.

- Line 280: The highly scrunched what? This sentence seems to be missing something.

Corrected to “Severely scrunched template in IC8 repositions the γ -mtRNAP N-terminal 519–528 region, located between the thumb and the intercalating hairpin, by $\sim 8 \text{ \AA}$.” in page 9 (paragraph 1).

- Line 282: C-tail not shown in referenced figure panel.

The C-tail in IC7 and IC8 are compared in Fig. 4e. Corrected.

- Line 307: What makes this shift significant?

For clarity, we have modified and reorganized the section in the last paragraph, Page 9.

“During the entire initiation process, template scrunching shifts the bulging -1 nucleotide to new locations in the polymerase cleft (Fig. 5a-c). The template -1 base stacks with the RNA:DNA duplex in IC2 and IC3, presumably to stabilize the +1 base-pair for initial RNA synthesis. In later ICs, the -1 base unstacks and shifts towards the thumb. Between IC2 and IC6, the template -1 nucleotide shifts $\sim 21 \text{ \AA}$ along the direction of the extending RNA:DNA (Fig. 5b), and between IC6 and IC8, the -1 base shifts $\sim 19 \text{ \AA}$ in a direction $\sim 95^\circ$ to the initial shift (Fig. 5c).”

- Lines 312-327: In this paragraph, the authors discuss the conformations of the fingers domain and RNA 3' end, and compare IC8 to the human mtRNAP EC structure. For the human mtRNAP EC, two states of RNA and fingers domain have been observed: pre-translocated (PDB 4BOC) and post-translocated (PDB 5OLA). From this paragraph, it is not clear how the states observed by the authors relate to these known conformations, which are typical intermediates for RNAPs. Do the authors observe novel conformations, or are the states observed alterations of known pre- and post-translocated conformations?

A kink between the RNA:DNA and downstream DNA exists in all single-subunit RNAP structures, including 4BOC and 5OLA, irrespective of whether the 3'-end of RNA occupies the N- or P-site.

Unexpectedly, this region is stretched in the γ -mtRNA IC8 structure, presumably because it represents a transient state. We have now modified the section for clarity on page 10 (paragraph 1).

“A template kink between the RNA:DNA and downstream DNA duplexes exists in all T7 RNAP, h-mtRNAP, and γ -mtRNAP IC structures but in IC8, where this region is stretched (Fig. 5d). Comparison of γ -mtRNAP IC8 with h-mtRNAP EC (Extended Data Fig. 9c) shows that the characteristic template kink is present before and after IC8 state. Hence, the unusually stretched template in the transient IC8 could be essential for upstream bubble collapse.”

- Lines 329-344: This paragraph discusses the rearrangements that occur between IC and EC. While the authors now correctly cite the previous work on the human mtRNAP IC (ref 20), I think it should be clearly stated that essentially all of the observations described in this paragraph have previously been described for the human system. In my opinion, the comparison of IC8 and EC9 of γ -mtRNAP described here is largely confirmatory of what has been previously proposed regarding the IC to EC transition in the human system, and does not provide major new insights into this part of mitochondrial transcription initiation.

We agree that the previous studies compared structures of early ICs with the EC to deduce the major conformational changes accompanying the transition, like the upstream promoter relocation. Here, by solving the structures of each intermediate up to the point where the transition occurs at IC8, we have defined the path leading to bubble collapse and MTF1 release that enables the major conformational changes to occur.

In response to this comment, we have modified the last paragraph on page 10 as:

“Previous studies of single-subunit RNAPs compared structures of EC with early ICs to deduce conformational changes accompanying IC to EC transition, such as the reorganization of the upstream DNA to a new location for the formation of an RNA-exit channel^{8,24,29,39}. By solving the structures of each intermediate up to the point where the transition occurs at IC8, we have defined a detailed path leading to bubble collapse and MTF1 release that enables visualizing the specific conformational changes. The IC8 shows how (i) the highly scrunched template and expanded NT push the thumb and MTF1 away from each other, (ii) the positioning of MTF1 hairpin and intercalating hairpin are destabilized by scrunched template, (iii) removal of C-tail out of the polymerase cleft, and (iv) the repositioning of template -1 to -3 bases adjacent to the complementing NT strand would lead to bubble collapse and IC to EC transition.”

- Line 378-379: What makes the authors speculate that γ -ins and TFAM have similar roles? Is this based on structural data presented here? To my understanding, γ -ins is involved in downstream DNA interactions, while TFAM has been shown to bind to the upstream promoter and tether h-mtRNAP (ref 20).

Although γ -ins stabilizes downstream DNA and TFAM upstream, both stabilize the IC states. We have modified the sentence on page 12:

“The γ -ins domain that is essential for transcription initiation by γ -mtRNAP is absent in h-mtRNAP, which recruits TFAM for initiation. Perhaps some of the tasks of γ -ins are delegated to TFAM as both stabilize the promoter.”

- Lines 381-383: The authors should mention that another open question regarding initiation by γ -mtRNAP is the role of the NTE, which in the human system forms a PPR domain and has been shown to interact with TFAM. Predictions show that the γ -mtRNAP may also form a PPR fold, and the Patel lab has shown previously that residues that are not visible in any of the currently available structures are necessary for promoter-specific initiation (ref 23).

As suggested, we have added the information in paragraph 1, page 4.

“The experimental density maps resolved the transcription bubbles (Extended Data Fig. 2) and surroundings. The N-terminal extension, which plays a regulatory role in y-mtRNAP²³, is disordered. This region in h-mtRNAP contains the tether-helix and PPR domain, which interacts with upstream DNA²⁰.”

- Line 373-383: The detailed structural insights obtained in this work on the interactions between MTF1 and the NT strand would possibly allow the authors to draw further comparisons to the human system: Are the residues in MTF1 that form stacking and base-specific interactions conserved in TFB2M? If so, a strong argument could be made that scrunching proceeds by a similar mechanism in the human system.

We have added the following text deduced from the comparison of MTF1 and TFB2M in paragraph 1, page 12.

“The Y103-X-W105 region of MTF1, essential for promoter melting in yeast¹⁹, is not conserved in TFB2M, in which the corresponding region has a large loop structure (residue L154 to S168)²⁰. This difference suggests that the homologous factors in y-mtRNAP and h-mtRNAP ICs may use distinct promoter melting mechanisms.”

Referee #2 (Remarks to the Author):

The authors have submitted an improved manuscript:

- crucial new structures (IC7 and IC8)
- new biochemical data

But issues raised in the previous review have not been addressed:

- Overall, the figures are still very difficult to digest. Insets of the overall structure need to be provided (in the same orientation as the zoom-in, with the region of the zoom-in indicated. 'Standard orientations' need to be defined so that the viewer understands what they are looking at. For some figures, only extreme zoom-ins are provided without context of the overall structure (Figs. 3b-f, 4a-c, pretty much all the views of Fig. 5, many supplemental Figs.).

We have carefully evaluated the comments to improve the figures. Now multiple overall structures are shown in Fig. 3f, Fig. 5a, and Fig. 5e to view the perspective of zoomed-in figures.

The description of Fig. 3a in the text focuses on what is happening at the downstream fork, i.e. the premature melting of +6, but this is shown at the far right edge of the figure and much of the relevant structure is cut off.

We have added a new Fig. 3b, clearly showing the melted +6 base pair.

Some Figs. provide a title that explains which structures are being viewed (for example, Fig. 2d 'IC3 -> IC4'), while other figures do not (Fig. 2c, 3c, 3d, 3f, 4b, 4c, 4e, 4f).

Now we have labeled all figures either by specific IC state or transition.

The Figs. have very complex color-coding; sometimes the color-key is given (for example, Fig. 2f), many figures an overall color-key is not given (Fig. 3e, 3f, all panels of Fig. 4 and 5).

Color keys are given in Fig. 2 and Fig. 5. Earlier Fig. 3b is now substituted with new simplified Fig. 3c and expanded in Extended Data Fig. 6.

Some Figs. seem to be redundant, for example, Fig. 4e shows the opening of the polymerase cleft/beginning of Mtf1 detachment, but this is already seen much more clearly in Figs. 3g, 3h.

Figures are rearranged. Figs 4c – g panels show key structural changes from IC7 to IC8. Old Fig. 4d is dropped. New Fig. 4d provides a quantitative analysis of the relative movements of MTF1, Thumb, and MTF1 hairpin from IC7 to IC8.

Fig. 5f is still confusing - since the DNA is not shown in the IC view, it is not possible to follow the conformational reposition of the DNA.

The overall views of IC8 and EC9 complexes are added as Fig. 5e and the regions zoomed in Fig. 5f are marked to avoid the indicated confusion. Also, the thumb and MTF1 hairpin are color coded.

It may be necessary to provide a simple schematic/model summarizing the mechanism deduced from the finding of this manuscript, otherwise it is extremely difficult for a reader to come to an overall understanding of the mechanism described here.

We agree that the paper has many new details from comparing 8 IC structures and the EC model. We show this in Supplementary Video 1 (for RNA/DNA) and 2 (for proteins). Also, key features of the structures are compared side-by-side in Fig. 1, and specific features are compared in Extended Data Figs. 2, 3, 5, and 6.

- Lines 90-91: Although it is explained in the rebuttal, it is not explained in the manuscript text why the DeltaN100 deletion is suitable for the present study.

We have added the following line on page 4.

“The Δ N100 was chosen because it is more stable for structural studies than full-length protein and fully active for transcription initiation²³.”

- The description of the cryoEM data processing (lines 506-515) is still poorly documented and lacks crucial details. For example: How many classes were used for the 3D classification? Were any masks used? What were the input models for reconstruction and where did they come from (ab initio/template). The FSC graphs and local resolution map depictions in the cryoEM pipelines, specifically the font size, is too small to read. The FSC plots are missing a legend to identify the differently colored curves. Without being able to tell what is blue and what is red in the angular distribution depictions, it seems they show notable biases. The authors should include a legend, calculate 3D-FSCs (<https://3dfsc.salk.edu/>) and at least comment on the impact of these observations for their structural interpretations.

We have added the following text to the “Cryo-EM data processing” section (page 23).

“The previously obtained density map for IC3 (EMDB EMD-10846/PDB 6YMW) was blurred to 40Å resolution and used as the template for initial 3D classifications. The homogenous particle sets from the best 3D class were used to calculate auto-refined maps and corresponding masks. For each structure, the particles were repicked at the 3D auto-refined positions and classified further (Supplementary Figs. 5-6) within its mask.”

We have also calculated 3D-FSC curves for all six structures using the half maps on the https://3dfsc.salk.edu/_server. In fact, the calculated resolutions are equal or marginally better than that we obtained from Relion processing. The resolutions are 3.47 Å (3.47 Å), 3.41 Å (3.44 Å), 3.32 Å (3.39 Å), 3.5 Å (3.62 Å), 3.6 Å (3.75 Å), and 3.6 Å (3.62 Å) for IC2, IC4, IC5, IC6, IC7, and IC8 maps, respectively; our reported values are in parenthesis. The 3D-FSC calculated plots are now added to the Supplementary Data file.

- New biochemical experiments (Figs. 2g and Supp Fig. 6) support that authors argument that nt-strand scrunching creates stressed intermediates that significantly contribute to abortive synthesis. This finding needs to be placed in context of previous work, however: As mentioned in the previous review, the experiments testing the 'scrunching stress' idea was first described by Vahia & Martin (Biochemistry, 2011) with T7 RNAP (this reference needs to be included, more importantly than the Samanta & Martin reference which did these experiments on the unrelated E. coli RNAP). The results of Vahia & Martin showed that knicking the template-strand upstream of the RNA/DNA hybrid did not have the expected effects on abortive initiation, contrary to the authors results with the mt-RNAP (Fig. 2g and Supp Fig. 6). The previous findings of Vahia & Martin need to be discussed, and the seeming contradiction needs to be addressed.

We have added the indicated T7 RNAP study and addressed the differences in paragraph one on page 7.

“Interestingly, a nick in the template strand between -2/-1 impairs all RNA synthesis. In contrast, a nicked template in the T7 promoter showed little impact on transcription initiation³². The different outcomes in γ -mtRNAP vs. T7 RNAP, which does not involve an initiation factor, may be due to differences in their promoter opening mechanisms.”

- In the previous review, we made the point that these structures are at equilibrium at 4°C and may not reflect the actual dynamics of initiation. There are prominent structural characteristics that could arise due to this issue (Nt-strand staircase, pre-melting of +6 in IC5...) and others are also possible. As mentioned previously, we do not expect the authors to address this experimentally at this time, but this issue was not addressed at all in the revised manuscript.

Four structures (IC6 – IC8) show the existence of the NT platform, providing convincing evidence that such a structure is formed during transcription. The lower temperature may slow down the dynamics, but the impact of the 4°C temperature (experimental condition) on the conformational states observed is likely to be minimal, if any. This is evident from the identical pattern of RNA synthesis seen in the transcription assays conducted at 25°C and 4°C.

We have now added a line in page 11 to point out this limitation.

“The dynamics associated with the discrete structural states and the transitions can be explored by time-resolved and molecular dynamics simulation studies.”

Reviewer Reports on the Second Revision:

Referees' comments:

Referee #1 (Remarks to the Author):

The authors here present a much-improved manuscript. In terms of scientific content, this manuscript is now acceptable for publication. However, I still find most figures very difficult to understand and not intuitive. Some examples:

- Non-intuitive arrangement of figure panels (see comment about Figure 2e)
- Non-consistent coloring: In Figure 2c, structures (IC2,3,4 etc.) are represented by different colors, while in Figure 2g the different structures are represented by different transparency.
- Fig. 2g: The arrow is very hard to see.
- Fig. 3c: Again, this figure is very hard to understand intuitively, as it is not indicated which coloring is IC4, which is IC5, what is protein, what is nucleic acid (they are both represented in the same style).
- Fig. 3d: For consistency, I would suggest to also show the density around Y103 – is it should be equally well-resolved as the NT strand?

These are just some examples. In general, most figures are virtually uninterpretable without careful study of their legends. I understand that it is very difficult to represent this many structures in just a few figures, but I think it would be worthwhile closely rethinking how the structures are presented and how a consistent color code and depiction could be used. I am convinced this would make the work much more appealing to many readers and would therefore also increase its impact.

Minor comments:

- Line 104: The hairpin is not the homolog. I would suggest to re-phrase to “the corresponding feature”. I find this comparison confusing, as in the cited paper of the h-mtRNAP, this loop is termed “lever loop” and not “B2 hairpin”.
- Figure 2e: I find the positioning of the schematic showing the nicked template confusing, as it is not positioned within panel e but instead appears to be part of panel d on first sight. I would suggest to organize this figure a bit differently so it is more intuitive.

Supplementary data:

SI Figure 5 & 6: The FSC plots are barely readable, and there is no legend showing which color line represents what data. Can the authors comment/explain on the dip in the FSC plots of some structures for the black curves?

Referee #2 (Remarks to the Author):

Unfortunately, this manuscript STILL has a totally inadequate description of the cryo-EM processing pipelines. In this 2nd revision, the Methods haven't truly improved. To name several issues specifically:

- Most importantly, settings/parameters for the steps (at least the important ones) aren't described anywhere (e.g., was the 3D classification performed with or without alignment?) what was the box size for particle extraction?
- it's not clear what's meant by 'blurred' - this could be low-pass filtering or a simple gaussian filter as Chimera implements it; needs to be clarified

- how do they define 'best 3D class' ? based on what property?
- 'improved by B polishing' - probably means Bayesian polishing?
- how did they calculate the local resolution maps, the orientation plots?

Author Rebuttals to Second Revision:

Referees' comments:

Referee #1 (Remarks to the Author):

The authors here present a much-improved manuscript. In terms of scientific content, this manuscript is now acceptable for publication. However, I still find most figures very difficult to understand and not intuitive. Some examples:

- Non-intuitive arrangement of figure panels (see comment about Figure 2e)

We appreciate the reviewer's concern on clarity of the figures. Now we have replaced Figure panels with new ones to maintain color consistency and clarity. Each figure is now focused on individual themes. Extended Data Figures also comply the common color scheme.

- Non-consistent coloring: In Figure 2c, structures (IC2,3,4 etc.) are represented by different colors, while in Figure 2g the different structures are represented by different transparency.

Fig. 2 is now focusing on abortive synthesis. The pol active site represented in old Fig. 2g has been removed. A new set of cartoons explaining the active site changes in all 8 structures is now added as Fig. 5a.

- Fig. 2g: The arrow is very hard to see.

This figure is now replaced as a part of the simplified cartoon in Fig. 5a.

- Fig. 3c: Again, this figure is very hard to understand intuitively, as it is not indicated which coloring is IC4, which is IC5, what is protein, what is nucleic acid (they are both represented in the same style).

Figure panels representing individual structures from IC2 to IC8 are color coded as protein parts (MTF1, yellow; thumb, green) and nucleic acid parts (green, RNA; blue NT; pink template). In figures comparing two structures, the leading structures follow the color scheme, and the trailing structures are in gray/light blue colors for contrast.

In new Fig 3c, the C-tail is shown as thicker ribbons and a part of MTF1 connecting C-tail is also displayed to avoid any confusion between protein and nucleic acid chains. Further, the NT bases are now shown as sticks.

- Fig. 3d: For consistency, I would suggest to also show the density around Y103 – is it should be equally well-resolved as the NT strand?

The density for Y103 is now shown in Fig. 3d.

These are just some examples. In general, most figures are virtually uninterpretable without careful study of their legends. I understand that it is very difficult to represent this many structures in just a few figures, but I think it would be worthwhile closely rethinking how the structures are presented and how a consistent color code and depiction could be used. I am convinced this would make the work much more appealing to many readers and would therefore also increase its impact.

Now we have introduced color labels in the figures comparing two structures for easy identification. For example, IC8 thumb is leveled in green and IC7 thumb is in gray in Fig 4c.

Minor comments:

- Line 104: The hairpin is not the homolog. I would suggest to re-phrase to “the corresponding feature”. I find this comparison confusing, as in the cited paper of the h-mtRNAP, this loop is termed “lever loop” and not “B2 hairpin”.

We have now referred this also as lever loop; however, due to the structural and functional analogy with MTF1 loop (in the yeast system), we also refer as B2 loop.

- Figure 2e: I find the positioning of the schematic showing the nicked template confusing, as it is not positioned within panel e but instead appears to be part of panel d on first sight. I would suggest to organize this figure a bit differently so it is more intuitive.

The Figure 2e is now aligned to avoid any confusion.

Supplementary data:

SI Figure 5 & 6: The FSC plots are barely readable, and there is no legend showing which color line represents what data. Can the authors comment/explain on the dip in the FSC plots of some structures for the black curves?

Corrected. We are sorry that we missed this correction in the previous version. The dip could be due to the fact that a comparatively a smaller number of data is available for the resolution range at the dip.

Referee #2 (Remarks to the Author):

Unfortunately, this manuscript STILL has a totally inadequate description of the cryo-EM processing pipelines. In this 2nd revision, the Methods haven't truly improved. To name several issues specifically:

- Most importantly, settings/parameters for the steps (at least the important ones) aren't described anywhere (e.g., was the 3D classification performed with or without alignment?) what was the box size for particle extraction?

We have now added following details to the data processing section in page 22.

The 3D classifications eliminated partially disordered and incomplete particles. The particles forming lower-resolution density maps (Supplementary Fig. 2) were eliminated by manually visualizing in Chimera. The lower resolution maps for an IC state did not define a conformation distinct from the accepted one. However, two distinct classes, IC8 and IC8', were detected in 3D classification stage and were separated as two independent datasets for further refinement.

The particles were extracted with box size of 256^3 pixels for IC2 and 192^3 for the remaining datasets.

- it's not clear what's meant by 'blurred' - this could be low-pass filtering or a simple gaussian filter as Chimera implements it; needs to be clarified

We are sorry for the confusion. We have now clarified (as discussed in response to the previous comment and shown in Supplementary Fig. 2) and used the term lower-resolution rather than blurrier map.

- how do they define 'best 3D class' ? based on what property?

The map quality in 3D stages were evaluated by visual assessment. Distinct high resolution map quality compared to remaining 3D classes.

- 'improved by B polishing' - probably means Bayesian polishing?

Expanded to Bayesian polishing.

- how did they calculate the local resolution maps, the orientation plots?

The local resolution maps were calculated using ResMap and the orientation plots were generated by Relion 3.1. This sentence is added to Cryo-EM data processing section (page 23).